# Impact of upper mantle convection on lithosphere hyper-extension and subsequent horizontally forced subduction initiation

Lorenzo G. Candioti[1], Stefan M. Schmalholz[1], and Thibault Duretz[2,1]

[1]Institut des sciences de la Terre, Bâtiment Géopolis, Quartier UNIL-Mouline, Université de Lausanne, 1015 Lausanne (VD), Switzerland
[2]Univ Rennes, CNRS, Géosciences Rennes UMR 6118, Rennes, France

**Correspondence:** Lorenzo G. Candioti (Lorenzo.Candioti@unil.ch)

**Abstract.** Many plate tectonic processes, such as subduction initiation, are embedded in long-term (>100 Myrs) geodynamic cycles often involving subsequent phases of extension, cooling without plate deformation and convergence. However, the impact of upper mantle convection on lithosphere dynamics during such long-term cycles is still poorly understood. We have designed two-dimensional upper mantle-scale (down to a depth of 660 km) thermo-mechanical numerical models of coupled lithosphere-mantle deformation. We consider visco-elasto-plastic deformation including a combination of diffusion, dislocation and Peierls creep law mechanisms. Mantle densities are calculated from petrological phase diagrams (Perple_X) for a Hawaiian pyrolite. Our models exhibit realistic Rayleigh numbers between $10^6$ and $10^7$ and model temperature, density and viscosity structures agree with geological and geophysical data and observations. We tested the impact of the viscosity structure in the asthenosphere on upper mantle convection and lithosphere dynamics. We also compare models in which mantle convection is explicitly modelled with models in which convection is parameterized by Nusselt number scaling of the mantle thermal conductivity. Further, we quantified the plate driving forces necessary for subduction initiation in 2D thermo-mechanical models of coupled lithosphere-mantle deformation. Our model generates a 120 Myrs long geodynamic cycle of subsequent extension (30 Myrs), cooling (70 Myrs) and convergence (20 Myrs) coupled to upper mantle convection in a single and continuous simulation. Fundamental features such as the formation of hyper-extended margins, upper mantle convective flow and subduction initiation are captured by the simulations presented here. Compared to a strong asthenosphere, a weak asthenosphere leads to the following differences: smaller value of plate driving forces necessary for subduction initiation (15 TN m$^{-1}$ instead of 22 TN m$^{-1}$) and locally larger suction forces. The latter assists in establishing single-slab subduction rather than double-slab subduction. Subduction initiation is horizontally forced, occurs at the transition from the exhumed mantle to the hyper-extended passive margin and is caused by thermal softening. Spontaneous subduction initiation due to negative buoyancy of the 400 km wide, cooled exhumed mantle is not observed after 100 Myrs in model history. Our models indicate that long-term lithosphere dynamics can be strongly impacted by sub-lithosphere dynamics. The first-order processes in the simulated geodynamic cycle are applicable to orogenies that resulted from the opening and closure of embryonic oceans bounded by magma-poor hyper-extended rifted margins, which might have been the case for the Alpine orogeny.

# 1 Introduction

## 1.1 Convection in the Earth's mantle

In general, the term convection can be used to describe any motion of a fluid driven by external or internal forces (Ricard et al., 1989). Prout (1834) derived this term from the Latin word "convectio" (to carry, or to convey) to distinguish between advection dominated heat transfer and conduction, radiation dominated heat transfer.

On Earth, heat transfer through the lithosphere is dominated by thermal conduction while heat transfer through the underlying mantle is dominated by advection of material (e.g. Turcotte and Schubert, 2014). Convection may involve either the whole mantle, down to the core-mantle boundary, or only specific mantle layers. At temperature and pressure conditions corresponding to a depth of about 660 km, the mineralogy of peridotite changes from $\gamma$-spinel to perovskite + magnesiowüstite. This phase transition is endothermic, which means it has a negative pressure-temperature, so-called Clapeyron slope. Therefore, the penetration of cold slabs subducting into the lower mantle and hot plumes rising into the upper mantle may be delayed (Schubert et al., 2001). The 660-km phase transition can therefore represent a natural boundary, that separates two convecting layers. Laboratory experiments, tomographic images and calculations on the Earth's heat budget deliver evidence that a mixed mode of both types best explains convection in the present-day Earth's mantle (Li et al., 2008; Chen, 2016).

Any convecting system can be described by a dimensionless number, the so-called Rayleigh number. It is defined as the ratio of the diffusive and the advective time scale of heat transfer (see also, eq. B1). The critical value of the Rayleigh number necessary for the onset of convection in the Earth's mantle is typically in the order of 1000 (Schubert et al., 2001). Convection in the Earth's mantle can occur at Rayleigh numbers in the range of $10^6$-$10^9$ depending on the heating mode of the system and whether convection is layered or it includes the whole mantle (Schubert et al., 2001). The higher the Rayleigh number, the more vigorous is the convection, i.e. advection of material occurs at a higher speed. Vigour of both whole and layered mantle convection is inter alia controlled by the mantle density, the temperature gradient across and the effective viscosity of the mantle. However, unlike the density and thermal structures, the viscosity structure of the mantle is subject to large uncertainty. Viscosity is not a direct observable and can only be inferred by inverting observable geophysical data such as data for glacial isostatic adjustment (e.g. Mitrovica and Forte, 2004) or seismic anisotropy data (e.g. Behn et al., 2004). Especially at depths of ca. 100-300 km, in the so-called asthenosphere, the inferred value for viscosity varies greatly (see fig. 2 in Forte et al. (2010)). Values for effective viscosity in this region can be up to two orders of magnitude lower than estimates for the average upper mantle viscosity of $\approx 10^{21}$ Pa s (Hirth and Kohlstedt, 2003; Becker, 2017).

## 1.2 Long-term geodynamic cycle and coupled lithosphere-mantle deformation

Many coupled lithosphere-mantle deformation processes, such as the formation of hyper-extended passive margins and the mechanisms leading to the initiation of subduction (e.g. Peron-Pinvidic et al., 2019; Stern and Gerya, 2018), are still elusive. Crameri et al. (2020) compiled a database from recent subduction zones to investigate whether subduction initiation was vertically (spontaneous) or horizontally forced (induced, see also Stern (2004) for terminology). They concluded that, during the last ca. 100 Myrs, the majority of subduction initiation events were likely horizontally forced. Recent numerical studies

have investigated thermal softening as a feasible mechanism for horizontally forced subduction initiation (Thielmann and Kaus, 2012; Jaquet and Schmalholz, 2018; Kiss et al., 2020). In these models, horizontally forced subduction was initiated without prescribing a major weak zone cross-cutting through the lithosphere. These models do not require further assumptions on other softening mechanisms, such as micro-scale grain growth or fluid- and reaction-induced softening. Therefore, these models are likely the simplest to study horizontally forced subduction initiation.

Geodynamic processes, such as lithosphere extension or convergence, are frequently studied separately. In fact, many studies show that these processes are embedded in longer term cycles, such as the Wilson cycle (Wilson, 1966; Wilson et al., 2019). Over large time scales (>≈80 Myrs), tectonic inheritance of earlier extension and cooling events (Chenin et al., 2019) together with mantle convection (Solomatov, 2004) had presumably a major impact on subsequent convergence and subduction. Certainly, subduction initiation at passive margins during convergence can be studied without a previous extension and cooling stage (e.g., Kiss et al. (2020)). An initial passive margin geometry and thermal field must be then constructed ad-hoc for the model configuration. However, it is then uncertain whether the applied model would have generated a stable margin geometry and its characteristic thermal structure during an extension simulation. In other words, it is unclear whether the initial margin configuration is consistent with the applied model.

Plenty of numerical studies modelling the deformation of the lithosphere and the underlying mantle do not directly model convective flow below the lithosphere (e.g. Jaquet and Schmalholz, 2018; Gülcher et al., 2019; Beaussier et al., 2019; Erdös et al., 2019; Li et al., 2019). Ignoring convection below the lithosphere in numerical simulations is unlikely problematic, if the duration of the simulated deformation is not exceeding a few tens of millions of years. In such short time intervals, the diffusive cooling of the lithosphere is likely negligible. However, convection in the Earth's mantle regulates the long-term thermal structure of the lithosphere (Richter, 1973; Parsons and McKenzie, 1978) and has, therefore, a fundamental control on the lithospheric strength. Furthermore, mantle flow can exert suction forces on the lithosphere (e.g. Conrad and Lithgow-Bertelloni, 2002). Numerical studies show that these suction forces can assist in the initiation of subduction (Baes et al., 2018). Therefore, coupling mantle convection to lithospheric scale deformation can potentially improve our understanding of processes acting on long-term geodynamic cycles of the lithosphere.

Here, we present two-dimensional (2D) thermo-mechanical numerical simulations modelling the long-term cycle of coupled lithosphere-mantle deformation. The modelled geodynamic cycle comprises a 120 Myrs history of extension–cooling–convergence leading to horizontally forced subduction. We include the mantle down to a depth of 660 km assuming that convection is layered. Timings and deformation velocities for the distinct periods have been chosen to allow for comparison of the model results to the Alpine orogeny. With these models, we investigate and quantify the impact of (1) the viscosity structure of the upper mantle and (2) an effective conductivity parameterization on upper mantle convection and lithospheric deformation. Applying this parameterization diminishes the vigour of convection, but maintains a characteristic thermal field for high Rayleigh number convection (explained in more detail in the next section and in appendix B). (3) We also test creep law parameters for wet and dry olivine rheology. Finally, we investigate whether forces induced by upper mantle convection have an impact on horizontally forced subduction initiation.

## 2 The applied numerical model

The applied numerical algorithm solves the partial differential equations for conservation of mass and momentum coupled to conservation of energy. We consider the deformation of incompressible visco-elasto-plastic slowly flowing fluids under gravity (no inertia). The equations are discretized on a 2D finite difference staggered grid in the Cartesian coordinate system. Material properties are advected using a marker-in-cell method (Gerya and Yuen, 2003). A 4th order Runge-Kutta scheme is employed for marker advection and a true free surface is applied (Duretz et al., 2016a). A detailed description of the algorithm is given in the appendix A and the results of a community convection benchmark test are presented in appendix C. The applied algorithm has already been used to model processes at different scales, such as deformation of eclogites on the centimetre-scale (Yamato et al., 2019), crystal-melt segregation of magma during its ascent in a meter-scale conduit (Yamato et al., 2015), rifting of continental lithosphere (Duretz et al., 2016b; Petri et al., 2019), stress calculations around the Tibetan Plateau (Schmalholz et al., 2019) and within and around the subduction of an oceanic plate (Bessat et al., 2020), as well as modelling Precambrian orogenic processes (Poh et al., 2020). Here, we test the algorithm for capability of reproducing first order features of a geodynamic cycle namely: (i) formation of hyper-extended magma-poor rifted margins during a 30 Myrs extension period applying an absolute extension velocity of 2 cm yr$^{-1}$. (ii) Separation of the continental crust and opening of a ca. 400 km wide marine basin floored by exhumed mantle material. (iii) Generation of upper mantle convection during a 70 Myrs cooling period without significant plate deformation. (iv) Subsequent convergence for a period of 20 Myrs and horizontally forced subduction initiation in a self-consistent way, that means without modifying the simulation by, for example, adding ad-hoc a prominent weak zone across the lithosphere or changing material parameters. During convergence, the self-consistently evolved passive margin system is shortened applying an absolute convergence velocity of 3 cm yr$^{-1}$.

### 2.1 Modelling assumptions and applicability

For simplicity, we consider lithosphere extension that generates magma-poor hyper-extended margins and crustal separation leading to mantle exhumation. This means that we do not need to model melting, lithosphere break-up, mid-ocean ridge formation and generation of new oceanic crust and lithosphere. Such Wilson-type cycles, involving only embryonic oceans, presumably formed orogens such as the Pyrenees, the Western and Central Alps and most of the Variscides of Western Europe (e.g. Chenin et al., 2019). Values for deformation periods and rates in the models presented here are chosen to allow for comparison of the model evolution to the Alpine orogeny.

Further, tomographic images from the Mediterranean show large p-wave anomalies in the transition zone (Piromallo and Morelli, 2003) indicating that the 660-km phase transition inhibits the sinking material to penetrate further into the lower mantle. Therefore, we do not include the lower mantle into the model domain and assume that mantle convection is layered.

### 2.2 Model configuration

The model domain is 1600 km wide and 680 km high and the applied model resolution is 801×681 grid points (fig. 1). Minimum $z$-coordinate is set to -660 km and the top +20 km are left free (no sticky air) to allow for build-up of topography. The top

surface (initially at $z = 0$ km) is stress free. Thus, its position evolves dynamically as topography develops. Mechanical boundary conditions on the remaining boundaries are set to free slip at the bottom and constant material inflow/outflow velocities

at the left and right boundary. The boundary velocity is calculated such that the total volume of material flowing through the lateral boundary is conserved. The transition between inflow and outflow occurs at $z = -330$ km and not at the initially imposed lithosphere-asthenosphere boundary (LAB). Values for deviatoric stresses at this depth are significantly lower compared to those at the base of the lithosphere. This choice avoids boundary effects close to the mechanical lithosphere and the LAB can develop freely away from the lateral model boundaries. We use material flow velocity boundary condition rather than bulk

extension rates to deform the model units. Applying bulk extension rates and deforming the model domain would change the height of the model domain which has strong control on the Rayleigh number of the system (eq. B1). Also, evolution of passive margin geometries becomes dependent on the model width when using bulk extension rates as mechanical boundary condition (Chenin et al., 2018). It is therefore more practical to use constant velocity boundary conditions with material flow in the type of models presented here.

Initial temperature at the surface is set to 15 °C and temperatures at the crust-mantle (Moho) and at the LAB are 600 °C and 1350 °C, respectively. Assuming an adiabatic gradient of 0.49 °C km$^{-1}$ (see appendix B), the temperature at the model bottom is 1612 °C. Thermal boundary conditions are set to isothermal at the bottom and at the top of the domain and the left and right boundaries are assumed to be insulating (i.e. no heat flows through lateral boundaries). Model units include a 33 km thick, mechanically layered crust which overlies an 87 km thick mantle lithosphere on top of the upper mantle. The resulting initial

thickness of the lithosphere is thus 120 km. The crust includes three mechanically strong and four mechanically weak layers. The thickness of the weak layers is set to 5 km each, the thickness of the uppermost and lowermost strong layer is set to 4 km, whereas the strong layer in the middle is 5 km thick. This thickness variation allows to match the total 33 km thickness of the crust without introducing an additional vertical asymmetry. Mechanical layering of the crust was chosen, because it is a simple way of considering mechanical heterogeneities in the crust. The layering leads to the formation of numerous structural features

observed in natural hyper-extended passive margins (Duretz et al., 2016b) without relying on pre-defined strain softening.

    We consider viscous, elastic and brittle-plastic deformation of material in all models presented here. Viscous flow of material is described as a combination of several flow laws. We use dislocation creep for the crustal units and dislocation, diffusion and Peierls creep for mantle units (see also appendix A). The initial viscosity profile through the upper mantle is calibrated to match viscosity data obtained by Ricard et al. (1989). The applied flow law parameters lie within the error range of the corresponding

laboratory flow law estimates (see tab. A1).

    The difference between mantle lithosphere and upper mantle is temperature only, i.e. all material parameters are the same. Density of the crustal phases is computed with an equation of state (eq. A3), whereas density of the mantle phase is precomputed using Perple_X (Connolly, 2005) for the bulk rock composition of a pyrolite (Workman and Hart (2005), fig. A1). A detailed description of the phase transitions and how the initial thermal field is calibrated is given in appendix B. Surface

processes (e.g., erosion and sedimentation) are taken into account by a kinematic approach: if the topography falls below a level of 5 km depth or rises above 2 km height, it undergoes either sedimentation or erosion with a constant velocity of 0.5 mm yr$^{-1}$. In case of sedimentation, the generated cavity between the old and corrected topographic level is filled with

**Table 1.** Parameters varied in models M1-6.

| Parameter | Unit | M1 | M2 | M3 | M4 | M5 | M6 |
|---|---|---|---|---|---|---|---|
| $k$ | W m$^{-1}$ K$^{-1}$ | 2.75 | 2.75 | 2.75 | **36** | **36** | 2.75 |
| $\eta_{\text{cutoff}}$ | Pa s | $1\times10^{18}$ | **$1\times10^{20}$** | **$5\times10^{20}$** | $1\times10^{18}$ | **$1\times10^{20}$** | $1\times10^{18}$ |
| $Ra_{\text{avg}}$ | - | $9.95\times10^{6}$ | $3.92\times10^{6}$ | $1.18\times10^{6}$ | $1.97\times10^{6}$ | $5.48\times10^{5}$ | $9.26\times10^{7}$ |
| Rheology mantle | - | Dry Olivine | Dry Olivine | Dry Olivine | Dry Olivine | Dry Olivine | **Wet Olivine** |

$k$ is thermal conductivity and $\eta_{\text{cutoff}}$ is the lower viscosity limiter. $Ra_{avg}$ is the arithmetic average of Rayleigh numbers >1000. Rayleigh numbers are computed locally at each cell center according to eq. B1 after 99 Myrs in model history for models M1-5 and after 26 Myrs for model M6. Bold font highlights the parameters varied compared to the reference model M1.

sediments, alternating between calcites and pelites every 2 Myrs. This simple parameterization of surface processes moderates the amplitude of topography and may affect geodynamic processes such as subduction.

## 2.3 Investigated parameters

Among the physical parameters controlling the long-term geodynamic evolution (f.e., mantle density, plate velocities), the viscosity structure of the mantle is one of the least constrained. Therefore, we test (1) models with different upper mantle viscosity structures. In model M1, the reference model, the asthenosphere is assumed to be weak with values for viscosity in the order of $\approx 10^{19}$ Pa s resulting from the applied flow laws. In models M2 and M3 the asthenosphere is assumed to be stronger. Values for viscosities in the asthenosphere are limited by a numerical cut-off value of $1\times10^{20}$ Pa s in M2 and $5\times10^{20}$ Pa s in M3. Coupling of lithosphere-mantle deformation is achieved by resolving numerically both lithospheric deformation and upper mantle convection in M1-3. (2) We further test the impact of parameterizing convection on the cycle by scaling the thermal conductivity to the Nusselt number of upper mantle convection (see app. B). In these models, we also assume both a weak asthenosphere (as in M1) in model M4 and a strong asthenosphere (as in M2) in model M5. The effective conductivity approach has been used, for example, in mantle convection studies for planetary bodies when convection in the mantle is too vigorous to be modelled explicitly (e.g. Zahnle et al., 1988; Tackley et al., 2001; Golabek et al., 2011). Also, it has been used in models of back-arc lithospheric thinning through mantle flow that is induced by subduction of an oceanic plate (e.g. Currie et al., 2008) and in models of lithosphere extension and subsequent compression (e.g. Jammes and Huismans, 2012). (3) We finally investigate the role of the olivine rheology. To this end, we perform an additional model M6 in which the material parameters of the dislocation and diffusion creep mechanism of a dry olivine rheology is replaced by the parameters of a wet olivine rheology (table A1). In M6, values for all the other parameters, both physical and numerical, are initially equal to those set in M1. Within the error range of values for activation volume and energy of the wet olivine rheology, the viscosity is calibrated to the data obtained by Ricard et al. (1989). However, using the highest possible values for the wet olivine flow law parameters, the maximum viscosity in the upper mantle is initially one order of magnitude lower compared to models M1-5. A summary of all simulations is given in table 1 and all material parameters are summarised in table A1.

## 3 Results

We first describe the evolution of the reference model M1 and of M6. Model M6 is stopped after the extensional stage, because its later evolution is not applicable to present-day Earth. Thereafter, we compare the results of M2-5 to the results of M1 for the individual deformation stages. Finally, we compare the evolution of the plate driving forces in models M1-5 during the entire cycle. The arithmetic average Rayleigh number in all models is significantly larger compared to the critical Rayleigh number $Ra_{crit}$ = 1000. The rifting and cooling period laterally perturb the thermal field sufficiently to initiate and drive the convection over large time scales in all models presented here.

### 3.1 Dry Olivine rheology: Model M1 - Reference run

Crustal break-up during the rifting phase in M1 occurs after ca. 8 Myrs (fig. 3(a) & (e)). The left continental margin has a length of ca. 200 km and the right margin has a length of ca. 150 km (fig. 3(e)). Velocity arrows indicate upward motion of hot material in the centre of the domain (fig. 3(e)). Two convection cells begin to establish at this stage below each margin. The viscosity of the upper mantle decreases to minimal values in the order of $10^{19}$ Pa s and increases again up to values in the order of $10^{21}$ Pa s at the bottom of the model domain (fig. 3(a) and fig. 2(c)). Towards the end of the cooling period (at 97 Myrs), M1 has developed circular shaped convection cells in the upper region of the upper mantle (above $z \approx$ -400 km, fig 3(b)). The average Rayleigh-number (see footnote of tab. 1) of the system computed at this late stage of the cooling period is ca. $9.95 \times 10^6$ and the size of the cells varies between ca. 50 km and ca. 300 km in diameter below the left and right margin respectively (fig.3(b)). Below the right margin at $z \approx$ -150 km and $x \approx$ +300 km the downward directed mantle flow of two neighbouring convection cells unifies (fig. 3(f)). The top ca. 100 km of the modelled domain remain undeformed; no material is flowing in this region (area without velocity arrows in fig. 3(f)). Convergence starts at 100 Myrs and at ca. 102 Myrs, a major shear zone forms breaking the lithosphere below the right margin (inclined zone of reduced effective viscosity in fig. 3(c) & (g)). Velocity arrows in the lithosphere indicate the far-field convergence, whereas velocity arrows in the upper mantle show that convection cells are still active. The exhumed mantle is subducted in one stable subduction zone below the right continental margin. Several convection cells are active in the upper mantle during subduction (see velocity arrows fig. 3(d) & (h)). A trench forms in which sediments are deposited (fig.3(d) & (h)). Folding of the crustal layers in the overriding plate indicates significant deformation of the crust. The crustal layers of the subducting plate remain relatively undeformed. The viscosity in the asthenosphere remains stable at values of $10^{19}$ Pa s during the entire model history.

### 3.2 Wet Olivine rheology: Model M6

Crustal necking in M6 starts at ca. 2 Myrs in model history (fig. 4(a) & (e)). Two convecting cells develop in the horizontal centre of the domain transporting material from $z \approx$ -200 km to $z \approx$ -100 km (fig. 4(b) & (f)). At ca. 13 Myrs convection cells are active in the upper 500 km of the domain (fig. 4(c)). Crustal thickness varies laterally between ca. 20 km and <5 km (fig. 4(g)). The mantle lithosphere is thermally eroded, indicated by a rising level of the $10^{21}$ Pa s contour in fig. 4(f)-(h) after 26 Myrs in model history. In contrast to M1, M6 does not reach the stage of crustal break-up (fig. 4(d) & (h)) within 30 Myrs. Two large

convection cells are active: in the left half of the domain convection occurs at relatively enhanced flow speeds, whereas in the right half of the domain flow speeds are relatively lower (compare relative length of velocity arrows in fig. 4(d)). The average Rayleigh number of the system at this stage is $9.26 \times 10^7$. Values for temperature at the Moho reach ca. 1000 °C locally (fig. 4(h) and fig. 2(a)). The horizontally-averaged density profile (fig. 2(b)) shows that values for density in the lithosphere are on average 100 kg m$^{-3}$ lower in M6 compared to M1. Values for effective viscosity in the asthenosphere decrease to minimal values in the order of $10^{18}$ Pa s and increase up to values of $10^{19}$ Pa s at a depth of 660 km at the end of the extension period (fig. 2(c)).

## 3.3 Comparison of reference run with models M2-5: Extension phase

In contrast to M1, M2 produces two conjugate passive margins that are both approximately 150 km long (fig. 5(b)) after 13 Myrs. Crustal separation has not occurred up to this stage in M3: the two passive margins are still connected by a crustal bridge of ca. 10 km thickness. Mantle material rises below the centre of the domain and then diverges below the plates in M2 and M3, but no convection cells have formed yet (fig. 5(b) & (c)), which is different compared to M1. The minimal value for effective viscosity in the upper mantle is at the applied cut-off value of $1 \times 10^{20}$ Pa s in M2 and $5 \times 10^{20}$ Pa s in M3 and increases up to ca. $1 \times 10^{21}$ Pa s at a depth of 660 km (fig. 2(c)) in both models. Similar to the reference model (M1), in M4, the left continental margin has a length of ca. 200 km and the right margin has a length of ca. 150 km (fig. 5(d)). Like in M1, two convection cells have formed below the two passive margins in this model (see arrows in fig. 5(d)) and the minimal value for effective viscosity below the lithosphere is in the order of $1 \times 10^{19}$ Pa s and increases to approximately $1 \times 10^{21}$ Pa s at a depth of 660 km (fig. 2(c)). Both margins in M5 are approximately equally long (ca. 150 km, fig. 5(e)) and values for viscosity are at the lower cut-off value of $1 \times 10^{20}$ Pa s in the upper mantle and increase up to ca. $1 \times 10^{21}$ Pa s at a depth of 660 km (fig. 2(c)). The overall evolution of the extension period in M5 is more similar to M2 than to M1. In M1-5, the 1350 °C isotherm does not come closer than ca. 30 km to the surface. Horizontally-averaged vertical temperature profiles are similar in M1-5 (fig. 2(a) & (g)). The level of the 1350 °C isotherm remains at its initial depth in M4 and M5, whereas it subsides by ca. 20 km in M1-3. Horizontally-averaged density profiles (fig. 2(b)) show density differences of <10 kg m$^{-3}$ between ca. 35 km and ca. 120 km depth.

## 3.4 Comparison of reference run with models M2-5: Cooling phase without plate deformation

Models M1-5 maintain a stable lithospheric thickness of ca. 90-100 km over 100 Myrs (top magenta viscosity contour in fig. 6) and no thermal erosion of the lithosphere occurs. Below, the upper mantle is convecting at decreasing Rayleigh numbers from M1-5. In M1, the vertical mantle flow speed within the convecting cell at $x \approx +350$ km is elevated (indicated by darker blue coloured region in fig. 6(a)) compared to the average flow speed of $\approx$1-2 cm yr$^{-1}$ in neighbouring cells). The size of the convection cells in M2 is larger compared to M1 and in the order of ca. 100-300 km in diameter. Characteristic is the more elliptical shape of the cells compared to the circular cells in M1. The magnitude of material flow velocity is similar but distributed more horizontally symmetric below both margins compared to M1 (compare arrows and colour field of fig.6(a) & (b)). A zone of strong downward directed movement develops below both margins (dark blue cells at $x \approx$ -300 km and $x$

$\approx$ +300 km in fig.6(b)). Magnitude of material flow speed is in the order of 1.5 cm yr$^{-1}$. The average Rayleigh-number of the convecting system in M2 is approximately $3.92 \times 10^6$ and is about a factor 2.5 lower compared to M1. M3 develops four large convection cells, two below each margin, that are active up to depths of approximately 600 km (see arrows fig. 6(c)). Downward directed movement of material occurs with ca. 2 cm yr$^{-1}$ (darker blue regions at $x \approx$ -500 km and $x \approx$ 500 km in fig.6(c)). The average Rayleigh number of the system is ca. $1.18 \times 10^6$, which is about a factor 8.4 smaller compared to M1. In M4 and M5, material transport occurs at absolute values for vertical velocity <0.5 cm yr$^{-1}$ (see coloured velocity field in fig. 6(d) & (e)) which is one order of magnitude lower compared to M1-3. Two horizontally symmetric convection cells develop in these models that are active between $z \approx$ -150 km and $z \approx$ -400 km (arrows in fig. 6(d) & (e)). The average Rayleigh number is ca. $1.97 \times 10^6$ and $5.48 \times 10^5$ in M4 and M5, respectively. Figure 6(f)-(j) shows the difference between the entire density field and the horizontally averaged vertical density profile at 99 Myrs (see fig. 2(e)) and fig. 7(b) shows a horizontal profile of this field averaged vertically over $-200 \leq z \leq -100$ km. The distribution of density differences becomes more horizontally symmetric with decreasing Rayleigh number (M1-5, see fig. 7(b)). In M1, the 2 kg m$^{-3}$ density contour line encloses a high-density anomaly below the right passive margin (see black contour line at $x \approx$ 300 km, $z \approx$ -200 km in fig. 6(f)). Calculating the suction, or buoyancy, force for this body according to eq. D4 yields a value of 0.25 TN m$^{-1}$. Since the distribution of density differences is laterally symmetric in M2-5 (fig. 7(b)) and defining an integration area is not trivial, a calculation of suction forces is not attempted for M2-5. Figure 7(a) shows the topography at ca. 99 Myrs in model history, that is 1 Myrs before the start of convergence. Topography does not exceed 1.5 km and the average depth of the basin is ca. 3.75 km.

Figure 8(a)-(e) shows the conductive heat flow of the entire domain in absolute values. M1-5 reproduce a heat flux of 20-30 mW m$^{-2}$ through the base of the lithosphere (indicated by the $10^{21}$ Pa s isopleth at a depth between 100-110 km). The conductive heat flow below the lithosphere is close to 0 mW m$^{-2}$ in M1-3. In M4 and M5, values for conductive heat flow remain at ca. 20 mW m$^{-2}$ through the entire upper mantle. Density differences in the upper part of the mantle lithosphere reach ca. 20 kg m$^{-3}$ between ca. 35-120 km in depth (fig. 2(e) & (j)). Values for effective viscosity range in the order of $10^{19}$ Pa s for M1 and M5 and in the order $10^{20}$ Pa s in M2-4 directly below the lithosphere and $10^{21}$ Pa s at the bottom of the upper mantle (fig. 2(f)).

### 3.5 Comparison of reference run with models M2-5: Convergence and subduction phase

In contrast to M1, two major symmetric shear zones develop in the lithosphere in M2, M3 and M5, one shear zone at each of the continental margins (fig.9(b), (c) & (e)). Like in M1, one shear zone forms below the right margin in M4 (fig. 9(d)). At this early stage of subduction initiation, the strain rate in the shear zone is in the order $10^{-14}$-$10^{-13}$ s$^{-1}$. In the region of the shear zones, the temperature is increased, which is indicated by the deflection of the isotherms (red contour lines in fig.9(a)-(e)). Horizontal profiles of gravitational potential energy (*GPE*, see appendix D) are in general similar (fig. 7(c)) for M1-5.

In contrast to the single-slab subduction evolving in M1, double-slab subduction is observed below both margins in M2-5 and sediments are deposited in two trenches as the subduction evolves (fig.10(b)-(e)). Folding of the crustal layers indicates deformation in both margins of M2-5. Values for viscosity in the upper mantle remain stable at values of $10^{19}$ Pa s in M4, whereas the viscosity values are at the applied lower cut-off value in M2, M3 and M5 throughout the entire simulation history.

Convection cells remain active during subduction. Similar to M1, a stable convection cell is active below one subducting slab in M4. The distribution of convection cells is more symmetric in front of the slabs in M2, M3 and M5 compared to M1. Defined by the applied boundary condition, material flows into the model domain up to $z = -330$ km at the lateral boundaries in all models (see section 2.2). Though, far away from the boundary in the centre of the domain in M1 and M2 the horizontal material inflow from the lateral boundaries is limited up to a depth of ca. 100-150 km (see horizontally directed arrows in fig. 10(a) & (b)). In M3-5 the lateral inflow of material reaches a depth of about 200 km in the centre of the domain (fig. 10(c)-(e)).

### 3.6 Estimates for plate driving forces

The vertically-integrated second invariant of the deviatoric stress tensor, $\bar{\tau}_{\mathrm{II}}$, is a measure for the strength of the lithosphere and twice its value is representative for the horizontal driving force (per unit length, $F_{\mathrm{D}}$ hereafter) during lithosphere extension and compression (appendix D). Figure 7(d) shows the evolution of $F_{\mathrm{D}}$ during the entire cycle. During the pure shear thinning phase in the first ca. 2 Myrs of extension, values for $F_{\mathrm{D}}$ reach 14 TN m$^{-1}$. At ca. 2-3 Myrs (1 in fig. 7(d)), this value drops below ca. 5 TN m$^{-1}$ at ca. 8 Myrs. At the end of the extension period $F_{\mathrm{D}}$ is stabilised at values between ca. 2-3 TN m$^{-1}$ for all models (2 in fig. 7(d)). This value remains relatively constant for all models during the entire cooling period. The maximum value for $F_{\mathrm{D}}$ necessary to initiate subduction in all models is observed in M3 and is ca. 23 TN m$^{-1}$ (thin blue curve in fig. 7(d)). The minimum value necessary for subduction initiation of ca. 13 TN m$^{-1}$ is observed in M4. The reference run M1 initiates subduction with a value of $F_{\mathrm{D}} \approx 17$ TN m$^{-1}$. Strain localization at ca. 102 Myrs is associated with a rapid decrease of $F_{\mathrm{D}}$ by ca. 2-5 TN m$^{-1}$ in all models (3 in fig. 7(d)). At ca. 105 Myrs, values for $F_{\mathrm{D}}$ increase again until the end of the simulation.

## 4 Discussion

### 4.1 Impact of mantle viscosity structure and effective conductivity on passive margin formation

Higher values for the lower viscosity cut-off (M2 & M3 compared to M1) do not only change the viscosity structure of the mantle, but also increase the strength of the weak layers. In consequence, the multi-layered crust necks effectively as a single layer. The resulting passive margins are slightly shorter and more symmetric (Duretz et al., 2016b, see also M3). The highest cut-off value of $5 \times 10^{20}$ Pa s (M3) leads to a two-stage necking as investigated by Huismans and Beaumont (2011). First, the lithosphere is necking while the crust deforms by more or less homogeneous thinning, leading to the development of a large continuous zone of hyper-extended crust, below which the mantle lithosphere has been removed. Second, the hyper-extended crust is breaking up later than the continental mantle lithosphere. During the rifting stage, the thermal field in all simulations is very similar (see fig. 2(a)), presumably because effects of heat loss due to diffusion are not significant over the relatively short time scale.

## 4.2 Onset of upper mantle convection and thermo-mechanical evolution of the lithospheric plates

Rifting in M1-5 causes up-welling of hot asthenospheric material in the horizontal centre of the domain. The resulting lateral thermal gradients are high enough to induce small-scale convection (Buck, 1986). Huang et al. (2003) derived scaling laws to predict the onset time for small-scale convection in 2D and 3D numerical simulations. They investigated the impact of plate motion, layered viscosity, temperature perturbations and surface fracture zones on the onset time. The observed onset time in layered viscosity systems becomes larger when the thickness of the weak asthenosphere decreases. Also, plate motion can delay the onset of small-scale convection. In contrast, fracture zones at the surface may lead to earlier onset of convection depending on the thermal structure of the fracture zones. Using their scaling law and the parameters used in our reference model M1, onset time of convection is predicted for ca. 43 Myrs. However, onset of convection in M1 is observed as early as ca. 8 Myrs (see fig. 3) which is consistent with onset times observed in numerical simulations conducted by Van Wijk et al. (2008). There may be several reasons for the discrepancy between the prediction and observation. First, the models presented here are likely a combination of the configurations tested by Huang et al. (2003). Second, the choice of boundary conditions and initial configuration is different which is probably important when testing the impact of plate motion on the onset time. They located the rift centre at the left lateral boundary, whereas the rift centre in our models is located far away from the lateral boundaries in the horizontal centre of the domain. This likely impacts the flow direction of hot material ascending beneath the rift centre and, consequently, alters lateral thermal gradients which are important for the onset of convection. The onset time for convection is delayed and at ca. 20 Myrs in M2 and at ca. 30 Myrs in M3. This delay is likely due to the increased viscosity in the asthenosphere in M2 & M3 compared to M1 which decreases the Rayleigh number of the system. This observation is in agreement with the general inverse proportionality of the onset time of convection to the Rayleigh number as predicted by Huang et al. (2003).

Once convection has started, it stabilises the temperature field (Richter, 1973; Parsons and McKenzie, 1978) over large time scales and controls the thickness of the lithosphere. Although we allow for material inflow up to $z$ = -330 km, the arrows in fig. 10(a) & (b) indicate that lateral inflow of material far away from the boundary is limited up to z $>\approx$ -150 km. Below this coordinate, the convection cells transport material even towards the lateral boundary, in the opposite direction of material inflow. This observation suggests, that the thermal thickness of the lithosphere adjusts self-consistently and far away from the boundary during the geodynamic cycle. The thermal thickness of the lithosphere seems to vary with decreasing Rayleigh number: in M5 the value for the thermal thickness of the lithosphere is $\approx$ 200 km (see arrows in fig. 10(e)). This observation suggests that the thermal thickness of the moving lithosphere is presumable regulated by the vigour of convection.

For realistic values for thermal conductivity in the upper mantle (M1-3), heat flow through the base of the mechanical lithosphere is between 20-30 mW m$^{-2}$ (fig. 8(a)-(c)). These values are in the range of heat flow estimates for heating at the base of the lithosphere confirmed by other numerical studies (Petersen et al., 2010; Turcotte and Schubert, 2014). Below the lithosphere, the conductive heat flux is essentially zero, because heat transport is mainly due to advection of material in the convecting cells. The effective conductivity approach maintains a reasonable heat flux directly at the base of the lithosphere (figs. 8(d) & (e)) and convection cells develop. However, all processes in the upper mantle are conduction dominated (see

elevated values for $q_z$ in fig. 8(d) & (e)). This implies that the characteristic physics of mantle convection are not captured correctly in these models. This becomes evident when comparing fig. 6(a) to fig. 6(d): although the physical parameters - except the thermal conductivity in the upper mantle - are the same in both simulations and temperature profiles are similar (see fig. 2(d) & (i)), the Rayleigh-numbers of the systems differ by one order of magnitude (see tab. 1) and the convective patterns are entirely different.

### 4.3 Mantle convection, thermal erosion and tectonics in the Archean

Model M6 underlines the importance of better constraining the rheology of the mantle. Due to significantly reduced viscosities (see fig. 2(c)), convection in the upper mantle occurs at an average Rayleigh number of ca. $9 \times 10^7$ in M6. Compared to estimates for present-day Earth's upper mantle convection (Torrance and Turcotte, 1971; Schubert et al., 2001) this Rayleigh-number is an order of magnitude higher. The lithosphere is recycled rapidly and the resulting values for density directly below the lithosphere are ca. $100 \ \mathrm{kg \ m^{-3}}$ lower in M6 compared to M1-5. The resulting density structure in the upper region of the mantle deviates significantly from the PREM model (Dziewonski and Anderson (1981), see fig. 2(b)). Convection at such high Rayleigh numbers leads to an enhanced temperature field. Resulting values for temperature at the Moho locally reach ca. 1000 °C. Our models suggest that such weak mantle rheology is actually not feasible for present-day plate tectonics, but this non-feasibility can only be observed when performing coupled mantle-lithosphere models as we have presented. In lithosphere-only models, a weak mantle would not generate the thermal erosion of the lithosphere bottom as observed in model M6, because the lithosphere bottom would be "stabilized" by the bottom boundary condition. In the Archean eon, the mantle potential temperature was probably 200-300 °C higher (Herzberg et al., 2010) and therefore convection was more vigorous (Schubert et al., 2001). Agrusta et al. (2018) have investigated the impact of variations in mantle potential temperature compared to present-day's value on slab-dynamics. Assuming a 200 °C higher value for the mantle potential temperature compared to present-day estimates leads to viscosity structures similar to the average viscosity structure we report for M6 (compare fig. 6c in Agrusta et al. (2018) to fig. 2(c) of this study). Hence, we argue that such vigorously convecting systems as presented in M6 may be applicable to the mantle earlier in Earth's history.

### 4.4 Spontaneous vs. induced subduction initiation and estimates for plate driving forces

Currently, the processes and tectonic settings leading to subduction initiation remain unclear (Stern, 2004; Stern and Gerya, 2018; Crameri et al., 2019). Stern (2004) proposed two fundamental mechanisms for subduction initiation:

(1) Spontaneous (or vertically forced, Crameri et al. (2020)) subduction initiation occurs, for example, due to densification of the oceanic lithosphere during secular cooling. Cloos (1993) proposed that the density increase of cooling, 80 $\mathrm{Myrs}$ old oceanic lithosphere compared to the underlying asthenosphere is in the order of $40 \ \mathrm{kg \ m^{-3}}$. According to Cloos (1993), this difference is sufficient to initiate subduction spontaneously by negative buoyancy of the oceanic lithosphere (see Stern (2004); Stern and Gerya (2018, & references therein) for detailed explanation). However, McKenzie (1977) and Mueller and Phillips (1991) showed that the forces acting on the lithosphere due to buoyancy contrasts are not high enough to overcome the strength

of the cold oceanic lithosphere. Observations of old plate ages (>100 Myrs) around passive margins in the South Atlantic (Müller et al., 2008) indicate their long-term stability.

In the models presented here, the applied thermodynamic density of the Hawaiian pyrolite leads to density differences between the exhumed mantle in the basin and the underlying asthenosphere of ca. $10\text{-}20\,\mathrm{kg\,m^{-3}}$ after 99 Myrs in model history. These buoyancy contrasts are ca. $2\times$ smaller than the contrast proposed by Cloos (1993). Boonma et al. (2019) calculated a density difference ($\rho_{\mathrm{ast}} - \rho_{\mathrm{lit}}$) of $+19\ \mathrm{kg\ m^{-3}}$ for an 80 km thick continental lithosphere and $-17\ \mathrm{kg\ m^{-3}}$ for an oceanic lithosphere of 120 Myrs of age. These values are in agreement with the values we report in our study. In the 2D models presented here, these buoyancy contrasts are insufficient to overcome the internal strength of the cooled exhumed mantle and initiate subduction at the passive margin spontaneously.

However, modelling spontaneous subduction initiation for an ad-hoc constructed passive margin geometry is possible, if the employed mechanical resistance is small, the density difference between lithospheric and oceanic mantle is large, and/or if an additional weak zone is imposed. Such passive margin configurations are indeed unstable and lead to spontaneous subduction initiation within a few million years (Stern and Gerya, 2018, & references therein). However, the passive margins we consider in our study have been stable for at least 60 Myrs before subduction initiation. Therefore, spontaneous subduction initiation for unstable passive margins is in contrast with the observation of long-term stability of the ancient Alpine Tethys margins (McCarthy et al., 2018) and the recent passive margins in the South Atlantic (Müller et al., 2008). Our results are, hence, in agreement with the stability of these passive margins. Modelling long-term geodynamic cycles, applicable to the evolution of the Alpine Tethys and the South Atlantic, requires appropriate density and rheological models which generate passive margins that are stable for more than 60 Myrs (Alpine Tethys) or more than 180 Myrs (South Atlantic). To evaluate whether models of spontaneous SI at passive margins are feasible, these models need to explain why the passive margins have been stable for more than 60 Myrs and only afterwards "collapsed" spontaneously, although they are cooled and mechanically strong.

(2) Induced (or horizontally forced, Crameri et al. (2020)) subduction initiation occurs, for example, due to far-field plate motion. In fact, many numerical studies that investigate subduction processes do not model the process of subduction initiation. In these studies, a major weak zone across the lithosphere is usually prescribed ad-hoc in the initial model configuration to enable subduction (Ruh et al., 2015; Zhou et al., 2020). Another possibility to model subduction is to include a prescribed slab in the initial configuration. This means that subduction has already initiated at the onset of the simulation (Kaus et al., 2009; Garel et al., 2014; Holt et al., 2017; Dal Zilio et al., 2018). In our models, subduction is initiated self-consistently, which means here: (i) we do not prescribe any major weak zone or an already existing slab, and (ii) the model geometry and temperature field at the onset of convergence (100 Myrs) were simulated during a previous extension and cooling phase with the same numerical model and parameters. Subduction is initiated during the initial stages of convergence and subduction initiation is, hence, induced by horizontal shortening. Notably, Crameri et al. (2020) analysed more than a dozen documented subduction zone initiation events from the last hundred million years and found that horizontally forced subduction zone initiation is dominant over the last 100 Myrs. During convergence in our models, shear heating together with the temperature and strain rate dependent viscosity formulation (dislocation creep flow law, eq. A7) causes the spontaneous generation of a lithosphere-scale shear zone that evolves into a subduction zone (Thielmann and Kaus, 2012; Kiss et al., 2020). However, shear heating is

a transient process which means the increase in temperature is immediately counterbalanced by thermal diffusion. Efficiency
of shear heating is restricted to the first ca. 2-3 Myrs after shear zone formation in the presented models. After this time span,
heat generated by mechanical work is diffused away. Therefore, thermal softening is unlikely the mechanism responsible for
stabilisation of long-term subduction zones, but likely important for initiating and triggering subduction zones.

The value of $F_D$ (see appendix D) represents the plate driving force. In M1, the maximal value of $F_D$ just before stress drop
caused by subduction initiation at $\approx$103 Myrs is $\approx$17 TN m$^{-1}$ (fig. 7(d)). However, $F_D \approx 2$ TN m$^{-1}$ at the end of the cooling
stage, resulting from stresses due to mantle convection and lateral variation of *GPE* between continent and basin. Therefore, this
value can be subtracted from the $\approx$17 TN m$^{-1}$. The required plate tectonic driving force for convergence-induced subduction
initiation is then $\approx$15 TN m$^{-1}$ for M1. Kiss et al. (2020) modelled thermal softening induced subduction initiation during
convergence of a passive margin, whose geometry and thermal structure was generated ad-hoc as initial model configuration.
Their initial passive margin structure was significantly less heterogeneous than ours. To initiate subduction, they needed a
driving force of $\approx$37 TN m$^{-1}$ which is significantly larger than the $\approx$15 TN m$^{-1}$ required in our model M1. Obviously, the
$\approx$15 TN m$^{-1}$ cannot be exceeded by the mantle convection ($\approx 2$ TN m$^{-1}$) modelled here. Even assuming an additional ridge
push force of 3.9 TN m$^{-1}$ (Turcotte and Schubert, 2014) would not be sufficient to initiate subduction spontaneously in the
models presented here. The boundary convergence providing the remaining ca. 9 TN m$^{-1}$ to initiate subduction in our models
are assumed to be caused by far-field plate driving forces. These are generated by global processes that are not modelled inside
our domain, such as slab pull, whole mantle convection, ridge push etc. For example, the closure of the Piemonte-Liguria basin
is assumed to be caused by the much larger scale convergence of the African and European plates (McCarthy et al., 2018).

We suggest that mechanical and geometrical heterogeneity inherited by previous deformation periods and thermal hetero-
geneity due to mantle convection reduces the required driving force necessary for subduction initiation by thermal softening.
Additionally, we suppose that the plate driving force necessary for horizontally forced subduction initiation in our models
could be further reduced by considering more heterogeneities, or a smaller yield stress in the mantle lithosphere. However,
the minimum value for $F_D$ which can still generate horizontally forced subduction initiation via thermal softening has to be
quantified in future studies. This is relevant, because the main argument against thermal softening as an important localization
mechanism during lithosphere strain localization and subduction initiation is commonly that the required stresses, and hence
driving forces, are too high. In nature, more softening mechanisms act in concert with thermal softening, such as grain damage
(e.g. Bercovici and Ricard, 2012; Thielmann and Schmalholz, 2020), fabric and anisotropy evolution (e.g. Montési, 2013) or
reaction-induced softening (e.g. White and Knipe, 1978), likely further reducing forces required for subduction initiation. Fur-
thermore, Mallard et al. (2016) showed with 3D spherical full-mantle convection models that a constant yield stress between
150 and 200 MPa in their outer boundary layer, representing the lithosphere, provides the most realistic distribution of plate
sizes in their models. The yield stress in their models corresponds to a deviatoric, von Mises, stress which is comparable to
the value of $\tau_{II}$ calculated for our models. If we assume a 100 km thick lithosphere, then $F_D = 15$ TN m$^{-1}$ yields a vertically-
averaged deviatoric stress, $\tau_{II}$, of 75 MPa for our model lithosphere. Therefore, vertically-averaged deviatoric stresses for our
model lithosphere are even smaller than deviatoric, or shear, stresses employed in global mantle convection models. Based on

the above-mentioned arguments, we propose that $\approx$15 TN m$^{-1}$ is a feasible value for the horizontal driving force per unit length.

## 4.5 Mantle convection stabilising single-slab subduction

Figure 7(c) shows the difference in *GPE* at 103 Myrs. The horizontal profiles do not reveal significant differences between the models. The *GPE* is sensitive to topography and density distribution throughout the model domain. Values for density differences are ca. 10-20 kg m$^{-3}$ across the mantle lithosphere. When integrated over the entire depth, such variations do not significantly impact the *GPE* profiles. Instead, the signal reflects the topography, which is similar in all the models due to the similar margin geometry (see fig. 7(a) and fig. 5(a)-(e)). In consequence, stress concentrations lead to localised deformation and shear zone formation at both margins in all models during the onset of convergence. Therefore, the evolution and stabilisation of a single-sided subduction requires an additional asymmetry of the system.

The average Rayleigh number (see appendix eq. B1) in M1 is ca. $9.95\times10^6$ (see tab. 1) which is close to estimated values for upper mantle convection (Torrance and Turcotte, 1971; Schubert et al., 2001). Potential lateral asymmetry caused by the convecting cells in the upper mantle is inherited from the cooling period. The vertical velocity field at the end of the cooling period of M1 (fig. 6(a)) reveals one convection cell at $x\approx$+350 km with flow speeds in the order of the convergence velocity applied later. This convection cell is induced by a high-density anomaly directly below the passive margin at which subduction will be initiated later in the model (see black contour line below right margin in fig. 6(f)). This sinking, high-density body probably induces an asymmetry in form of an additional suction force exerted on the lithospheric plate. Conrad and Lithgow-Bertelloni (2002) quantified the importance of slab-pull vs. slab-suction force and showed that the slab pull force and the suction force of a detached slab sinking into the mantle induces similar mantle flow fields. They argued that a detached fraction of a slab sinking into the mantle can exert shear traction forces at the base of the plate and drive the plate. Baes et al. (2018) showed with numerical simulations that sinking of a detached slab below a passive margin can contribute significantly to the initiation of subduction. We suggest that the high-density anomaly observed in M1 and the associated mantle flow also generates a force similar to a slab-suction force. To quantify the suction force induced by the sinking, high-density body in M1, we calculated the difference of the density in this region with respect to the horizontally averaged density field presented in fig. 2(e) and integrated this buoyancy difference spatially over the area enclosed by the 2 kg m$^{-3}$ density contour below the right passive margin (see fig. 6(f)). The resulting buoyancy force per unit length is ca. 0.25 TN m$^{-1}$. This value is relatively low compared to the plate driving forces acting in the model, but likely induces a sufficiently high asymmetry to stabilise the single-slab subduction in simulation M1.

Values for the Rayleigh number in M2 and M3 are lower ($3.92\times10^6$ and $1.18\times10^6$ respectively) compared to M1 (see tab. 1), because of the relatively higher viscosity (see eq. B1). With decreasing Rayleigh-number the asymmetry of convection also decreases (Schubert et al., 2001). In consequence, the size of the convecting cells becomes larger and more elliptic (M2 compared to M1) and the number of active cells decreases (M3 compared to M1). Enhancing the thermal conductivity (included in the denominator in eq. B1) by ca. one order of magnitude in models M4 and M5 also decreases the Rayleigh number by one order of magnitude compared to M1 (see table 1). Decreasing the Rayleigh-number (M2-5) leads to more laterally symmetric

convective flow patterns and decreases the speed and distribution of material flow (see fig. 6(b)-(e)). In turn, this probably leads to more equally distributed density differences (fig. 7(b)) and, therefore, suction forces exerted on the plates. Hence, we argue that decreasing the Rayleigh-number, assuming a relatively strong asthenosphere or applying an enhanced thermal conductivity likely favours the initiation of divergent double-slab subduction. The resulting slab geometries resemble a symmetric push-down (M2-5), rather than a stable asymmetric subduction (M1). This impacts the deformation in the lithosphere, especially in the crust: the crustal layers of both plates are strongly folded (see fig. 10(b)-(e)). In M1, the deformation of the crustal layers only occurs in the overriding plate.

## 4.6    Comparison with estimates of Earth's mantle viscosity and thermal structure

To apply our models to geodynamic processes on Earth, we compare several model quantities with measurements and indirect estimates of these quantities. Even after 118 Myrs of model evolution, the mantle density structure of our model remains in good agreement with the preliminary reference Earth model (PREM Dziewonski and Anderson (1981), see fig. 2(b) & (e)). The geotherm of the conduction dominated regime remains well in the range of pressure-temperature ($P$-$T$) estimates from mantle xenolith data. Also, the geotherm of the convection dominated regime remains within the range for adiabatic gradients and potential temperatures applicable to the Earth's mantle (Hasterok and Chapman (2011), see fig. 2(a) & (d)) during the entire long-term cycle. Viscosity profiles lie within the range of estimates inferred by inversion of observable geophysical data and from experimentally determined flow law parameters of olivine rheology (Mitrovica and Forte (2004); Behn et al. (2004); Hirth and Kohlstedt (2003), see fig. 2(c) & (f)). Lithosphere and mantle velocities are in a range of several $\mathrm{cm\ yr^{-1}}$ which is in agreement with predictions from boundary layer theory of layered mantle convection (Schubert et al., 2001) and plate velocity estimates from GPS measurements (Reilinger et al., 2006).

## 4.7    Formation and reactivation of magma-poor rifted margins: potential applications

Our model can be applied to some first-order geodynamic processes that were likely important for the orogeny of the Alps. Rifting in the Early to Middle Jurassic (Favre and Stampfli, 1992; Froitzheim and Manatschal, 1996; Handy et al., 2010) lead to the formation of the Piemont-Liguria ocean which was bounded by the hyper-extended magma-poor rifted margins of the Adriatic plate and the Briançonnais domain on the side of the European plate. We follow here the interpretation that the Piemont-Liguria ocean was an embryonic ocean which formed during ultra-slow spreading and was dominated by exhumed subcontinental mantle (e.g. Picazo et al., 2016; McCarthy et al., 2018; Chenin et al., 2019; McCarthy et al., 2020). If true, there was no stable mid-ocean ridge producing a several 100 km wide ocean with a typically 8 km thick oceanic crust and our model would be applicable to the formation of an embryonic ocean with exhumed mantle bounded by magma-poor hyper-extended rifted margins. Our models show the formation of a basin with exhumed mantle bounded by hyper-extended margins above a convecting mantle. Hence, our model may describe the first-order thermo-mechanical processes during formation of an embryonic ocean. During closure of the Piemont-Liguria ocean, remnants of those magma-poor ocean-continent transitions escaped subduction and are preserved in the Eastern Alps (Manatschal and Müntener, 2009). We follow the interpretation that the initiation and at least the early stages of ocean closure were caused by far-field convergence between the African and

510 European plates (e.g. Handy et al., 2010). We further assume that subduction was induced, or horizontally forced, by this convergence and was not initiated spontaneously due to buoyancy of a cold oceanic lithosphere (De Graciansky et al., 2010). This interpretation agrees with recent results of Crameri et al. (2020) who suggest that most subduction zones that formed during the last 100 Myrs were likely induced by horizontal shortening. Our models show that a cooling exhumed mantle does not subduct spontaneously because buoyancy forces are not significant enough to overcome the strength of the lithosphere.

However, the models show that convergence of the basin generates a horizontally forced subduction initiation at the hyper-extended passive margin causing subduction of the exhumed mantle below the passive margin. Such subduction initiation at the passive margin agrees with geological reconstructions which suggest that the Alpine subduction initiated at the hyper-extended margin of Adria (e.g. Manzotti et al., 2014) and not, for example, within the ocean. Overall, the here modelled, more than 100 Myrs long, geodynamic cycle is thus in agreement with several geological reconstructions of the Alpine orogeny. During

convergence, several of our models show the formation of a divergent double-slab subduction. Such divergent double-slab subduction likely applies to the eastward and westward dipping subduction of the Adriatic plate (Faccenna and Becker, 2010; Hua et al., 2017). Subduction started significantly earlier below the Dinarides compared to the westward directed subduction (Handy et al., 2010). Since about 30 Ma, the Adriatic plate undergoes a divergent double slab subduction, for which the two subduction zones have started at different times. In model M4, subduction initiation does not occur simultaneously below both

margins. The subduction initiation below the left margin occurs ca. 10 Myrs after subduction initiation below the right margin. During the evolution of the model, the subduction switches from the right margin to the left margin and then back again (see also video supplement). Therefore, divergent double-slab subduction does not require that subduction initiation at the passive margins occurs at the same time. However, the Alpine orogeny exhibits a distinct three-dimensional evolution including major stages of strike-slip deformation and a considerably radial shortening direction so that any 2D model can always only address

the fundamental aspects of the involved geodynamic processes.

Another example of divergent double-slab subduction is presumably the Paleo-Asian Ocean, which has been subducted beneath both the southern Siberian Craton in the north and the northern margin of the North China Craton in the south during the Paleozoic (Yang et al., 2017). Furthermore, a divergent double-slab subduction was also suggested between the North Qiangtang and South Qiangtang terrane (Li et al., 2020; Zhao et al., 2015). Our models show that a divergent double-slab

subduction is a thermo-mechanically feasible process during convergence of tectonic plates.

Tomographic images from the Mediterranean show large p-wave anomalies in the transition zone (Piromallo and Morelli, 2003) indicating that the 660-km phase transition inhibits the sinking material to penetrate further into the lower mantle. This observation suggests that convection in the Alpine-Mediterranean region could be two-layered and largely confined to the upper region of the upper mantle. The convective patterns resulting from the presence of a weak asthenosphere simulated in

our study are in agreement with these observations. We speculate that upper mantle convection might have played a role in the formation of the Alpine orogeny in the form of inducing an additional suction force below the Adriatic margin and assisting in the onset of subduction.

## 5 Conclusions

Our 2D thermo-mechanical numerical models of coupled lithosphere-mantle deformation are able to generate a 120 Myrs long geodynamic cycle of subsequent extension (30 Myrs), cooling (70 Myrs) and convergence (20 Myrs) in a continuous simulation. The simulations capture the fundamental features of such cycles, such as formation of hyper-extended margins, upper mantle convective flow or subduction initiation, with model outputs that are applicable to Earth. We propose that the ability of a model to generate such long-term cycles in a continuous simulation with constant parameters provides further confidence that the model has captured correctly the first-order physics.

Our models show that the viscosity structure of the asthenosphere and the associated vigour of upper mantle convection has a significant impact on lithosphere dynamics during a long-term geodynamic cycle. In comparison to a strong asthenosphere with minimum viscosities of $5 \times 10^{20}$ Pa s, a weak asthenosphere with minimum viscosities of ca. $10^{19}$ Pa s generates the following differences: (1) locally larger suction forces due to convective flow, which are able to assist in establishing a single-slab subduction instead of a divergent double-slab subduction, and (2) smaller horizontal driving forces to initiate horizontally forced subduction, namely ca. 15 TN m$^{-1}$ instead of ca. 22 TN m$^{-1}$. Therefore, quantifying the viscosity structure of the asthenosphere is important for understanding the actual geodynamic processes acting in specific regions.

In our models, subduction at a hyper-extended passive margin is initiated during horizontal shortening and by shear localization due to mainly thermal softening. In contrast, after 70 Myrs of cooling without far-field deformation, subduction of a 400 km wide exhumed and cold mantle is not spontaneously initiated. The buoyancy force due to the density difference between lithosphere and asthenosphere is too small to overcome the mechanical strength of the lithosphere.

The first-order geodynamic processes simulated in the geodynamic cycle of subsequent extension, cooling and convergence are applicable to orogenies that resulted from the opening and closure of embryonic oceans, which might have been the case for the Alpine orogeny.

*Data availability.* The data presented in this study are available on request from Lorenzo G. Candioti.

*Video supplement.* The videos (https://doi.org/10.5446/48939) and (https://doi.org/10.5446/48940) show the entire model evolution of M1 (reference run) and the convergence stage of M4, respectively, as discussed in the article.

## Appendix A: Numerical algorithm

**Table A1.** Physical parameters used in the numerical simulations M1-6.

| Parameter | Unit | Strong Crust[1] | Weak Crust[2] | Calcite[3] | Mica[4] | Dry Mantle[5] | Wet Mantle[6] |
|---|---|---|---|---|---|---|---|
| $\rho_0$ | kg m$^{-3}$ | 2800 | 2800 | 2800 | 2800 | - | - |
| $G$ | Pa | $2\times10^{10}$ | $2\times10^{10}$ | $2\times10^{10}$ | $2\times10^{10}$ | $2\times10^{10}$ | $2\times10^{10}$ |
| $c_P$ | J kg$^{-1}$ K$^{-1}$ | 1050 | 1050 | 1050 | 1050 | 1050 | 1050 |
| $k$ | W m$^{-1}$ K$^{-1}$ | 2.25 | 2.25 | 2.37 | 2.55 | 2.75 | 2.75 |
| $H_R$ | W m$^{-3}$ | $0.9\times10^{-6}$ | $0.9\times10^{-6}$ | $0.56\times10^{-6}$ | $2.9\times10^{-6}$ | $2.1139\times10^{-8}$ | $2.1139\times10^{-8}$ |
| $C$ | Pa | $10^7$ | $10^6$ | $10^7$ | $10^6$ | $10^7$ | $10^7$ |
| $\varphi$ | ° | 30 | 5 | 30 | 5 | 30 | 30 |
| $\alpha$ | K$^{-1}$ | $3\times10^{-5}$ | $3\times10^{-5}$ | $3\times10^{-5}$ | $3\times10^{-5}$ | $3\times10^{-5}$ | $3\times10^{-5}$ |
| $\beta$ | Pa$^{-1}$ | $1\times10^{-11}$ | $1\times10^{-11}$ | $1\times10^{-11}$ | $1\times10^{-11}$ | $1\times10^{-11}$ | $1\times10^{-11}$ |
| **Dislocation** | | | | | | | |
| $A$ | Pa$^{-n-r}$ s$^{-1}$ | $5.0477\times10^{-28}$ | $5.0717\times10^{-18}$ | $1.5849\times10^{-25}$ | $10^{-138}$ | $1.1\times10^{-16}$ | $5.6786\times10^{-27}$ |
| $n$ | - | 4.7 | 2.3 | 4.7 | 18 | 3.5 | 3.5 |
| $Q$ | J mol$^{-1}$ | $485\times10^3$ | $154\times10^3$ | $297\times10^3$ | $51\times10^3$ | $530\times10^3$ | $460\times10^3$ |
| $V$ | m$^3$ mol$^{-1}$ | 0 | 0 | 0 | 0 | $14\times10^{-6}$ | $11\times10^{-6}$ |
| $r$ | - | 0 | 0 | 0 | 0 | 0 | 1.2 |
| $f_{H_2O}$ | Pa | 0 | 0 | 0 | 0 | 0 | $10^9$ |
| **Diffusion** | | | | | | | |
| $A^*$ | Pa$^{-n-r}$ m$^m$ s$^{-1}$ | - | - | - | - | $1.5\times10^{-15}$ | $2.5\times10^{-23}$ |
| $n$ | - | - | - | - | - | 1 | 1 |
| $Q$ | J mol$^{-1}$ | - | - | - | - | $370\times10^3$ | $375\times10^3$ |
| $V$ | m$^3$ mol$^{-1}$ | - | - | - | - | $7.5\times10^{-6}$ | $20\times10^{-6}$ |
| $m$ | - | - | - | - | - | 3 | 3 |
| $r$ | - | - | - | - | - | 0 | 1 |
| $f_{H_2O}$ | Pa | - | - | - | - | 0 | $10^9$ |
| $d$ | m | - | - | - | - | $10^{-3}$ | $10^{-3}$ |
| **Peierls** | | | | | | | |
| $A$ | s$^{-1}$ | - | - | - | - | $5.7\times10^{11}$ | $5.7\times10^{11}$ |
| $Q$ | J mol$^{-1}$ | - | - | - | - | $540\times10^3$ | $540\times10^3$ |
| $\sigma_P$ | Pa | - | - | - | - | $8.5\times10^9$ | $8.5\times10^9$ |
| $\gamma$ | - | - | - | - | - | 0.1 | 0.1 |

Flow law parameters: [1]Maryland Diabase (Mackwell et al., 1998), [2]Wet Quartzite (Ranalli, 1995), [3]Calcite (Schmid et al., 1977), [4]Mica (Kronenberg et al., 1990), [5]Dry Olivine (Hirth and Kohlstedt, 2003) and [6]Wet Olivine (Hirth and Kohlstedt, 2003). Peierls creep: (Goetze and Evans, 1979) regularised by Kameyama et al. (1999). [*]Converted to SI units from original units: $A = 2.5\times10^7$ ([MPa])$^{-n-r}$ ([μm])$^m$ ([s])$^{-1} = 2.5\times10^7 \times (10^{-6n-6r}$ [Pa]$^{-6n-6r}) \times (10^{-6m}$ [m]$^m) \times$ ([s]$^{-1}) = 2.5\times10^{-23}$ [Pa$^{-2}$m$^3$s$^{-1}$].

Continuity and force balance equations for an incompressible slowly flowing (no inertial forces) fluid under gravity are given by

$$\frac{\partial v_i}{\partial x_i} = 0 \tag{A1}$$

$$\frac{\partial \sigma_{ij}}{\partial x_j} = -\rho \, a_i \, , \tag{A2}$$

where $v_i$ denotes velocity vector components and $x_i$ spatial coordinate components, where $(i,j=1)$ indicates the horizontal direction and $(i,j=2)$ the vertical direction, $\sigma_{ij}$ are components of the total stress tensor, $\rho$ is density and $a_i = [0; g]$ is a vector with $g$ being the gravitational acceleration. Density is a function of pressure $P$ (negative mean stress) and temperature $T$ computed as a simplified equation of state for the crustal phases like

$$\rho(P,T) = \rho_0 \, (1 - \alpha \Delta T) \, (1 + \beta \Delta P) \, , \tag{A3}$$

where $\rho_0$ is the material density at the reference temperature $T_0$ and pressure $P_0$, $\alpha$ is the thermal expansion coefficient, $\beta$ is the compressibility coefficient, $\Delta T = T - T_0$ and $\Delta P = P - P_0$. Effective density for the mantle phases is pre-computed using the software package Perple_X (Connolly, 2005) for the bulk rock composition of a Hawaiian pyrolite (Workman and Hart, 2005). Figure A1 shows the density distribution for the calculated pressure and temperature range.

The visco-elastic stress tensor components are defined using a backward-Euler scheme (e.g., Schmalholz et al., 2001) as

$$\sigma_{ij} = -P\delta_{ij} + 2 \, \eta^{\text{eff}} \, \dot{\varepsilon}_{ij}^{\text{eff}} + J_{ij} \, , \tag{A4}$$

where $\delta_{ij} = 0$ if $i \neq j$, or $\delta_{ij} = 1$ if $i = j$, $\eta^{\text{eff}}$ is the effective viscosity, $\dot{\varepsilon}_{ij}^{\text{eff}}$ are the components of the effective deviatoric strain rate tensor,

$$\dot{\varepsilon}_{ij}^{\text{eff}} = \left( \dot{\varepsilon}_{ij} + \frac{\tau_{ij}^o}{2G\Delta t} \right) \, , \tag{A5}$$

where $G$ is shear modulus, $\Delta t$ is the time step, $\tau_{ij}^o$ are the deviatoric stress tensor components of the preceding time step and $J_{ij}$ comprises components of the Jaumann stress rate as described in detail in Beuchert and Podladchikov (2010). A visco-elasto-plastic Maxwell model is used to describe the rheology, implying that the components of the deviatoric strain rate tensor $\dot{\varepsilon}_{ij}$ are additively decomposed into contributions from the viscous (dislocation, diffusion and Peierls creep), elastic and brittle plastic deformation as

$$\dot{\varepsilon}_{ij} = \dot{\varepsilon}_{ij}^{\text{ela}} + \dot{\varepsilon}_{ij}^{\text{dis}} + \dot{\varepsilon}_{ij}^{\text{dif}} + \dot{\varepsilon}_{ij}^{\text{pei}} + \dot{\varepsilon}_{ij}^{\text{pla}} \, , \tag{A6}$$

In case deformation is effectively visco-elastic, a local iteration cycle is performed on each cell/node until eq. A6 is satisfied (e.g., Popov and Sobolev, 2008). The viscosity for the dislocation and Peierls creep flow law is a function of the second invariant of the respective strain rate components $\dot{\varepsilon}_{\mathrm{II}}^{\mathrm{dis,pei}} = \tau_{\mathrm{II}}/(2\eta^{\mathrm{dis,pei}})$

$$\eta^{\mathrm{dis}} = \frac{2^{\frac{1-n}{n}}}{3^{\frac{1+n}{2n}}} A^{-\frac{1}{n}} \left(\dot{\varepsilon}_{\mathrm{II}}^{\mathrm{dis}}\right)^{\frac{1}{n}-1} \exp\left(\frac{Q+PV}{nRT}\right) \left(f_{\mathrm{H_2O}}\right)^{-\frac{r}{n}} , \tag{A7}$$

where the ratio in front of the pre-factor $A$ results from conversion of the experimentally derived 1D flow law, obtained from laboratory experiments, to a flow law for tensor components (e.g., Schmalholz and Fletcher, 2011). For the mantle material diffusion creep is taken into account and its viscosity takes the following form

$$\eta^{\mathrm{dif}} = \frac{1}{3} A^{-1} d^m \exp\left(\frac{Q+PV}{RT}\right) \left(f_{\mathrm{H_2O}}\right)^{-r}, \tag{A8}$$

where $d$ is grain size and $m$ is a grain size exponent. Effective Peierls viscosity is calculated using the regularised form of Kameyama et al. (1999) for the experimentally derived flow law by Goetze and Evans (1979) as

$$\eta^{\mathrm{pei}} = \frac{2^{\frac{1-s}{s}}}{3^{\frac{1+s}{2s}}} \hat{A} \left(\dot{\varepsilon}_{\mathrm{II}}^{\mathrm{pei}}\right)^{\frac{1}{s}-1} , \tag{A9}$$

where $s$ is an effective, temperature dependent stress exponent:

$$s = 2\gamma \frac{Q}{RT}(1-\gamma) . \tag{A10}$$

$\hat{A}$ in Eq. (A9) is

$$\hat{A} = \left[A \exp\left(-\frac{Q(1-\gamma)^2}{RT}\right)\right]^{-\frac{1}{s}} \gamma\sigma_{\mathrm{P}} , \tag{A11}$$

where $A_{\mathrm{P}}$ is a pre-factor, $\gamma$ is a fitting parameter and $\sigma_{\mathrm{P}}$ is a characteristic stress value. The parameters $A$, $Q$, $V$, $m$ and $r$ are defined independently for each deformation mechanism (e.g., $A^{\mathrm{dis}}$, $A^{\mathrm{dif}}$, $A^{\mathrm{pei}}$). However, for practical reasons we omit the corresponding superscripts (dis, dif, pei) for these material parameters.

In the frictional domain, stresses are limited by the Drucker-Prager yield function

$$F = \tau_{\mathrm{II}} - P \sin\phi - C \cos\phi , \tag{A12}$$

where $\phi$ is the internal angle of friction and $C$ is the cohesion. If the yield condition is met ($F \geq 0$), the equivalent plastic viscosity is computed as

$$\eta^{\mathrm{pla}} = \frac{P \sin\phi + C \cos\phi}{2\dot{\varepsilon}_{\mathrm{II}}^{\mathrm{eff}}} \tag{A13}$$

and the effective deviatoric strain rate is equal to the plastic contribution of the deviatoric strain rate (eq. A5). At the end of the iteration cycle, the effective viscosity in eq. A4 is either computed as the inverse of the quasi-harmonic average of the visco-elastic contributions

$$
\eta^{\mathrm{eff}} = \begin{cases} \left( \frac{1}{G \Delta t} + \frac{1}{\eta^{\mathrm{dis}}} + \frac{1}{\eta^{\mathrm{dif}}} + \frac{1}{\eta^{\mathrm{pei}}} \right)^{-1} & , F < 0 \\ \eta^{\mathrm{pla}} & , F \geq 0 \end{cases} \tag{A14}
$$

or is equal to the viscosity $\eta^{\mathrm{pla}}$ calculated at the yield stress according to eq. A13. Thermal evolution of the model is calculated with the heat transfer equation

$$
\rho \, c_{\mathrm{P}} \frac{\mathrm{D}T}{\mathrm{D}t} = \frac{\partial}{\partial x_i} \left( k \, \frac{\partial T}{\partial x_i} \right) + H_{\mathrm{A}} + H_{\mathrm{D}} + H_{\mathrm{R}} \, , \tag{A15}
$$

where $c_{\mathrm{P}}$ is the specific heat capacity at constant pressure, $\mathrm{D}/\mathrm{D}t$ is the material time derivative, $k$ is thermal conductivity, $H_{\mathrm{A}} = T \alpha v_z g \rho$ is a heat source or sink resulting from adiabatic processes assuming lithostatic pressure conditions, $H_{\mathrm{D}} = \tau_{ij} \left( \dot{\varepsilon}_{ij} - \dot{\varepsilon}_{ij}^{\mathrm{ela}} \right)$ results from the conversion of dissipative work into heat (so-called shear heating) and $H_{\mathrm{R}}$ is a radiogenic heat source. To initiate the deformation, we perturbed the initial marker field with a random amplitude vertical displacement like

$$
z_{\mathrm{M}} = z_{\mathrm{M}} + A \, \exp \left( -\frac{x_{\mathrm{M}}}{\lambda} \right) , \tag{A16}
$$

where $z_{\mathrm{M}}$ is the vertical marker coordinate in $\mathrm{km}$, $A$ is a random amplitude varying between -1.25 $\mathrm{km}$ and 1.25 $\mathrm{km}$, $x_{\mathrm{M}}$ is the horizontal marker coordinate in $\mathrm{km}$ and $\lambda = 25 \, \mathrm{km}$ is the half-width of the curve. The perturbation is applied to the horizontal centre of the domain between -75 $\mathrm{km}$ and 75 $\mathrm{km}$.

All physical parameters are summarised in table A1.

## Appendix B:  Nusselt number scaling laws and phase transitions

Modelling thermal convection beneath an actively deforming lithosphere can be numerically expensive, because the convection velocities can be as high as or even higher than the motion of the lithospheric plates, depending on the vigour of the convecting system. This significantly reduces the maximum time step necessary to ensure numerical stability. In consequence, it takes more time steps to run a simulation to the same physical time when convection is modelled together with deformation in the lithosphere. Hence, the computational time can be twice as long compared to models, where only the deforming lithosphere is modelled. However, it is possible to include the effect of convection in the mantle on the thermal field and keep a constant vertical heat flux through the lithosphere-asthenosphere boundary (LAB) into a numerical model without explicitly modelling convection by using an effective thermal conductivity for the mantle material below the lithosphere. Two dimensionless quan-

640 tities have to be defined, namely the Rayleigh and Nusselt numbers. The Rayleigh number is the ratio of the thermal diffusion and advection time scale

$$Ra = \frac{t_{\text{Dif}}}{t_{\text{Adv}}} = \frac{\rho g \alpha \Delta T D^3}{\kappa \eta_{\text{eff}}} \ , \tag{B1}$$

where $\rho$ is density, $g$ is gravitational acceleration, $\alpha$ is a coefficient of thermal expansion, $\Delta T$ is the temperature difference between the top and the bottom and $D$ is the thickness of the convecting layer, $\kappa = k/\rho/c_P$ is the thermal diffusivity and $\eta_{\text{eff}}$ is
645 the effective viscosity. The Nusselt number can be expressed in terms of the Rayleigh number as

$$Nu = \left( \frac{Ra}{Ra_{\text{crit}}} \right)^{\beta} \ , \tag{B2}$$

where $Ra_{\text{crit}}$ is the critical Rayleigh number at which convection starts, typically in the order of $10^3$, and $\beta$ is a power-law exponent (Schubert et al., 2001). The Nusselt number is the ratio of advective heat flux, $q_{\text{Adv}}$, which is the vertical heat flux through the base of the lithosphere, imposed by the convecting upper mantle to the diffusive heat flux, $q_{\text{Dif}}$, imposed by the
650 lithosphere on top of the convecting upper mantle as

$$Nu = \frac{q_{\text{Adv}}}{q_{\text{Dif}}} \ . \tag{B3}$$

Using this relationship, it is possible to scale the thermal conductivity to the Nusselt number of the Earth's mantle and to maintain a constant heat flow through the base of the lithosphere via conduction when convection is absent. Assuming $Ra = 2 \times 10^6$ and $\beta = 1/3$ for the Earth's upper mantle convection, eq. B2 predicts $Nu = 13$. This implies that the heat flow
provided by advection is $13\times$ higher than the heat flow provided by conduction. Using an effective conductivity approach, the heat flow provided by advection is mimicked using an enhanced conductive heat flow in the upper mantle. The effective conductivity can be determined by scaling the standard value of thermal conductivity of the upper mantle material to the Nusselt number of the convecting system like

$$k_{\text{eff}} = Nu \ k. \tag{B4}$$

For this study, the standard value for $k = 2.75$ of the upper mantle material and $Nu = 13$. The effective conductivity according to eq. B4 is $k_{\text{eff}} = 36$. To avoid a strong contrast of conductivities directly at the base of the lithosphere, we linearly increase the conductivity from 2.75-36 W m$^{-1}$ K$^{-1}$ over a temperature range of 1350-1376 °C. Applying this effective conductivity approach reduces the number of time steps necessary for computation up to the same physical time by ca. a factor 2 in M4 compared to M1.
As mentioned above, the vigour of convection is defined by the Rayleigh number (eq. B1). For $Ra \gg Ra_{\text{crit}}$, the time scale for thermal diffusion is much larger than the time scale for advection of material. This means that the entropy of the system remains

relatively constant in time. By definition, such a system is adiabatic (Kondepudi and Prigogine, 2014). In the presented models, the density and entropy for the mantle phases is pre-computed using Perple_X for a given bulk rock composition. Assuming that the temperature gradient in the upper mantle is adiabatic and stress conditions are close to lithostatic (i.e., deviatoric stresses are negligible), the temperature at any depth can be determined by following an isentrop from the Perple_X database. Starting coordinates in pressure-temperature space are the (lithostatic) conditions at the base of the lithosphere. From these pressure and temperature values one can follow the closest isentrop (black line Fig. A1) until the (lithostatic) pressure value at target depth (in this study 660 km, red diamond Fig. A1) is reached and extract the corresponding temperature value. Trubitsyn and Trubitsyna (2015) derived an analytical solution to calculate temperatures assuming an adiabatic gradient for given depths. We determined the temperature at the bottom of the model domain using both approaches and the obtained values that differ by only 0.01 °C.

Involving phase transitions necessitates mainly two major assumptions: (1) is compressibility of material due to large density variations important and (2) does latent heat released or consumed at a phase transition significantly change the convective pattern? Bercovici et al. (1992) concluded that compressibility effects on the spatial structure of mantle convection are minor when the superadiabatic temperature drop is close to the adiabatic temperature of the mantle, which is the case for the Earth. Although the net density varies largely in *P-T* space of the phase diagram used in this study (fig. A1), the maximum value for the density time derivative computed from M1 is two orders of magnitude lower than the velocity divergence. We assume that density changes due to volumetric deformation are, hence, still negligible and density changes are accounted for in the buoyancy force only. This means the classical Boussinesq approximation is still valid. However, not considering adiabatic heating in the energy conservation equation leads to a significant deviation of the thermal structure from the initially imposed temperature gradient over large time scales (>100 Myrs). The resulting temperature profile is constant throughout the upper mantle and the newly equilibrated constant temperature is equal to the imposed temperature at the bottom boundary. In consequence, the density structure read in from the phase diagram table according to pressure and temperature values is significantly wrong. To avoid these problems, we use the extended Boussinesq approximation, i.e., the adiabatic heating term is included in the energy conservation equation. As a result, the initially imposed adiabatic (or isentropic) temperature gradient can be maintained over large time scales. The resulting density structure agrees well with the PREM model (Dziewonski and Anderson, 1981) as shown in this study. A detailed comparison between different approximations of the system of equations is clearly beyond the scope of this study.

Latent heat that is released or consumed by a phase transition can perturb the thermal field by up to 100 K and induce a buoyancy force aiding or inhibiting the motion of cold, especially low-angle, subducting slabs (van Hunen et al., 2001) or hot rising plumes. However, when the lateral differences in temperature are small, the deflection of the phase transition by an ascending plume or a subducting slab has a much bigger impact on the buoyancy stresses than the latent heat released or consumed by the phase transition (Christensen, 1995). Because a detailed parametric investigation of the impact of latent heat on buoyancy stresses is beyond the scope of study, we neglect latent heat for simplicity.

## Appendix C: Convection benchmark

In this section, we present the results of a convection benchmark performed by the algorithm used in this study. Equations for continuity and force balance are solved as in eq. A1 and A2, density is a function of temperature only and calculated as

$$\rho(T) = \rho_0(1 - \alpha T) \tag{C1}$$

and the total stress tensor is decomposed into a pressure and a deviatoric part as

$$\sigma_{ij} = -P\delta_{ij} + \tau_{ij} \ . \tag{C2}$$

Transfer of heat is calculated as in eq. A15. Stresses and strain rates ($\dot{\varepsilon}_{ij}$) are related to each other via the viscosity $\eta$ as

$$\tau_{ij} = 2\eta\dot{\varepsilon}_{ij} \ . \tag{C3}$$

Viscosity is computed via a linearized Arrhenius law, also called Frank-Kamenetskii approximation (Kamenetskii, 1969):

$$\eta(T,z) = \exp(-\gamma_T + \gamma_z) \ , \tag{C4}$$

with $\gamma_T = \log(\eta_T)$ and $\gamma_z = \log(\eta_z)$. By choosing $\eta_z = 1$, $\gamma_z = 0$ in eq. C4 and, therefore, the viscosity is only temperature dependent.

The dimensionless equations are discretized over a domain that extends from 0 to 1 in both horizontal and vertical directions and a small amplitude perturbation ($A = 0.01$) is applied to the initial temperature profile as

$$T(x,z) = (1-z) + A\cos(\pi x)\sin(\pi z) \ . \tag{C5}$$

As mentioned above, the vigour of the convecting system is described by its Rayleigh number (eq. B1). A local $Ra = 10^2$ is applied to the top boundary by setting $\alpha = 10^{-2}$, $g = 10^4$ and all other parameters of eq. B1 are set to 1. The applied viscosity decrease by choosing $\eta_T = 10^5$ in eq. C4 results in a global $Ra = 10^7$. All mechanical boundaries are set to free slip, the thermal boundary conditions are constant temperature at the top ($T = 0$) and bottom ($T = 1$) and insulating (i.e., zero flux) at the two vertical boundaries. Tosi et al. (2015) tested several algorithms, including finite element, finite differences, finite volume and spectral discretization, on their capability of modelling distinct rheologies of the mantle, from temperature dependent viscosity only up to visco-plastic rheologies. We have chosen the simplest test, case one in Tosi et al. (2015), and report the results of two distinct diagnostic quantities: the average temperature over the entire modelling domain

$$\langle T \rangle = \int\limits_0^1 \int\limits_0^1 T \, \mathrm{d}x\mathrm{d}z \tag{C6}$$

and the root mean square velocity at the surface

$$725 \quad u_{\text{RMS}}^{\text{surf}} = \left( \int\limits_0^1 v_x^2 \Big|_{z=1} \, dx \right)^{\frac{1}{2}} . \tag{C7}$$

The model develops one convection cell below a stagnant lid (fig. A2(a)). We tested numerical resolutions of $50^2$, $100^2$, $150^2$ and $300^2$. Only for resolutions $>100^2$ the desired convective pattern developed. The numerical algorithms tested by Tosi et al. (2015) passed the benchmark already for lower resolutions. This is due to the fact that the algorithm presented here uses a uniform grid size across the domain. The algorithms tested by Tosi et al. (2015) used refined meshes. Sufficient resolution of the thermal boundary layers at the top and at the bottom is crucial to develop the desired pattern. Using a refined mesh in these regions, allows for lower total resolution, whereas using a regular mesh necessitates a much higher resolution in total. Nevertheless, values for the diagnostic quantities reproduced by the presented algorithm lie well within the minimum and maximum values calculated by the algorithms tested in Tosi et al. (2015) (grey areas in fig. A2(b) & (d)). This shows that the convection in the upper mantle, where the viscosity is essentially temperature dependent, in the models presented in this study in which convection is not parameterized, is accurately modelled.

### Appendix D:  Gravitational potential energy and plate driving forces

We use the gravitational potential energy per unit surface (*GPE*) to quantify the impact of convection induced density variations in the upper mantle during the distinct stages of the simulations. The *GPE* varies along the horizontal *x*-direction and is computed as

$$740 \quad GPE(x) = \int\limits_{Sb}^{St(x)} P_{\text{L}}(x, z) \, dz, \tag{D1}$$

where $P_{\text{L}}$ is the lithostatic pressure calculated as

$$P_{\text{L}}(x, z) = \int\limits_z^{St(x)} \rho(x, z') \, g \, dz' \tag{D2}$$

and $St(x)$ is the stress-free surface and $Sb$ is the model bottom. Horizontal variations in *GPE*, $\Delta GPE$, are calculated by subtracting the leftmost value as a reference value from all other values. The *GPE* gives an estimate on the plate driving forces per unit length (Molnar and Lyon-Caen, 1988; Schmalholz et al., 2019; Bessat et al., 2020) acting in the system during the

different stages of the hyper-extension and convergence cycle. We also calculate the vertical integral of the second invariant of the deviatoric stress tensor

$$
\bar{\tau}_{\mathrm{II}}(x) = \int_{Sb}^{St(x)} \tau_{\mathrm{II}}(x,z) \, \mathrm{d}z \ . \tag{D3}
$$

The value of $F_{\mathrm{D}} = 2 \times \bar{\tau}_{\mathrm{II}}^{\mathrm{avg}}(x)$, where $\bar{\tau}_{\mathrm{II}}^{\mathrm{avg}}(x)$ is calculated averaging the average of $\bar{\tau}_{\mathrm{II}}(x)$ both over the left and rightmost 100 km of the domain, is also identical to the vertical integral of the difference between the horizontal total stress and the lithostatic pressure, if shear stresses are negligible (Molnar and Lyon-Caen, 1988; Schmalholz et al., 2019). This condition is strictly satisfied by the choice of boundary conditions at the top and the bottom of the model domain. We therefore chose to show the profiles which have been integrated over the entire domain height ($z$=-660 km). Since the deviatoric stresses below the lithosphere are indeed negligibly small, values for $F_{\mathrm{D}}$ integrated from depths of $z = -660$km, $z = -330$km and $z = -120$km do not reveal significant differences. Therefore, the value of $F_{\mathrm{D}}$ is essentially independent on the integration depth (if deeper than the lithosphere thickness). $F_{\mathrm{D}}$ can therefore be used to estimate the plate driving force (see fig.7(d)). For calculation of the suction force per unit length ($F_{\mathrm{S}}$) induced by the mantle flow (fig. 6(a) & (f)) we used the following formula

$$
F_{\mathrm{S}} = \int_{b}^{a} \int_{d}^{c} \Delta\rho g \, \mathrm{d}z \, \mathrm{d}x \ , \tag{D4}
$$

where $a$, $b$, $c$ and $d$ are the integration bounds and $\Delta\rho$ is the difference in density between the entire density field at the end of the cooling period and a reference density value (profiles in fig. 2(e)).

*Author contributions.* Lorenzo G. Candioti configured and performed the numerical simulations and the convection benchmark, interpreted the numerical results, generated the figures and wrote the manuscript. Stefan M. Schmalholz designed the numerical study, helped in interpreting the results and designing the figures and contributed to writing the manuscript. Thibault Duretz developed the applied numerical algorithm and helped in configuring the model and the interpretation of the results.

*Competing interests.* The authors declare that they have no conflict of interest.

*Acknowledgements.* We gratefully thank Zhong-Hai Li and two anonymous reviewers for their constructive comments during the review process. This work is supported by SNF grant No. 200020 163169.

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

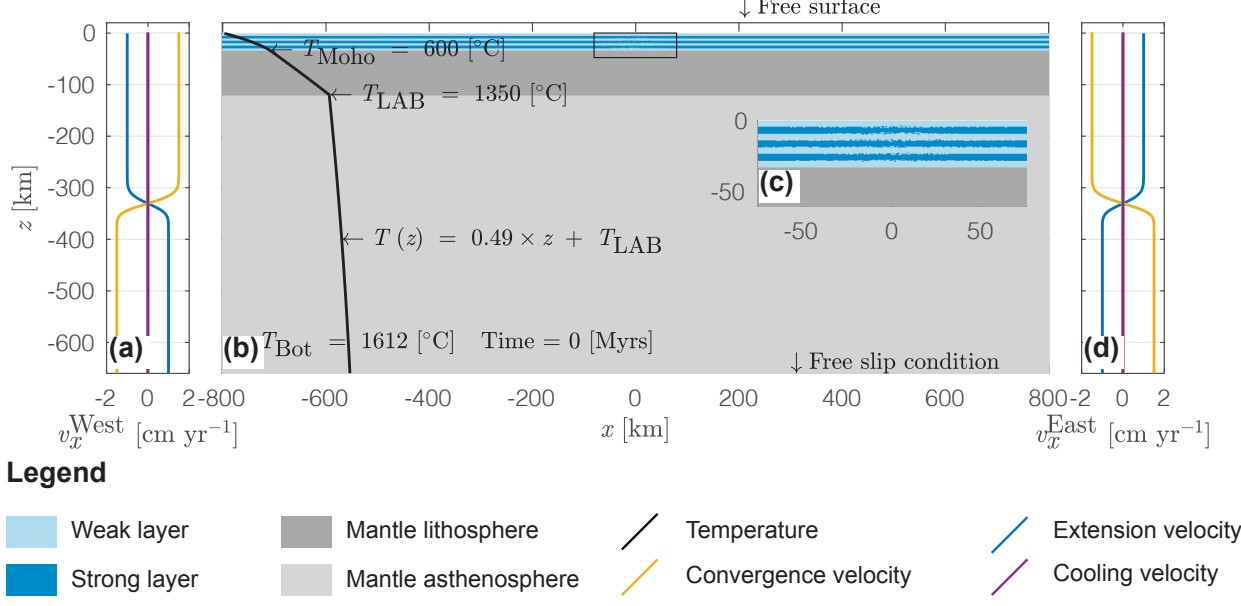

**Figure 1.** Initial model configuration and boundary conditions. Dark blue colours represent strong and light blue colours represent weak crustal units, dark grey colours represent the mantle lithosphere and light grey colours the upper mantle. (a) Profile of horizontal velocity for material inflow and outflow along the western model boundary. Blue line indicates the profile for the extension, purple line indicates the profile for the cooling and the yellow line indicates the profile for the convergence stage. (b) Entire model domain including the material phases (colour-code as explained above) and initial vertical temperature profile (black line), (c) enlargement of the centre of the model domain showing the initial random perturbation on the marker field (see appendix A) used to localise deformation in the centre of the domain and (d) is the same profile as shown in (a), but along the eastern model boundary.

Workman, R. K. and Hart, S. R.: Major and trace element composition of the depleted MORB mantle (DMM), Earth and Planetary Science Letters, 231, 53–72, 2005.

Yamato, P., Duretz, T., May, D. A., and Tartese, R.: Quantifying magma segregation in dykes, Tectonophysics, 660, 132–147, 2015.

Yamato, P., Duretz, T., and Angiboust, S.: Brittle/ductile deformation of eclogites: insights from numerical models, Geochemistry, Geophysics, Geosystems, 2019.

Yang, F., Santosh, M., Tsunogae, T., Tang, L., and Teng, X.: Multiple magmatism in an evolving suprasubduction zone mantle wedge: the case of the composite mafic–ultramafic complex of Gaositai, North China Craton, Lithos, 284, 525–544, 2017.

Zahnle, K. J., Kasting, J. F., and Pollack, J. B.: Evolution of a steam atmosphere during Earth's accretion, Icarus, 74, 62–97, 1988.

Zhao, Z., Bons, P., Wang, G., Soesoo, A., and Liu, Y.: Tectonic evolution and high-pressure rock exhumation in the Qiangtang terrane, central Tibet, Solid Earth, 6, 457, 2015.

Zhou, X., Li, Z.-H., Gerya, T. V., and Stern, R. J.: Lateral propagation–induced subduction initiation at passive continental margins controlled by preexisting lithospheric weakness, Science Advances, 6, eaaz1048, 2020.

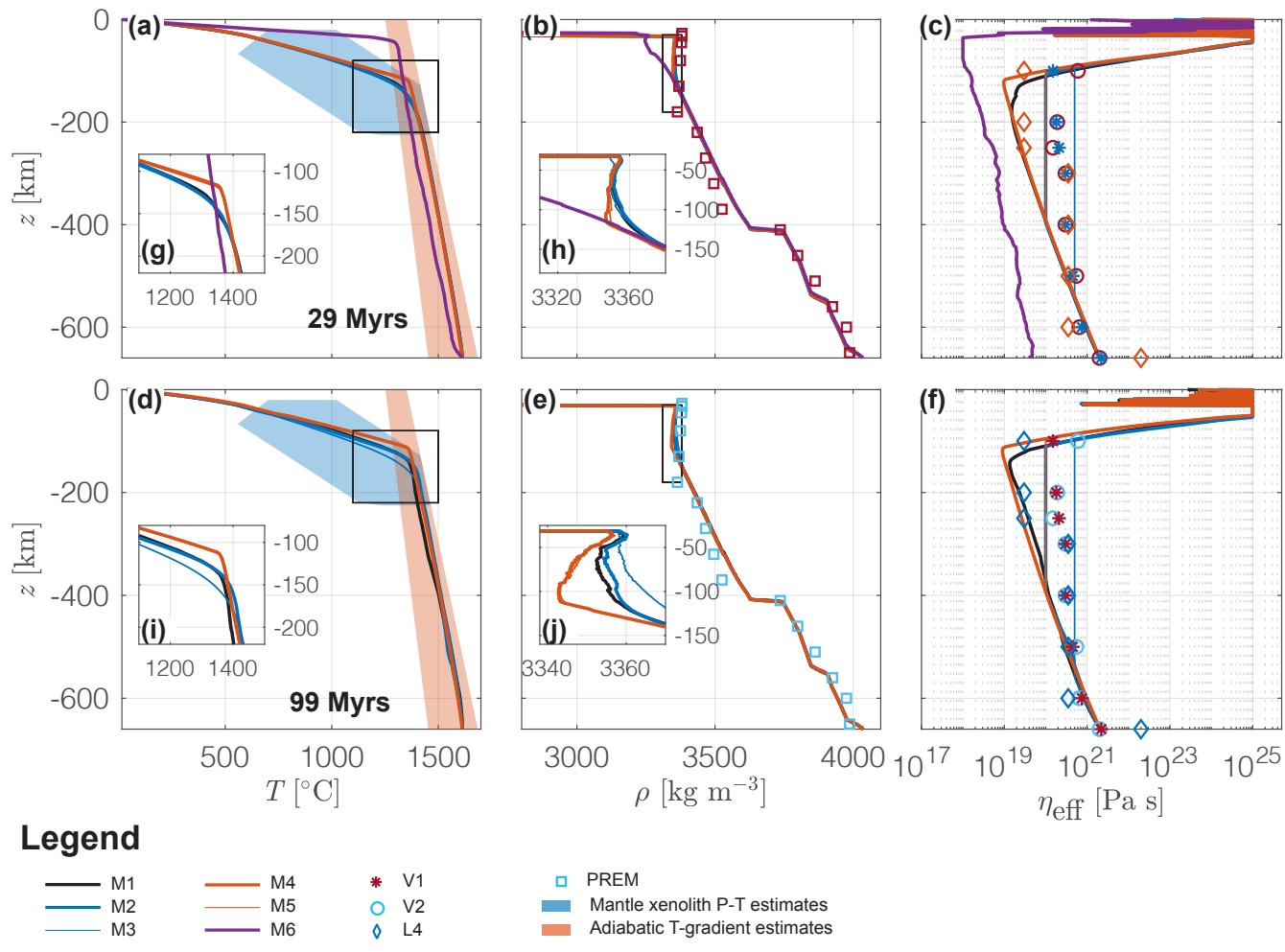

**Figure 2.** Horizontally averaged vertical profiles of temperature, $T$, density, $\rho$, and effective viscosity, $\eta_{\text{eff}}$. Top row: after 29 Myrs, bottom row: after 99 Myrs in model history. (a) and (d) show horizontally averaged temperature, (b) and (e) show horizontally averaged density and (c) and (f) show horizontally averaged viscosity. (g)-(j) show an enlargement of the parental subfigure. Coloured lines show the results of models M1-6 as indicated in the legend. In (a) and (d): blue area indicates *P-T* condition estimates from mantle xenolith data and orange area indicates estimates for a range of adiabatic gradients both taken from Hasterok and Chapman (2011) (fig. 5). Estimates for adiabatic temperature gradients are extrapolated to 660 km depth. In (b) and (e): squares indicate density estimates from the preliminary reference Earth model (PREM) (Dziewonski and Anderson, 1981). In (c) and (f): circles and stars indicate viscosity profiles inferred from Occam-style inversion of glacial isostatic adjustment and convection related data originally by Mitrovica and Forte (2004), diamonds show a four-layer model fit of mantle flow to seismic anisotropy originally by Behn et al. (2004). All profiles taken from Forte et al. (2010) (fig. 2). The median is used as statistical quantity for averaging, because it is less sensitive to extreme values compared to the arithmetic mean.

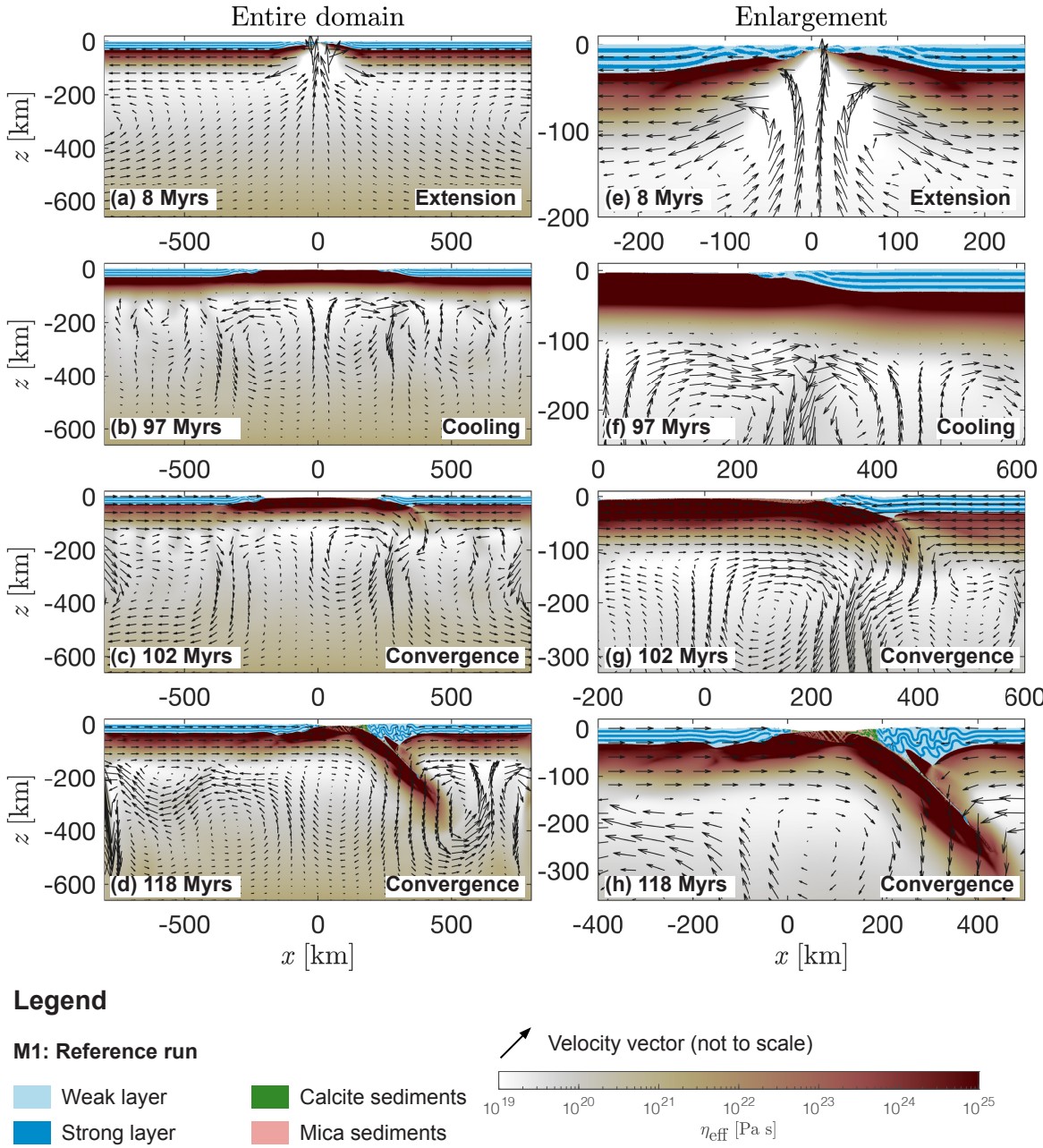

**Figure 3.** Model evolution of M1 (reference run). Left column: Entire domain. Right column: Enlargement. Dark and light blue colours, salmon and green colours indicate the material phase of weak and strong layers and mica and calcite sediments, respectively. White to red colours indicate the effective viscosity field calculated by the algorithm. Arrows represent velocity vectors and the length of the arrows is not to scale. Scientific colour maps used in all figures are provided by Crameri (2018).

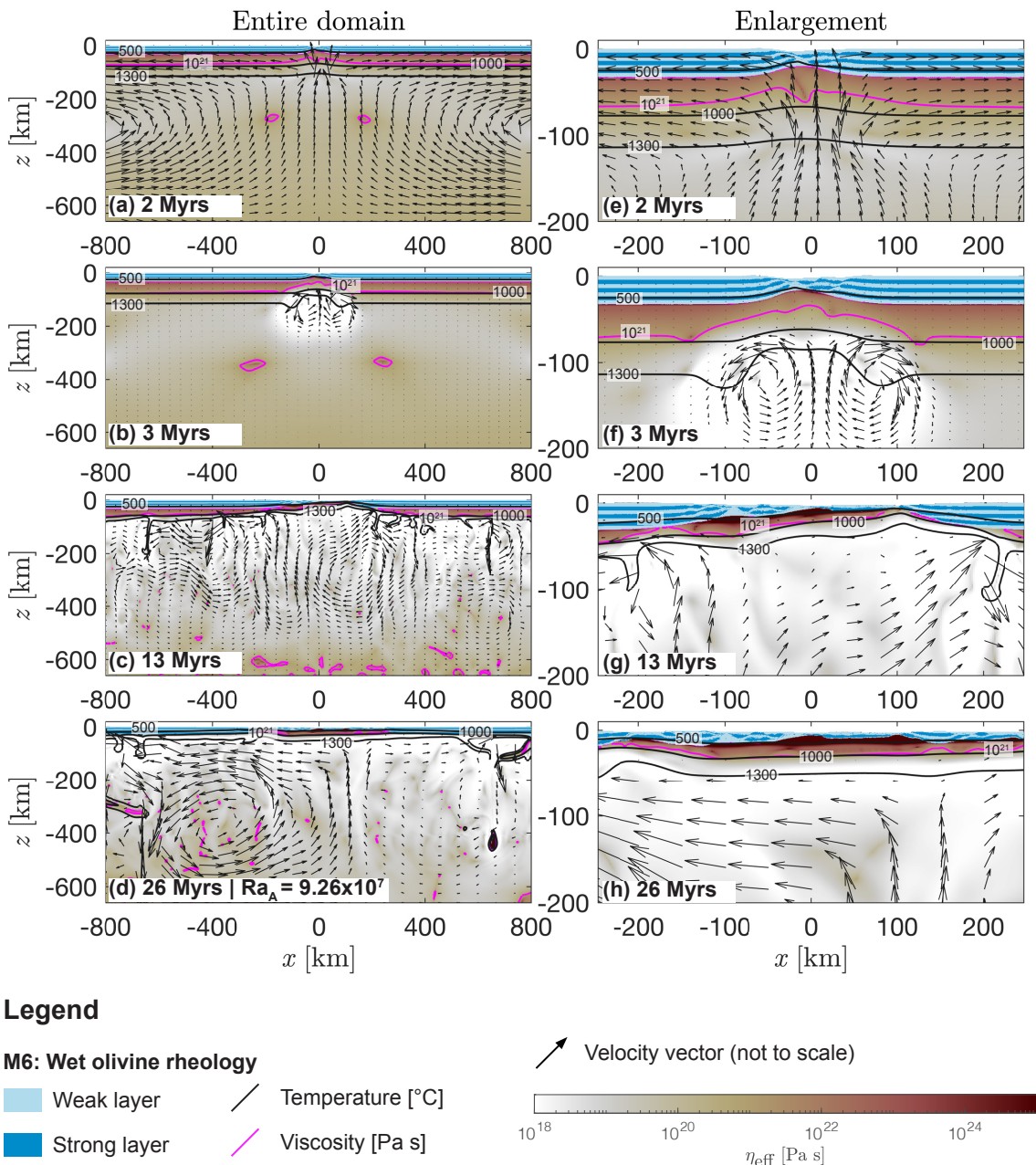

**Figure 4.** Evolution of model M6 at (a) & (e) 2 Myrs, (b) & (f) 3 Myrs, (c) & (g) 13 Myrs and (d) & (h) 26 Myrs. Blue colours indicate mechanically weak (light) and strong (dark) crustal units. White to red colours show the effective viscosity field calculated by the numerical algorithm. Black lines show the level of 500 °C, 1000 °C and 1300 °C isotherm and the magenta coloured line shows the level of the $10^{21}$ Pa s viscosity isopleth. This contour line represents the mechanical boundary between the mantle lithosphere and the convecting upper mantle. Arrows represent velocity vectors and the length of the arrows is not to scale.

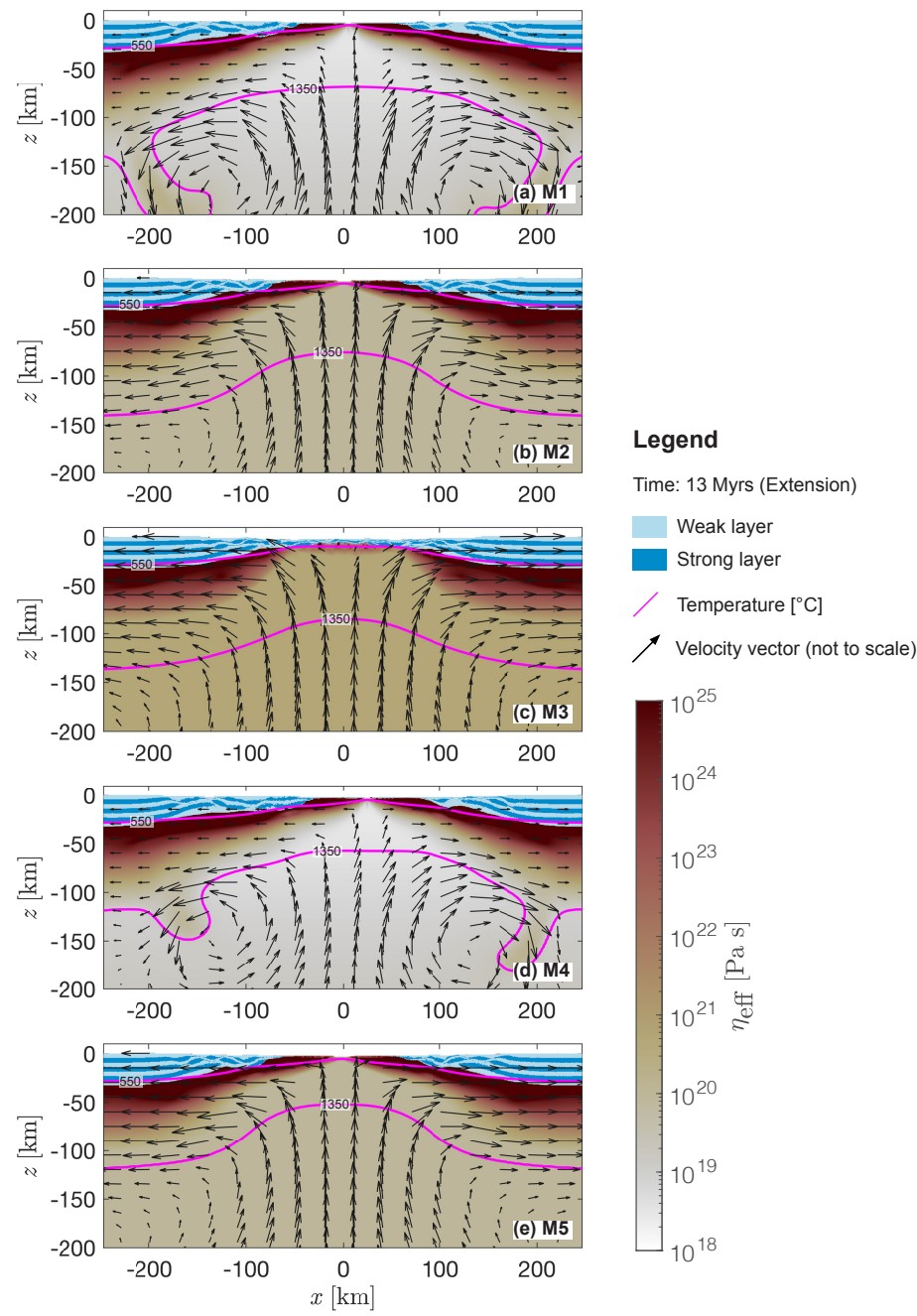

**Figure 5.** Comparison of M1-5 during the extension stage. (a)-(e) Enlargements of results of models M1-5 at 13 Myrs of modelled time. Blue colours indicate mechanically weak (light) and strong (dark) crustal units. White to red colours show the effective viscosity field calculated by the numerical algorithm. Magenta lines show the level of 550 °C and 1350 °C isotherm. Arrows represent velocity vectors and the length of the arrows is not to scale.

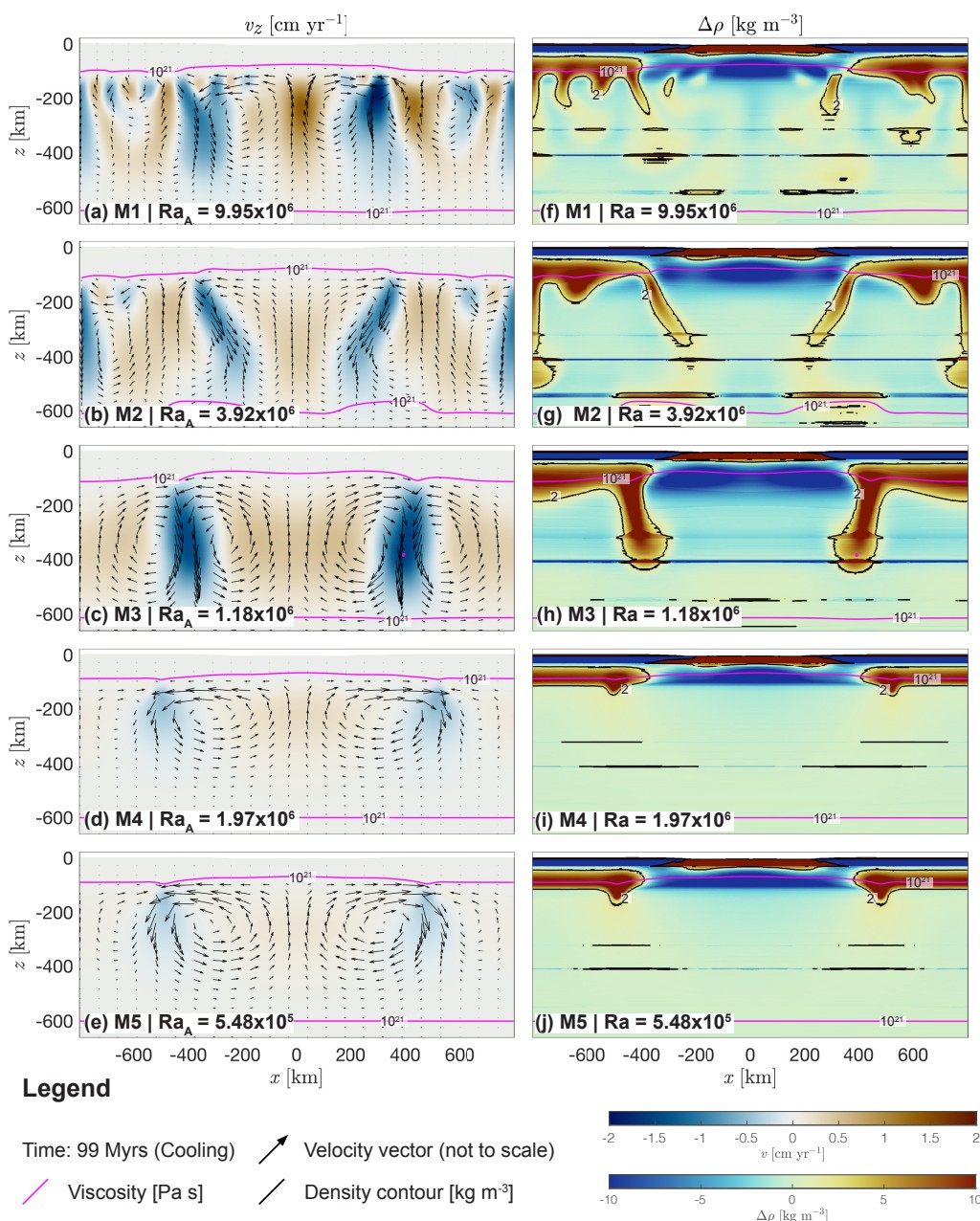

**Figure 6.** Results of models M1-5 at 99 Myrs (end of cooling period). (a)-(e) Blue to red colours indicate the vertical velocity magnitude calculated by the numerical algorithm. Black arrows show not to scale velocity vectors to visualise the material flow field. (f)-(j) Blue to red colours indicate the difference between the entire density field and the horizontal average density profile as shown in fig. 2(e). Black lines are the 2 kg m$^{-3}$ density contour. The magenta coloured line shows the level of the $10^{21}$ Pa s viscosity isopleth in all sub-figures. This contour line represents the mechanical boundary between the rigid mantle lithosphere (no velocity glyphs) and the convecting upper mantle.

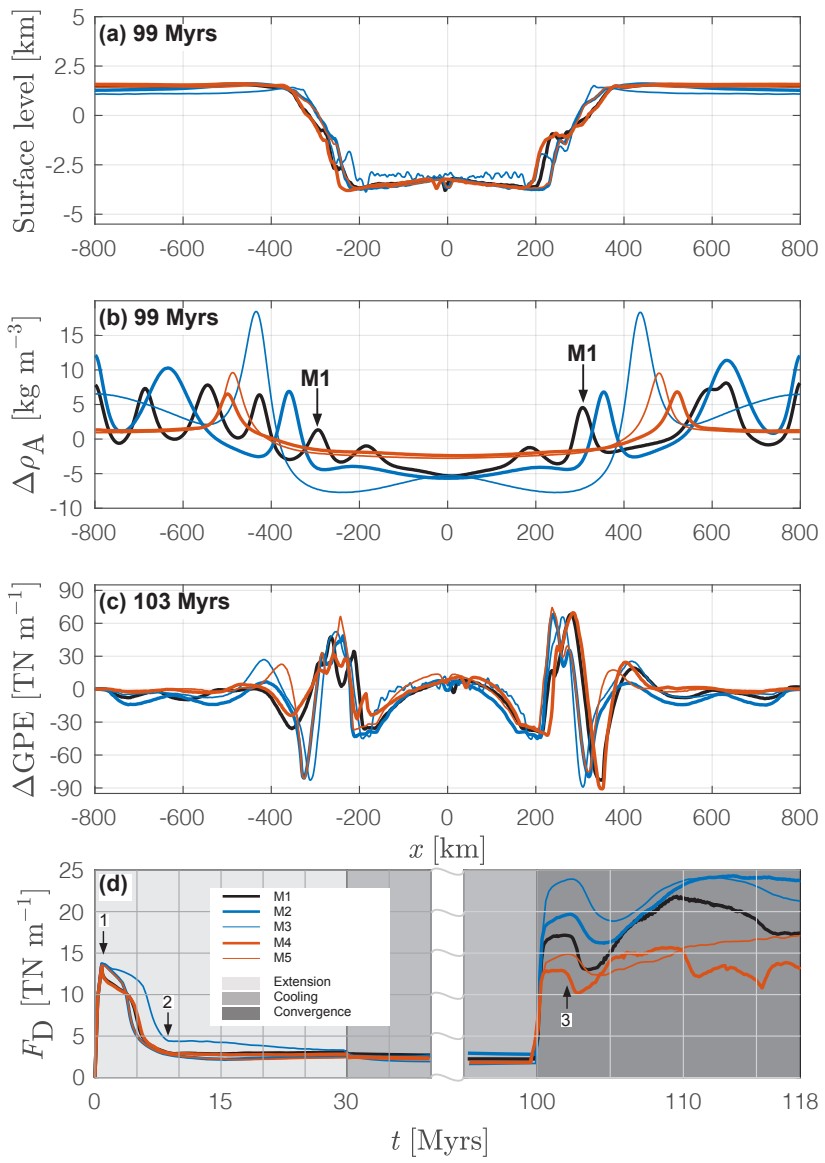

**Figure 7.** (a) Model topography at the end of the cooling period. (b) Vertically averaged ($-200\text{km} \leq z \leq -100\text{km}$) difference between entire density field and horizontal average density (see also fig. 6((f)-(j)). (c) Difference in *GPE* of models M1-5 after $103$ Myrs. (d) Estimated plate driving forces ($F_\text{D}$, see appendix D) through the entire model time. Grey regions indicate the stages of extension (light grey), cooling without plate deformation (grey) and convergence (dark grey). The cooling period is not entirely displayed, because values for $F_\text{D}$ remain constant when no deformation is applied to the system. Numbers indicate important events in the evolution of the models: 1 = Initiation of crustal necking, 2 = crustal break-up, opening of the marine basin in which the mantle is exhumed to the sea floor, 3 = subduction initiation via thermal softening. The legend shown in (d) is valid for all panels above as well.

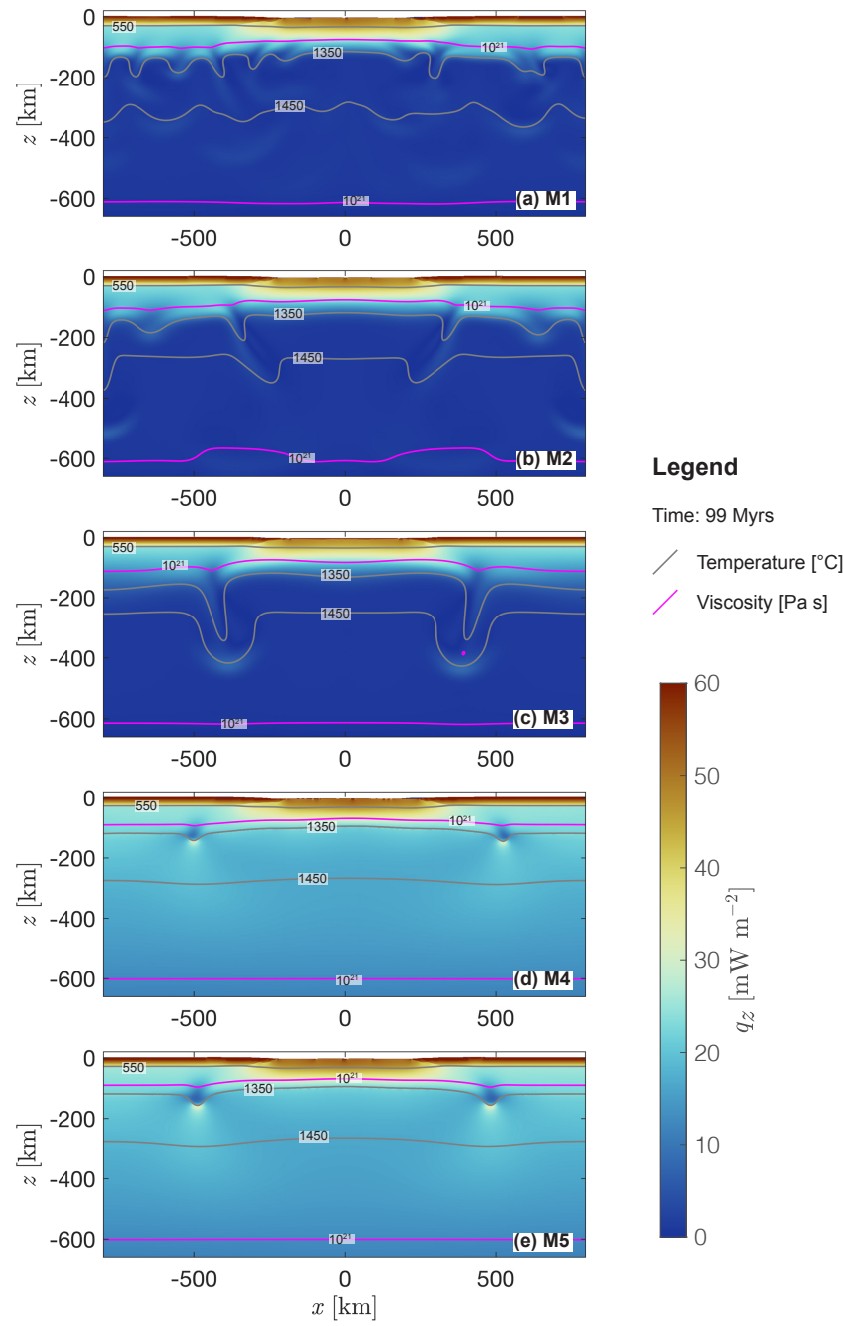

**Figure 8.** Vertical conductive heat flow represented by blue to red colours for models M1-5 ((a)-(e)) after 99 Myrs. The grey lines indicate the depth of the 550 °C, 1350 °C and 1450 °C isotherm. The magenta line indicates the depth of the $10^{21}$ Pa s isopleth which represents approximately the mechanical transition between the mantle lithosphere and the convecting upper mantle.

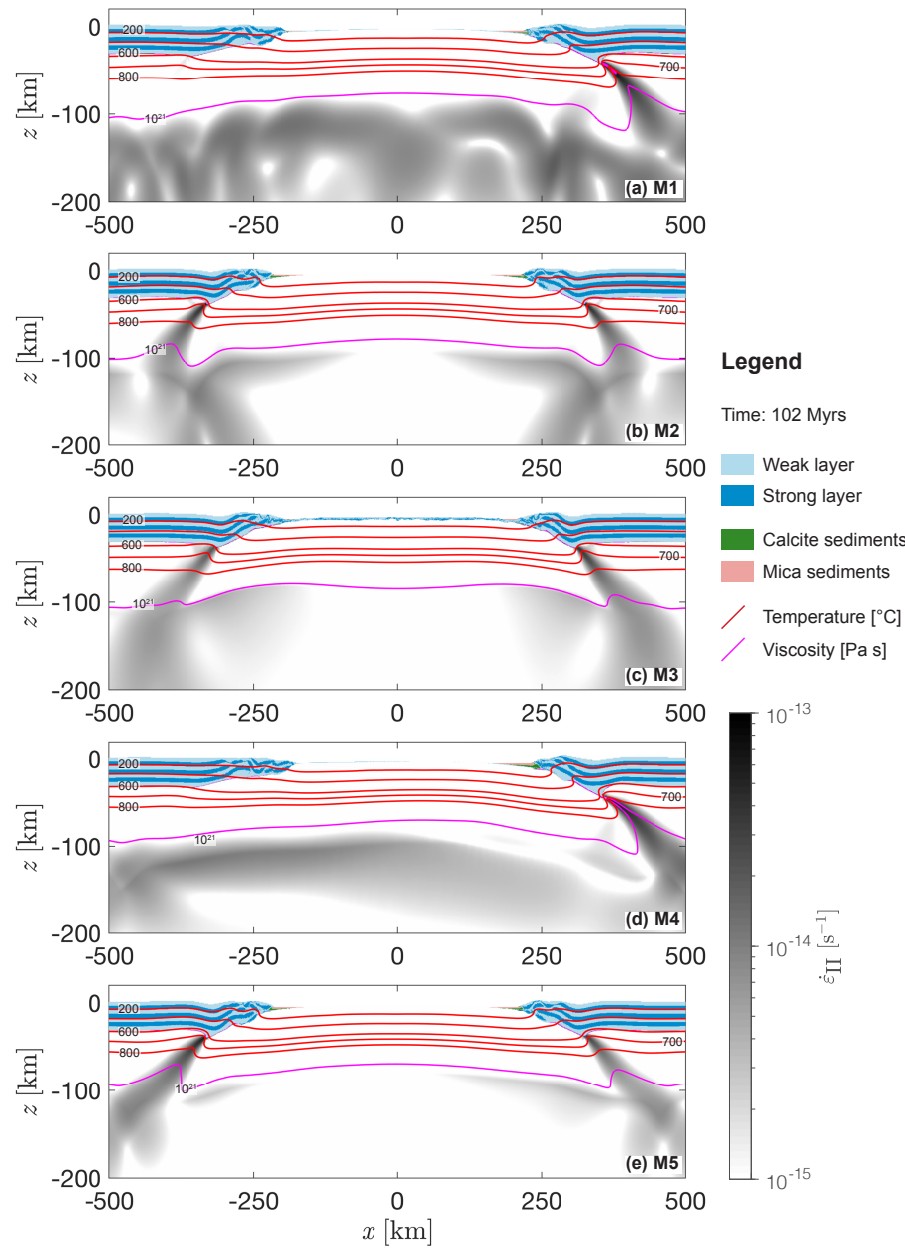

**Figure 9.** Stage of subduction initiation. (a)-(e) Results of models M1-5 after 102 Myrs of simulated deformation. Blue colours indicate mechanically weak (light) and strong (dark) crustal units, green and salmon colours indicate the sedimentary units (only minor volumes in trench regions). White to black colours indicate the second invariant of the strain rate tensor field calculated by the numerical algorithm. Red lines show levels of several isotherms and the magenta coloured line shows the level of the $10^{21}$ Pa s viscosity isopleth. This contour line represents approximately the mechanical transition between the mantle lithosphere and the convecting upper mantle.

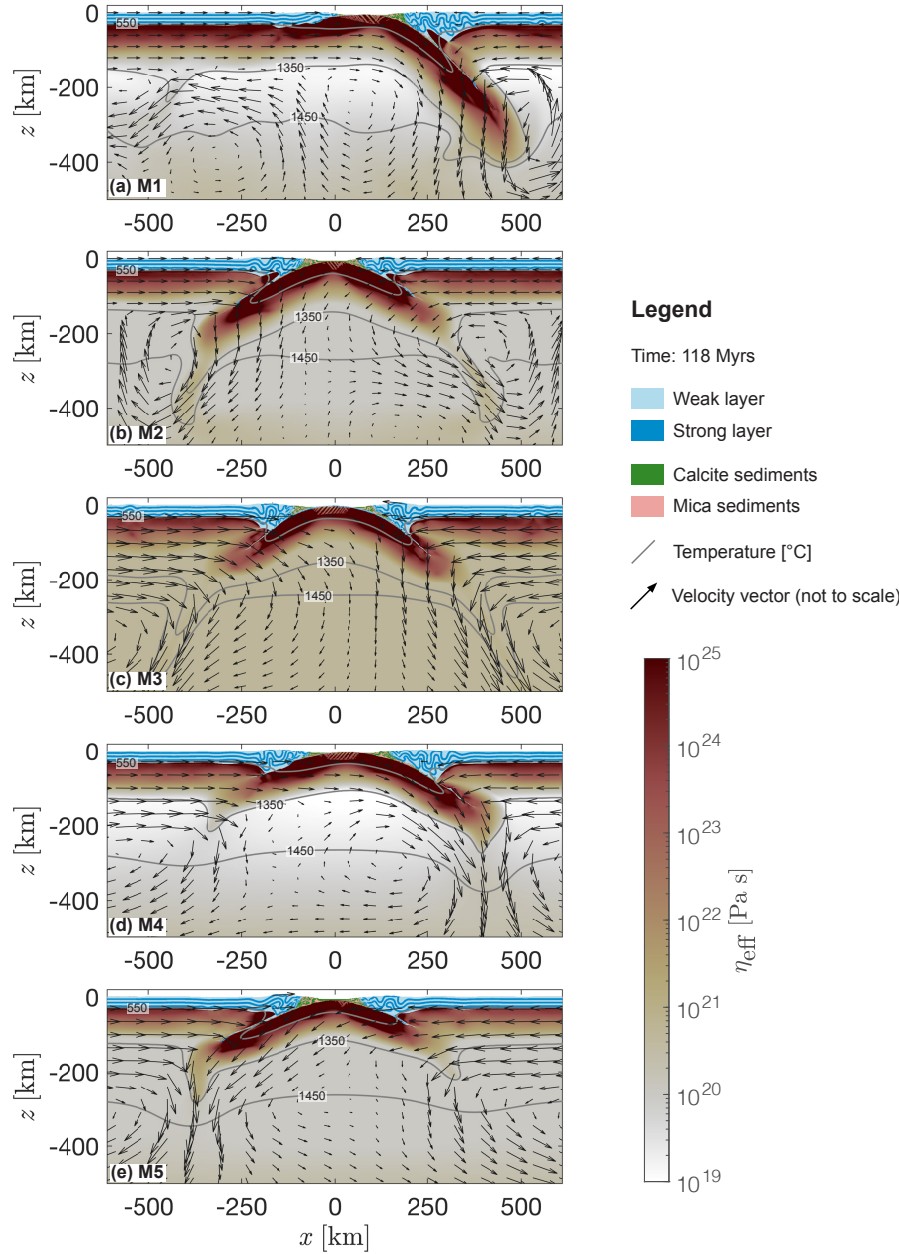

**Figure 10.** Evolution of subduction zones. (a)-(e) Results of models M1-5 after 118 Myrs of simulated deformation. Blue colours indicate mechanically weak (light) and strong (dark) crustal units, green and salmon colours indicate the sedimentary units. White to red colours show the effective viscosity field calculated by the numerical algorithm for the mantle lithosphere and the upper mantle. Arrows represent velocity vectors and the length of the arrows is not to scale. Grey lines indicate the 550 °C, 1350 °C and 1450 °C isotherm.

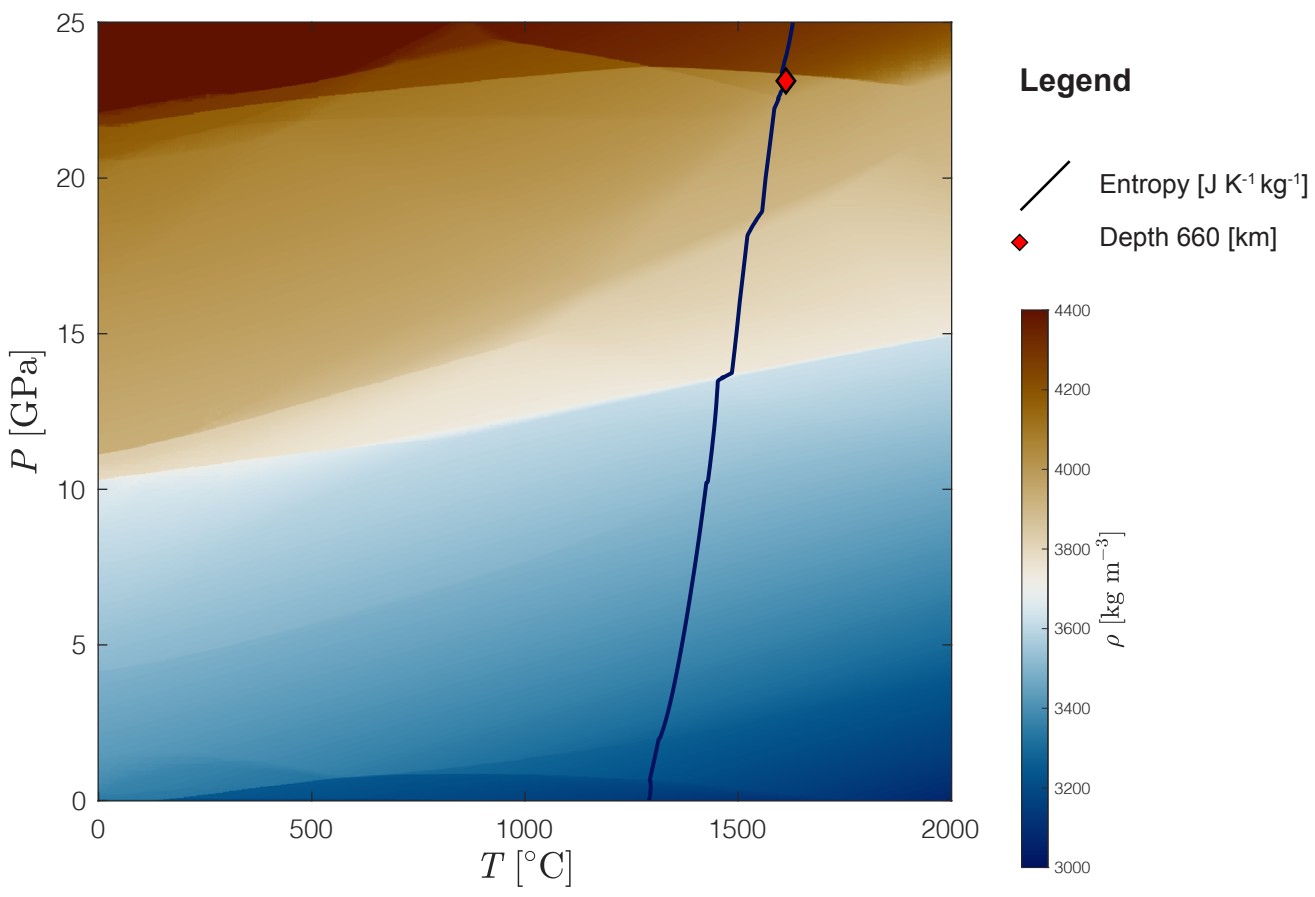

**Figure A1.** Hawaiian pyrolite phase diagram calculated with Perple_X (Connolly, 2005). Bulk rock composition in weight amount: 44.71 (SiO$_2$), 3.98 (Al$_2$O$_3$), 8.18 (FeO), 38.73 (MgO), 3.17 (CaO) and 0.13 (Na$_2$O). Bulk rock composition taken from Workman and Hart (2005). Blue to red colours indicate density calculated for the given pressure and temperature range, black line indicates the isentrop for a temperature of 1350 °C at the base of a 120 km thick lithosphere and the red diamond shows the pressure and temperature conditions at 660 km depth following this isentrop.

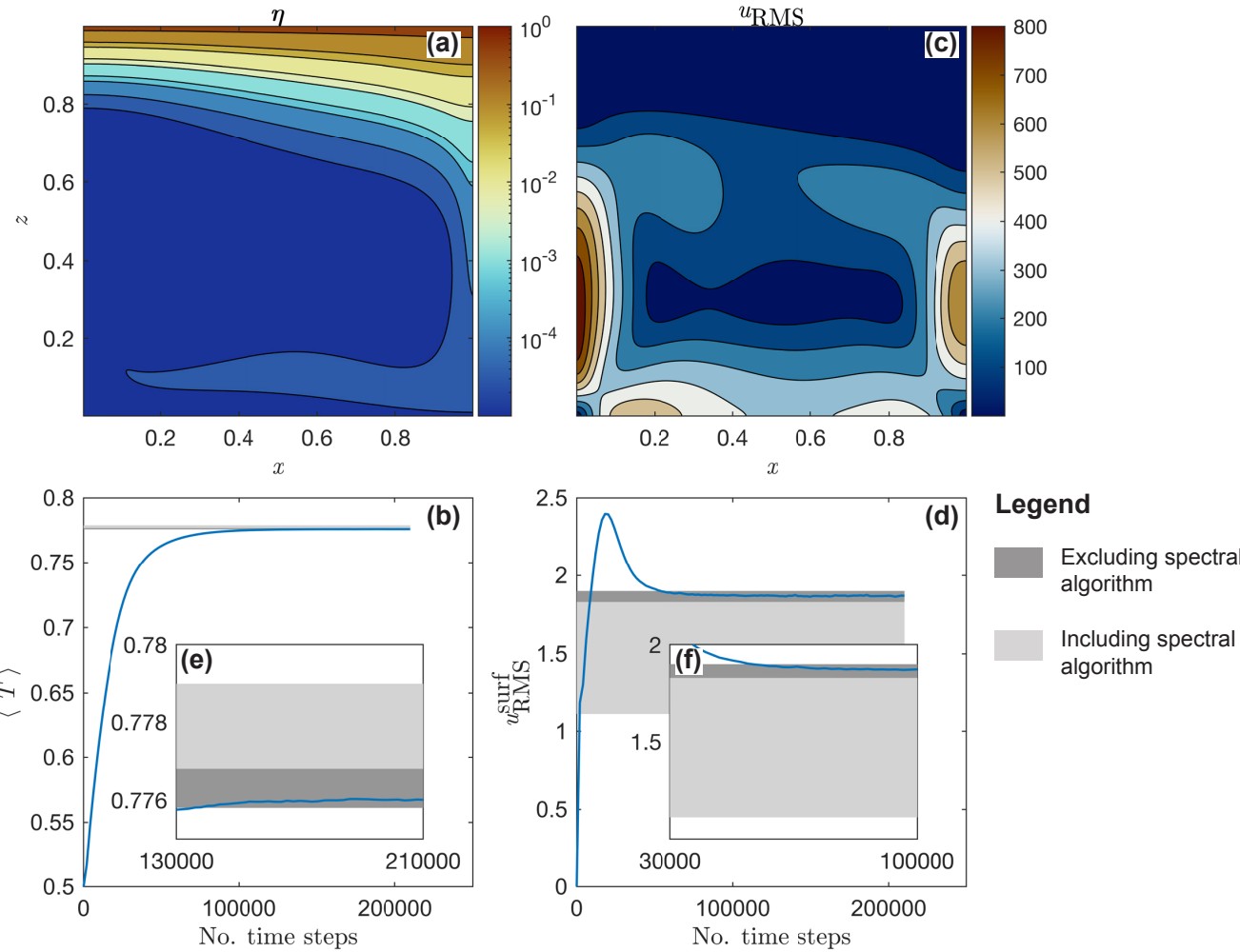

**Figure A2.** Results of 2D convection benchmark. (a) effective viscosity after convection reached a steady state, (b) normal average temperature calculated as in eq. C6 for the entire model history, (c) root mean square velocity field for the entire domain after convection reached a steady state, (d) root mean square velocity at the surface of the modelled domain of the entire model history calculated as in eq. C7, (e) normal average temperature calculated as in eq. C6 for the time period at which the convection reaches a steady state and (f) root mean square velocity at the surface of the modelled domain calculated as in eq. C7 for the time period at which the convection reaches a steady state. In fig. A2(b) and fig. A2(d)-(f), the dark grey area only shows the range of minimum maximum values for the given diagnostic quantity obtained by the algorithms tested by Tosi et al. (2015) excluding the spectral algorithm. Dark combined with light grey areas indicate the range obtained by all algorithms tested by Tosi et al. (2015) including the spectral algorithm.