# Peer review of "Impact of upper mantle convection on lithosphere hyper-extension and subsequent horizontally forced subduction initiation"

_Solid Earth, 2020_

## Referee Comment (RC1) · Anonymous Referee #1 · 26 Jun 2020

GENERAL COMMENTS

The manuscrit "Impact of upper mantle convection on lithosphere hyper-extension and subsequent convergence-induced subduction" by Candioti et al. presents 2-D thermo-mechanical simulations of successive geodynamic processes (continental extension, cooling with small-scale convection, convergence leading to single or double subduction initiation), with variations between the 6 models of either thermal conductivity parameterisation near the lithosphere-asthenosphere boundary, lower bound for viscosity, or rheological parameterization (wet vs. dry) of diffusion creep and dislocation creep in olivine.

[Figure]

Scientific significance: good

A motivation of the study is to model part of the alpine orogenic cycle, with continental rifting leading to a short-lived ocean that then subducts before collision. In my opinion, the most substantial contribution of this manuscript to the field of solid Earth dynamics is the novel model design that allows a self-consistent formation of structural inheritance at passive margins, which proves crucial for later subduction initiation geometry. As argued in section 4.5, this is a clear progress relative to the implementation of ad-hoc weak zones that do not capture the structural complexity related to the extension history. The paper also provides estimates of the force (per unit width) required to initiate subduction.

Scientific quality: fair

If the overall model design and parameterization seem valid, I have several issues with the model set-up:

- the fluid material is described as incompressible in section 2.1, but the thermal gradient is adiabatic, mantle densities vary greatly along the domain high due to phase transitions (Fig. A1) and the authors use the extended Boussinesq approximation for compressible fluid to solve the conservation equations (Appendix B line 579)

- phase transitions are implemented through the variable density (Fig. A1), but latent heat associated to phase change is missing from equation (A14), and this approximation should be justified.

- the side velocity boundary conditions during the extension or convergence phase (Fig. 1a,d) are likely to induce a sheared weak zone near the side boundaries at the transition depth between lateral inflow and outflow (340 km depth). Also, for the extension set-up, the suction created by the divergence in uppermost mantle is likely to generate a bulk ascending mantle channel in the middle of the domain (X=0 km).

- more generally, the flow pattern over the entire model domain is never shown when

in/out flows are imposed at the sides, and this is an issue when discussing application to Earth (section 4.4): what is the geological justification that the divergence or convergence was not only active at the plates' surface but also across 300 km in the mantle below the plates? The justification of the side velocity boundary conditions should be developed, and the global flow pattern (velocity glyphs/arrow) should be shown during extension and convergence phases.

I also have other issues with the methods and results analysis:

- the choice of which input parameter are varied is not explained: why choose to vary the minimum viscosity between models M1, M3 and M5, rather than for example the initial extension rate, the duration of the thermal relaxation phase or the inflow/outflow side velocity profiles? The discussion does not well explain why there is a single subduction in M1, but double subductions in M3/M5.

- a methodology study on the comparison of "explicitly modelled convection" and "effective conductivity mimicking a convective heat flow" is inserted in the middle of the main geodynamics study. This hinders the continuous read of the paper, and I suggest all analysis and related figures of models M2/M4 are moved to Appendix B, along with the heat flow profiles of Figure 10.

- the simulations have numerous features that do not seem relevant for the scientific question (erosion, sedimentations of alternating calcites and pelites), but add yet another set of free parameters that make the interpretation of the simulations more complex.

- despite its central importance, the paper lacks a clear definition of "convection", that sometimes means "advection" or "drag" or "flow". The manuscript could also maybe reference the following papers dealing with the plate-asthenosphere interactions or subduction initiation or various scales of mantle convection:

L. Husson. The dynamics of plate boundaries over a convecting mantle. Physics of the

Earth and Planetary Interiors, Elsevier, 2012, 212-213, pp.32-43.

V. S. Solomatov. Initiation of subduction by small-scale convection. Journal of Geophysical Research: Solid Earth, 2004.

F. Lévy, C. Jaupart. The initiation of subduction by crustal extension at a continental margin. Geophysical Journal International, Volume 188, Issue 3, March 2012, Pages 779–797.

N. Coltice et al. Interactions of scales of convection in the Earth's mantle. Tectonophysics. Volume 746, 30 October 2018, Pages 669-677

Presentation quality: poor

This is a major flaw of the manuscript, which scientific contributions are hard to unearth because of confusing text and figure organization.

For example :

- showing vertical and horizontal velocity background colors (Fig. 4, Fig. 9) makes it difficult for the reader to visualize the flow pattern > could the authors show velocity glyphs or arrows, to better reveal e.g. the wavelength of small-scale convection in Fig. 4

- Figure 5 should be referenced in the methods section since achieving realistic temperature, density and viscosity model output is rather a constrain on the input parameters than a surprising result

- Figure 8 and 10 are referred to very early in the text, whereas they belong in the discussion (or appendix?) rather than in the results section same for text on lines 187-199,156-164

- Appendix A belongs to the main text, otherwise the parameters of Table 2 are not defined

- a time-bar could be included in Fig. 1 showing to scale the 3 stages of boundary conditions with colours corresponding to the velocity profiles shown in Figure 1a,d (also please add a null-velocity profile fo the thermal relaxation). The same time bar could then be put on other figure to know at a glance which stage the figures belong to.

SPECIFIC COMMENTS

- in the introduction, the authors should define what they mean by "convection" and discuss the different scales

- the exhumation of hot mantle (Fig. 2) is expected to lead to melting, please comment

- you need to support some statements with results/data, i.e. "Alternating activity of the subduction zones is observed." (line 219)

- if you mention the importance of apply a force rather than a velocity BC, then you should also mention the importance of setting the lateral flow in/out of the domain

- I am not sure you can compare your model to the Atlantic (line 272) since old oceanic lithosphere there is much older than in your models

- I disagree with the statement "the models are in a state of isostatic equilibrium at the onset of subduction initiation." (lnie 278): the convergence velocity and the topograpic low above the new trench (Fig. 8) suggest a dynamicl topography

- line 294-307: the explanation of the control of signe-sided subduction is not clear

- you cannot claim that "mantle convection seems active and largely confined to the upper region of the upper mantle. The convective patterns simulated in our study are in agreement with these observations." (line 399) since you impose the hight of your simulation domain to be restricted to the upper mantle.

- you should not boast that "the model has captured correctly the first order physics of the investigated processes." since model M6 shows the immense importance of rheology parameterisation - that is far from being constrained...

- what do you mean by "If convection in the mantle is suppressed by high effective thermal conductivities or high, lower viscosity limits" (line 453)? During the convergence phases, the mantle still flows in the domain (which is why you should show the velocity glyphs).

- did you try to run models without shear heating to estimate the relative role of structural vs. thermal softening for localization? (lines460-462)

- Figure 9 and Figure 2a,b: comment on the "slab-like" features between 100 and 200 km depth below the extended margins in M1 and M2

- Fig.8d: what is the X-locations and the depth range for integration of the second invariant of deviatoric stress tensor?

- explain in caption of Fig. 8 "values for tau_II remain constant when no deformation is applied to the system" whereas Fig. 4 shows large convection cells un the mantle that may deform the plates above

- appendix B: it is not clear why D should be thickness of the whole upper mantle whereas Fig. 4a,f shows small convection cells

- equation B1: how is effective viscosity average over the domain?

- the explanation on lines 554-561 is not convincing: what take a constant Rayleigh number that on D and on k and then claim that D and k can be adjusted?

- the isentrop is Fig. A1 does not match the temperature profile in Fig. 1

TECHNICAL CORRECTIONS

- "cooling" is more appropriate than "thermal relaxation" for stage 2

- the initial velocity condition is not given

- do you have more references for the "common approach" to indirectly include the effects of thermal convection ? (line 48)

- line 97-98 : what does "free slip with constant material inflow/outflow velocities" mean?

- 7 units of 5 km each make a thickness of 35 km (not 33)(line 104)

- line 104 : why describe a 87-km thick mantle lithosphere if all parameters are the same (line 114)

- Table 1 : please higlight (bold ?) which parameters differ from model M1 for all models.

- Table 2 : how are the column of the 2 sediments different? link with pelites/calcites or with sediments 1/2 of Figure 6?

- Table 2 : why no diffusion creep in the crust ?

- Table 2 : which rock are analogue for strong and weak crust?

- section 3.1.1: how do you define the length of the margin (threshold in crust thickness?)

- line 160: is the second invariant tensor of the deviatoric stress calculated for the whole lithosphere including the crust?

- why do you take the $10^{21}$ Pa.s contour as the base of the lithosphere? Why not take th $1350°C$ isotherm?

- line 176: give X-location of special flow fiel at 120 km depth

- line 184: why doyou claim that the lithosphere is delaminating whereas the iso-viscous contour is almost flat?

- line 207: define GPE

- line 223: Figure 8d rather than 8b?

- rephrase "In our models, subduction is initiated self-consistently, without prescribing any major weak zone or an already existing slab." (line 286) since they are weak

heterogeneities in the passive margin

- line 319 "if shear stresses are negligible" = is that really the case at suduction onset?

- equation A3: define alpha and beta (which is different from the beta in Eq. B2 I guess...)

- Fig. 1a,d: the depth looks smaller than 680 km

- Fig. 1c: initial random perturbations look denser between -20 and + 20 km, is that the case?

- Fig. 3d: issue with the bottom of the plit nera -200 km (vertical grey line?)

- Fig. 5: dashed lines for M4 and M5 are barely visible, I suggest you use thick lines with other colours

- Fig. 8: it would be helpful to have label on the topopagray such as "trench", and to mark the subduction initiation in the timeline of Figure 8d

- Fig. 10: what is the new information brought by this figure compared to Fig. 4?

---

## Referee Comment (RC2) · Anonymous Referee #2 · 7 Jul 2020

The study presented here uses 2D thermo-mechanical models to investigate formation of hyper-extended passive margins (extension), thermal relaxation, and subduction initiation during convergence. I find it an interesting study, with lots of complex aspects and I think considerable work was put into it. The manuscript has the potential to show some interesting results regarding subduction initiation and with respect to structural inheritance, however, I find the manuscript needs some further polish of the arguments in order to be publishable. Some parts are well written (i.e., Discussion) and the figures are high quality, but other parts need more work and clarification.

Most importantly, the key findings do not stand out after just one read of the paper.

[Figure]

Both the abstract and the conclusion are very long and very stuffy. Also, there is a weak correlation between arguments in introduction-results (investigation)-discussion. I therefore recommend the manuscript to be published in Solid Earth, after some major and minor comments below have been addressed.

Major points:

1. Clarity of the manuscript. The model is very complex and it has lots of details, but the authors haven't always explained the concepts clearly or properly. I have written down some specific examples that the authors can fix easily. However, they should try to verify that their findings are backed by arguments that are explained in a logical way.

The abstract should be shortened to include the top 3 most important results, and be revised for clarity and shorter sentences. For example, what do the authors want the paper be known/cited for?

In the introduction, the link between mantle convection and lithosphere deformation is quite abrupt (with a sentence about age of the Earth that is irrelevant to this study). The question 'why is convection important?' is not satisfactorily introduced or linked to coupled lithosphere-mantle deformation.

General suggestion: too many commas. Try to rephrase/split sentences with more than 2 commas or that are longer than 2 lines.

2. Results section. I think the reference model (M1) should be described separately (evolution between extension, relation, convergence). Then compare models M2-M6 with M1 to highlight the effect of various factors. Figures should be adapted accordingly. The reason for this are the following: - in current form, the comparison is all over the place and it is confusing. It is not very clear which simulations the main text is referring to sometimes. - the current arrangement of figures is random. It starts with 2, 5, 8, 4, 10 etc. Their placement should follow a logical order of arguments.

The comparison between M1-M6 should be done in terms of Ra. The k, viscosity cutoff,

flow laws, they essentially affect the Ra.

3. Thermal softening. A quick search in the manuscript finds 'thermal softening' only in the abstract, very late discussion and conclusion, yet it is suggested as a key process that controls subduction initiation. I'm pointing out that it is incompletely described and linked to the hypothesis of the study and results.

For example, Line 425: thermal softening is introduced only now. not clear why bring it up here?

Line 461: say that structural and thermal softening are important, but they were introduced late, without much context.

Moreover, the authors suggest in multiple places that it is the structural softening (inheritance) and convection (slab suction) that help initiation. The authors need to clarify what are the main findings, and arguments need to be revised. One finding that I think is important: the required driving force to initiate subduction is much smaller, when convection and structural inheritance are considered.

4. Modelled vs parameterised convection. In Line 126, 3 types of simulations are introduced: 1) model convection with a weak asthenosphere, 2) parametrised convection, by scaling the thermal conductivity to the Nusselt number 3) impact of different viscosity structures

First, the treatment of the mantle convection is not clear in the main text (Lines 124-130). What drives convection? How is the applied parametrised convection different? When is the onset of convection? Is convection only during the thermal relaxation stage? What controls the size of the convection cells? Also, explain how the Ra_avg is calculated.

In point 3) above which approach are you using: modelled/parameterized convection? While it is explained better in Appendix B, the differences between them are not clear in the main text. For example, 1) would be M1, while 2) is M, and 3) is M6?

5. Other questions.

The geodynamic cycle modelled: 1) 30 Myr extension at 2cm/yr 2) 70 Myr thermal relaxation 3) 20 Myr convergence at 3 cm/yr

What is the motivation behind these choices: 1) why thermal relaxation 2) why those time intervals 3) why those extension/convergence rates? Also, what are the boundary conditions during thermal relaxation?

Why the choice of those parameters to change?

Are the surface processes important? Have you run models without? Do they introduce further heterogeneities in the model that affect the outcome?

6. Subduction initiation. It seems like symmetric vs asymmetric spreading also controls to a large extent subduction initiation, whether it is single/double subduction. I feel very little discussion is about that, and more on structural and thermal softening.

Also, there are other previous efforts to model extension/compression to obtain structural inheritance and subduction initiation (i.e. Gulcher et al 2019). The authors discuss simpler treatments of subduction initiation in paragraph 280, but do not relate to newer efforts to avoid the use of artificial features. So, are these newer models better for studying subduction initiation?

Minor points:

Line 8-10: revise sentence

Line 10: only from the abstract it is not clear what the parameters were used, so saying that a viscosity of 5e20 Pa.s was used (as compared to what?) is not very meaningful. Rephrase

Line 20: multiple use of 'geodynamic' in the same sentence

Line 29-30: rephrase.

Line 31: while it is an interesting fact - the calculation of the age of the earth - is not very relevant to the manuscript.

Line 35: rephrase

Line 40: unlikely to be problematic

Line 41: delete likely

Line 55: authors relate to numerical aspects such as time step size, without mentioning why? The context was on physical aspects of convection.

Line 64-65: should be in the first paragraph of introduction

Line 68: Why only upper mantle? This is discussed late in discussion (section 4.4, paragraph 395)

Line 69: delete 'of applying'

Line 74: revise sentence - its meaning is not clear to someone who hasn't read the methods/results section.

Line 88: reference to the code how it was benchmarked? (Info in appendix A, but should be in the main text too)

Line 89: rephrase

Line 96: repeats with Line 92, also Duretz et al 2016/2016a?

Line 105: based on the sentence the crust should be: 3*5+4*5 = 35 km thick. But a sentence earlier it is 33km

Line 103: what is the mathematical expression for the perturbation? in case the model needs to be reproduced?

Line 111: more details on the rheology? Indicate appendix A for reference

Line 113: reference to "corresponding laboratory flow law estimates"?

Line 113-114: rephrase. i.e. The mantle lithosphere is rheologically stronger than the mantle asthenosphere due to the temperature gradient.

Line 120: what is the motivation for alternating between calcites and pelites for sedimentation algorithm?

Line 130 - give reference to Table 1.

Line 133: viscosity cutoff for M1 is not provided to understand the difference.

Line 134: realistic value? Are the other values not realistic?

Line 148: Figure 2

-> can define a variable F = 2xtau_II

Line 160: introduce the horizontal driving force per unit length, but what is it proxy for?

Line 168: you can't see to a depth of 660 km as indicated

Line 195: values

Line 197: what is the delta GPE showing? (Info given in appendix)

Line 224: reference to fig 9a, yet that figure is for extension stage.

Paragraph 220-225: confusing.

Line 228: what is mechanical heterogeneity? increases the strength of the weak layers

Line 233: breaks later than the continental. after gives the impression of location.

Line 234: what do you mean ' Mantle convection does not establish as early as rifting and crustal separation.'?

Lines 243, 245: use of realistic. close to the Ra estimated for the Earth.

Line 250: which modulates mantle velocities.

Line 250-254: why the discussion on time step size (a numerical feature) here?

Lines 255-257: which simulation results are the authors referring here?

Paragraph 258: reference figure 5e,j in this paragraph. Also, maybe plot density averages in passive margins/exhumed mantle separately?

Line 272: you jump from density differences to values of tectonic forces. An additional sentence needs to connect them (i.e. estimate the buoyancy force due to modelled density differences). How much is needed to initiate subduction? (a similar calculation is done in line 315)

Paragraph 280: this should come before the Cloos 1993 paragraph

Line 286: yes, but under convergence

Line 294-295: total convergence is double sided, while in M1 is single-sided (asymmetric). Not clear why subduction initiation is stable only in M1. Convection cell size important? how about thickness of lithosphere at the point? M2-M5 are quite symmetric and they all have Ra_avg ∼1e5, while M1 has Ra_avg∼1e6. That should have an effect.

Line 312-315: - suction force induced by down-welling in the convection cell in M1. What is the similar force in the other simulations?

Line 330: not sure what the reference is for. The double-subduction term was not coined by those workers.

Line 332: sentence not clear. Which simulation are you referring? would say M2-5 are more or less symmetric double subduction

Line 343: onset of convergence - unclear when this happens?

Line 357-361: use of 'realistic'

Paragraph 395: this paragraph should be in the methods, as it motivates/explains your

model domain until 660 km. The sentence 'The convective patterns simulated in our study are in agreement with these observations.' is irrelevant because you don't model the lower mantle.

Paragraph 410: this should come earlier - I had questions about it earlier. on previous work on subduction initiation.

Line 418: most definitely will have an impact

Paragraph 430: and melting

Line 444-446: rephrase/simplify.

Figures and Tables: Table 1: thermal conductivity should be 'k' without the 'th' subscript. The authors can also provide the formula for the Ra number in the main text. How was the Ra_avg calculated?

Table 2: there should be a column 'Description' to describe the meaning of each parameter i.e. 'rho0' - reference density. Use k instead of k_th for thermal conductivity. What is dry/wet mantle? I assume wet mantle applies only to M6? Plastic and elastic parameters are also listed. Not very clear in the main text.

Figure 4: why plot the vertical velocity field separate from the horizontal? should plot arrow/streamlines field to see the convection cells.

Figure 10: should be merged with Figure 4. One column velocity, one column temperature.

Figure 5: What if you plot the profiles at the rift axis (within a distance) vs off-axis on either flanks of the rifts? Caption: g-j show enlarged areas.

Figure 6: legend: temp contours are red.

Figure 8: the line plots are not entirely clear. Maybe use a dotted line instead of dashed line? and same thickness.

Appendix A

Line 481: that's a strange notation of i,j indices (Einstein notation).

Eq A2: if written in Einstein notation, then vectors are written in terms of scalar components ($a\_i$ should not be bold). Same in Line 482 a=[0,g]. -> revise this appendix for completeness of sentences, and explanation of all parameters. For example, what is Ap, tauP etc. Gamma value? in eq A10

Appendix B

Paragraph 531: rephrase

Line 560: not clear

Appendix C

Line 595: gamma_T=1?

Line 601: g=10ˆ4?

───────────────────────────

---

## Author Comment (AC1) · 4 Aug 2020

We gratefully thank the reviewer for the very constructive criticism. Implementing the suggestions helped to better visualize the simulation results, focus on the main findings and significantly improve the manuscript. During the review process, we have changed the structure of the manuscript significantly. In the results section of the revised manuscript, we present the evolution of the reference model and the wet olivine model separately. The results of the remaining models are presented in comparison to the results of the reference run for the distinct deformation stages. We then discuss the implications of our findings on several aspects, such as for example the impact of

the viscosity structure on the convection, the onset of convection and the impact of convection on subduction, in the discussion section. The order of the figures and the style of visualization has been adapted accordingly.

Below we have answered to all the comments from the reviewer. Our answers are marked with "A:" and are below the original comments.

- the fluid material is described as incompressible in section 2.1, but the thermal gradient is adiabatic, mantle densities vary greatly along the domain high due to phase transitions (Fig. A1) and the authors use the extended Boussinesq approximation for compressible fluid to solve the conservation equations (Appendix B line 579)

A: The maximum value for the density time derivative is two orders of magnitude smaller compared to the velocity divergence. Also, (Bercovici, Schubert , & Glatzmaier, 1992) concluded that compressibility effects on the spatial mantle structure are minor when the superadiabatic temperature drop is close to the adiabatic temperature of the mantle, which is the case for the Earth. We therefore assume here that the Boussinesq approximation is still valid and suggest that density changes due to volumetric deformation are negligible. We consider only small density changes affecting the buoyancy stresses. Not considering adiabatic heating in the energy conservation equation leads to a significant deviation of the thermal structure from the initially imposed adiabatic temperature gradient over large time scales (>100 Myrs). The resulting vertical temperature profile (if adiabatic heating is neglected) is constant throughout the upper mantle and the newly equilibrated vertically-constant temperature is equal to the imposed temperature at the bottom boundary. In consequence, the density structure taken from the phase diagram table according to pressure and temperature values is wrong. By using the extended Boussinesq approximation, i.e., the adiabatic heating term is included in the energy conservation equation but not in the continuity equation, the initially imposed adiabatic (or isentropic) gradient is maintained over long time scales. The resulting density structure agrees well with the PREM model as shown in this study. A more detailed comparison between different approximations of the conti-

nuity equation is beyond the scope of this study.

- phase transitions are implemented through the variable density (Fig. A1), but latent heat associated to phase change is missing from equation (A14), and this approximation should be justified.

A: Indeed, latent heat released or consumed by a phase transition can perturb the thermal field by up to 100 K and induce a buoyancy force aiding or inhibiting the motion of cold subducting slabs (van Hunen, van den Berg, & Vlaar, 2001) or hot rising plumes. However, when the lateral differences in temperature do not vary much, the deflection of the phase transition by an ascending plume or a subducting slab has a much bigger impact on the buoyancy stresses than the latent heat released or consumed by the phase transition (Christensen , 1995). Also, most of the studies on the impact of latent heat rely on the assumption that density is temperature dependent only. In the models presented here, density is a function of temperature and pressure, which makes it difficult to estimate the impact of temperature changes due to latent heat near a phase transition on the buoyancy term a priori. Because a detailed parametric investigation of the impact of latent heat on buoyancy stresses for temperature and pressure dependent density is beyond the scope of study, we neglect latent heat for simplicity.

- the side velocity boundary conditions during the extension or convergence phase (Fig. 1a,d) are likely to induce a sheared weak zone near the side boundaries at the transition depth between lateral inflow and outflow (340 km depth). Also, for the extension set-up, the suction created by the divergence in uppermost mantle is likely to generate a bulk ascending mantle channel in the middle of the domain (X=0 km). - more generally, the flow pattern over the entire model domain is never shown when in/out flows are imposed at the sides, and this is an issue when discussing application to Earth (section 4.4): what is the geological justification that the divergence or convergence was not only active at the plates' surface but also across 300 km in the mantle below the plates? The justification of the side velocity boundary conditions should be developed, and the global flow pattern (velocity glyphs/arrow) should be shown during

extension and convergence phases. I also have other issues with the methods and results analysis:

A: As mentioned correctly by the reviewer, material inflow/outflow velocity boundary conditions may lead to a shear zone forming at the transition point between inflow and outflow. To avoid such boundary condition effects close to the mechanical lithosphere, we set the transition point deeper than the initially imposed lithospheric thickness. We chose z=-330 km as the point of transition, because values for deviatoric stresses at this depth are significantly smaller compared to those at the base of the lithosphere. The mechanical thickness of the lithosphere may vary greatly over time mainly due to temperature changes and is therefore difficult to constrain a priori. By setting the transition point of the velocity inflow/outflow boundary condition deeper than the initially imposed mechanical thickness of the lithosphere, we allow for self-consistent adjustment of the mechanical thickness of the lithosphere during the evolution of the model. We have added velocity arrows to many of our revised figures. These figures show that the thickness of the lithosphere, with respect to consistent horizontal velocities, is not controlled by the inflow/outflow boundary away from the model boundaries.

- the choice of which input parameter are varied is not explained: why choose to vary the minimum viscosity between models M1, M3 and M5, rather than for example the initial extension rate, the duration of the thermal relaxation phase or the inflow/outflow side velocity profiles? The discussion does not well explain why there is a single subduction in M1, but double subductions in M3/M5.

A: One aim of this study is to quantify the impact of convection in the upper mantle on the long-term extension–cooling–convergence cycle of the lithosphere. Convection is controlled by the Rayleigh-number, which is a function of the viscosity of the convecting layer. Since the viscosity of the mantle is poorly constrained and estimated values vary by two orders of magnitude, this parameter is the most interesting for us to investigate. A frequently used technique to parameterize the impact of convection is the effective conductivity approach. This way the temperature field can be stabilized without having to calculate an enhanced convective velocity field in the mantle (which can be numerically challenging). We therefore compare this approach to the explicitly modelled convection, which calculates an enhanced convective velocity field in the mantle. The initial extension rate has been chosen to model the formation of an approximately 400 km wide basin containing exhumed mantle in an ultra-slow to slow spreading environment. The timing of the extension, cooling and convergence phases are motivated by data from the Alpine orogeny, which is now clearer explained in the revised manuscript. Testing the impact of different extension and compression rates as well as different cooling durations is beyond the scope of this study.

- a methodology study on the comparison of "explicitly modelled convection" and "effective conductivity mimicking a convective heat flow" is inserted in the middle of the main geodynamics study. This hinders the continuous read of the paper, and I suggest all analysis and related figures of models M2/M4 are moved to Appendix B, along with the heat flow profiles of Figure 10.

A: The manuscript and result presentation have been re-structured based on the constructive comments of both reviewers. However, we keep the results of the simulations with an "effective conductivity" in the main manuscript, because these results are part of our main geodynamics study.

- the simulations have numerous features that do not seem relevant for the scientific question (erosion, sedimentations of alternating calcites and pelites), but add yet another set of free parameters that make the interpretation of the simulations more complex.

A: We include erosion and sedimentation to avoid too high or low topography. To achieve this, we decided on one particular erosion/sedimentation model. A parametric study on the impact of different sedimentation/erosion processes or sediment transport mechanisms on the deformation of the lithosphere is beyond the scope of this study.

- despite its central importance, the paper lacks a clear definition of "convection", that

sometimes means "advection" or "drag" or "flow".

A: The word "convection" can be used to describe the motion of any fluid. Convection in a fluid can either develop freely, via thermal or compositional variations, or it can be induced by external forces (Ricard, 2007). In mantle convection simulations, free material motion in the mantle can be initiated by for example buoyancy contrasts due to variations in temperature or chemical composition. A geological example for induced mantle flow is a rigid plate that moves on top of the mantle. These statements have been incorporated to the introduction during the review process.

The manuscript could also maybe reference the following papers dealing with the plate-asthenosphere interactions or subduction initiation or various scales of mantle convection: L. Husson. The dynamics of plate boundaries over a convecting mantle. Physics of the Earth and Planetary Interiors, Elsevier, 2012, 212-213, pp.32-43. V. S. Solomatov. Initiation of subduction by small-scale convection. Journal of Geophysical Research: Solid Earth, 2004. F. Lévy, C. Jaupart. The initiation of subduction by crustal extension at a continental margin. Geophysical Journal International, Volume 188, Issue 3, March 2012, Pages 779–797. N. Coltice et al. Interactions of scales of convection in the Earth's mantle. Tectono- physics. Volume 746, 30 October 2018, Pages 669-677

A: We have added some of the references to the revised version of the manuscript.

Presentation quality: poor

This is a major flaw of the manuscript, which scientific contributions are hard to unearth because of confusing text and figure organization. For example : - showing vertical and horizontal velocity background colors (Fig. 4, Fig. 9) makes it difficult for the reader to visualize the flow pattern > could the authors show velocity glyphs or arrows, to better reveal e.g. the wavelength of small-scale convection in Fig. 4

A: We have added the velocity arrows to the figures. This was a very constructive

comment for better visualization – thank you.

- Figure 5 should be referenced in the methods section since achieving realistic temperature, density and viscosity model output is rather a constrain on the input parameters than a surprising result

A: This has been done during the review process.

- Figure 8 and 10 are referred to very early in the text, whereas they belong in the discussion (or appendix?) rather than in the results section same for text on lines 187-199,156-164

A: The figure order has been changed in the review process.

- Appendix A belongs to the main text, otherwise the parameters of Table 2 are not defined

A: We have decided to shift the parameter table 2 to the appendix rather than describing all equations in the main text.

- a time-bar could be included in Fig. 1 showing to scale the 3 stages of boundary conditions with colours corresponding to the velocity profiles shown in Figure 1a,d (also please add a null-velocity profile fo the thermal relaxation). The same time bar could then be put on other figure to know at a glance which stage the figures belong to.

A: We have tried to add the time bar, but the figures become too busy. We have changed the order of the figures and implemented much of the constructive comments.

SPECIFIC COMMENTS - in the introduction, the authors should define what they mean by "convection" and discuss the different scales

A: This suggestion has been implemented in the revised version of the manuscript.

- the exhumation of hot mantle (Fig. 2) is expected to lead to melting, please comment

A: This is explained in the introduction line 70 ff. and also in the methods section.

- you need to support some statements with results/data, i.e. "Alternating activity of the subduction zones is observed." (line 219)

A: This has been done in the review process.

- if you mention the importance of apply a force rather than a velocity BC, then you should also mention the importance of setting the lateral flow in/out of the domain

A: The timing of the deformation periods is chosen to allow for comparison with orogenies such as, f.e. the Alps. By choosing inflow/outflow we simulate the movement of the plates, decoupled from the rest of the domain, which we consider here more realistic than extending or compressing the entire side walls of the model domain with a height of 660 km.

- I am not sure you can compare your model to the Atlantic (line 272) since old oceanic lithosphere there is much older than in your models

A: . . . Even that old oceanic crust did not undergo spontaneous subduction yet, which was the point to mention it as an example here.

- I disagree with the statement "the models are in a state of isostatic equilibrium at the onset of subduction initiation." (line 278): the convergence velocity and the topographic low above the new trench (Fig. 8) suggest a dynamic topography

A: Indeed, this was our mistake: we chose the wrong time step to show the topography at the end of the cooling period. This has been updated during the revision. In general, it is also true that the system cannot be in isostatic equilibrium by definition, since there are deviatoric stresses holding the topography of the passive margins. However, the difference between the height of margins and the depth of the basin is ca. 5 km, which is the calculated topographic difference for an idealised block of 30 km thick crust floating on top of the mantle (Turcotte & Schubert, 2014). We therefore argue that the topography across the passive margin system produced by our model is close to isostatic equilibrium at the end of the cooling period.

- line 294-307: the explanation of the control of single-sided subduction is not clear

A: The down-welling of two convecting cells meet directly below the margin at which subduction is going to be initiated later. The enhanced downward motion of upper-mantle material in this region likely exerts a suction force that assists in initiating and stabilising only one single-slab subduction.

- you cannot claim that "mantle convection seems active and largely confined to the upper region of the upper mantle. The convective patterns simulated in our study are in agreement with these observations." (line 399) since you impose the height of your simulation domain to be restricted to the upper mantle.

A: We want to highlight here that we model a convective pattern that is observed in nature, namely convection in the upper mantle, that is below the lithosphere and above 660 km. Other people might argue that such "upper-mantle convection" does not exist and only "one-layered convection" of the entire mantle, down to ca. 2900 km, exists in nature. We confined our model domain using the justification that in the Alps the convection seems to be two-layered. Besides that, all models are confined to the upper mantle, but the reach of the convection cells extends to different depth, depending on the Rayleigh-number (compare M1 to M3). This is now more obvious when visualising the flow pattern with velocity vectors.

- you should not boast that "the model has captured correctly the first order physics of the investigated processes." since model M6 shows the immense importance of rheology parameterisation - that is far from being constrained...

A: As mentioned, we calibrated our initial model configurations in such a way that they match data from the PREM model and from GIA estimates. Starting from this point, we let the model freely evolve and do not change material parameters or geometries anymore. During rifting we generate margins of realistic first-order geometry, during cooling, the basin subsides to realistic depths and convection has realistic Rayleigh numbers, and finally subduction is initiated self-consistently (i.e. without imposing any

major triangular weak zone cutting through the entire lithosphere at the OCT) via thermal softening. Therefore, we argue that model M1 captures the first order physics of the extension-cooling-subduction cycle correctly. M6 shows a scenario in which the initial viscosity profile is not backed-up by natural data. This model becomes unrealistic and is not applicable to the present-day Earth after the rifting phase. Hence, this model does not capture the first-order physics correctly, because Ra-numbers are too high and the lithospheric thickness becomes too thin. We therefore show end-member models that either capture the first order physics of the present-day mantle correctly, or not.

- what do you mean by "If convection in the mantle is suppressed by high effective thermal conductivities or high, lower viscosity limits" (line 453)? During the convergence phases, the mantle still flows in the domain (which is why you should show the velocity glyphs).

A: We want to say that the vigor of convection is significantly reduced, for example, absolute magnitudes of convection velocities are significantly reduced. We modified the text accordingly.

- did you try to run models without shear heating to estimate the relative role of structural vs. thermal softening for localization? (lines460-462)

A: We did not run models without shear heating. (Jaquet & Schmalholz, 2018) and (Kiss, Candioti, Duretz, & Schmalholz, 2020) investigated the importance of shear heating for shear zone formation and subduction initiation. They showed that in absence of any other active weakening mechanism (f.e. brittle-plastic strain weakening) the deactivation of thermal softening results in large scale folding of the crust without localisation of a major shear zone. The geometry at the onset of convergence in the models presented here is similar to the initial geometry of (Kiss, Candioti, Duretz, & Schmalholz, 2020). We, therefore, expect similar behaviour for our model. Since we do not employ any other strain weakening mechanism in the models presented here,

we rely on thermal softening for localisation of shear zones.

- Figure 9 and Figure 2a,b: comment on the "slab-like" features between 100 and 200 km depth below the extended margins in M1 and M2

A: These features result from the convection cells that are already active.

- Fig.8d: what is the X-locations and the depth range for integration of the second invariant of deviatoric stress tensor?

A: The second invariant of the deviatoric stress tensor is integrated vertically over the entire domain. To estimate the plate driving forces we first average the vertically integrated second invariant of the deviatoric stress tensor horizontally over the left most and right most 100 km. Second, the estimated value for the plate driving force is computed as the average of the two average values obtained before. We have clarified this in the revised version of the manuscript.

- explain in caption of Fig. 8 "values for tau_II remain constant when no deformation is applied to the system" whereas Fig. 4 shows large convection cells un the mantle that may deform the plates above

A: We wanted to say: "when no far-field inflow/outflow deformation is applied"; we clarified the text. Values for tau_II are not equal to 0, because there are always deviatoric stresses, f.e. to sustain the margin geometry or shear forces induced by mantle flow at the bottom of the more-or-less rigid plates. However, there is no significant deviation from these "background" stresses when no far-field deformation is applied to the system. Thus, the values for tau_II remain relatively constant during the cooling period.

- appendix B: it is not clear why D should be thickness of the whole upper mantle whereas Fig. 4a,f shows small convection cells

A: This paragraph has been rephrased for clarity during the review process.

- equation B1: how is effective viscosity average over the domain?

A: In our study, we calculate a local Rayleigh-Number on each grid point. The average Rayleigh number is calculated as the arithmetic average of local Rayleigh numbers >1000.

- the explanation on lines 554-561 is not convincing: what take a constant Rayleigh number that on D and on k and then claim that D and k can be adjusted?

A: The goal of this exercise is to match a realistic Nusselt number for the Earth's mantle by assuming a conductive heat flow through the upper mantle. This means that q_LAB is parameterized via an enhanced, conductive heat flux. To match the Nusselt number of the Earth's mantle, this artificial heat flow has to be 13x larger than the realistic conductive heat flow of the mantle. We therefore enhance the thermal conductivity in the upper mantle by a factor 13.

- the isentrop is Fig. A1 does not match the temperature profile in Fig. 1 TECHNICAL CORRECTIONS

A: This has been corrected in the revised version of the manuscript.

- "cooling" is more appropriate than "thermal relaxation" for stage 2

A: This has been changed in the revised version of the manuscript.

- the initial velocity condition is not given

A: This been corrected in Fig. 1 in the revised version of the manuscript

- do you have more references for the "common approach" to indirectly include the effects of thermal convection ? (line 48)

A: They are given line 52.

- line 97-98 : what does "free slip with constant material inflow/outflow velocities" mean? - 7 units of 5 km each make a thickness of 35 km (not 33)(line 104)

A: This mistake has been corrected in the revised version of the manuscript. We have

also clarified the description of the initial configuration.

- line 104 : why describe a 87-km thick mantle lithosphere if all parameters are the same (line 114)

A: To indicate the initial depth of the LAB. We have explained it in more detail during the review process.

- Table 1 : please highlight (bold ?) which parameters differ from model M1 for all models.

A: This suggestion has been implemented in the revised version of the manuscript.

- Table 2 : how are the column of the 2 sediments different? link with pelites/calcites or with sediments 1/2 of Figure 6?

A: This has been changed in the figure legends.

- Table 2 : why no diffusion creep in the crust ?

A: At low temperatures, the strain rate is a nonlinear function of the stress, which suggests that the active deformation mechanism of the crust is likely dominated by dislocation creep. Diffusion creep is usually active at high temperatures and low deviatoric stresses and is therefore more important in the upper mantle.

- Table 2 : which rock are analogue for strong and weak crust?

A: The strength of crustal rocks depends on temperature, pressure and deformation rates. In the models presented here, the strength of the weak and strong layers should be regarded relative to each other. They represent a more heterogeneous crust, which is more realistic than a unified homogenous material for the entire crust. The weak layers represent for example silica-rich metasediments and the strong units represent for example mafic material (see also (Petri, et al., 2019) for more details).

- section 3.1.1: how do you define the length of the margin (threshold in crust thickness?)

A: Thickness reduction from original thickness to <10 km following (Sutra & Manatschal, 2012).

- line 160: is the second invariant tensor of the deviatoric stress calculated for the whole lithosphere including the crust?]

A: Yes, we clarified this in the revised version of the manuscript.

- why do you take the 10ËĘ21 Pa.s contour as the base of the lithosphere? Why not take th 1350◦C isotherm?

A: The viscosity contour remains horizontally straight, whereas the 1350 °C isotherm is deflected by the convection cells indicating mantle material flow rather than a rigid plate boundary. Above the viscosity contour the length of the velocity vectors is essentially zero, but below this contour line the convection cells are active. We have clarified this in the revised version of the manuscript.

- line 176: give X-location of special flow field at 120 km depth

A: This has been clarified in the revised version of the manuscript.

- line 184: why do you claim that the lithosphere is delaminating whereas the isoviscous contour is almost flat?

A: This has been addressed in the revised version of the manuscript.

- line 207: define GPE

A: A definition of GPE is given in line 197. A more detailed explanation is given in the appendix.

- line 223: Figure 8d rather than 8b?

A: Yes, this has been changed in the revised version of the manuscript.

- rephrase "In our models, subduction is initiated self-consistently, without prescribing any major weak zone or an already existing slab." (line 286) since they are weak heterogeneities in the passive margin

A: We argue that in our models, subduction is initiated "self-consistently", because we do not ad-hoc prescribe any major weak zone cutting through the entire lithosphere at the passive margin. Also, in our initial model configuration, we do not impose any major weak zones, or seeds, to force mantle exhumation and separation of the continental crust. The heterogeneities at the passive margin have been modelled self-consistently within the same continuous numerical simulation. Also, the layered heterogeneities are only present in the crust and no major heterogeneities are imposed in the mantle lithosphere. We modified the text to clarify our statement.

- line 319 "if shear stresses are negligible" = is that really the case at subduction onset?

A: Horizontally far away from the subduction zone, which is where we calculate the force, this assumption is valid.

- equation A3: define alpha and beta (which is different from the beta in Eq. B2 I guess...)

A: This has been corrected in the revised version of the manuscript.

- Fig. 1a,d: the depth looks smaller than 680 km

A: Indeed, this is not clear enough in the model configuration: depth is -660 km, surface level is 0 km and +20 km are left free to allow for topography. This has been addressed during the review process.

- Fig. 1c: initial random perturbations look denser between -20 and + 20 km, is that the case?

A: This has also been clarified during the review process.

- Fig. 3d: issue with the bottom of the plit nera -200 km (vertical grey line?)

A: I do not see the vertical grey line. The comment is not clear enough.

- Fig. 5: dashed lines for M4 and M5 are barely visible, I suggest you use thick lines with other colours

A: We have changed the line style and colour for the figure in the revised version of the manuscript.

- Fig. 8: it would be helpful to have label on the topography such as "trench", and to mark the subduction initiation in the timeline of Figure 8d

A: This suggestion has been implemented in the revised version of the manuscript.

- Fig. 10: what is the new information brought by this figure compared to Fig. 4?

A: Figure 10 shows the conductive heat flow of the entire domain. Whenever convection is modelled properly, the conductive heat flow in the upper mantle must be close to 0. In models M4, M5 the conductive heat flow is still high due to the enhanced thermal conductivity. This illustrates, that this approach mimics the convective thermal structure but does not capture the physical process of convection in the upper mantle correctly. We have combined figure 10 with figure 4 in the review process.

Bibliography Bercovici, D., Schubert , G., & Glatzmaier, G. A. (1992). Three-dimensional convection of an infinite-Prandtl-number compressible fluid in a basally heated spherical shell. Journal of Fluid Mechanics, 239, 683-719. Christensen , U. (1995). Effects of phase transitions on mantle convection. Annual Review of Earth and Planetary Sciences, 23(1), 65-87. Gülcher, A. J., Beaussier, S. J., & Gerya, T. V. (2019). On the formation of oceanic detachment faults and their influence on intra-oceanic subduction initiation: 3D thermomechanical modeling. Earth and Planetary Science Letters, 506, 195-208. Jaquet, Y., & Schmalholz, S. M. (2018). Spontaneous ductile crustal shear zone formation by thermal softening and related stress, temperature and strain rate evolution. Tectonophysics, 746, 384-397. Kiss, D., Candioti, L. G., Duretz, T., & Schmalholz, S. M. (2020). Thermal softening induced subduction initiation

at a passive margin. Geophysical Journal International, 220(3), 2068-2073. Petri, B., Duretz, T., Mohn, G., Schmalholz, S. M., Karner, G. D., & Müntener, O. (2019). Thinning mechanisms of heterogeneous continental lithosphere. Earth and Planetary Science Letters, 512, 147-162. Ricard, Y. (2007). Physics of mantle convection. Treatise on Geophysics, 31-88. Sutra, E., & Manatschal, G. (2012). How does the continental crust thin in a hyperextended rifted margin? Insights from the Iberia margin. Geology, 40(2), 139-142. Tosi, N., Stein, C., Noack, L., Hüttig, C., Maierova, P., Samuel, H., . . . Glerum, A. (2015). A community benchmark for viscoplastic thermal convection in a 2‐D square box. Geochemistry, Geophysics, Geosystems, 16(7), 2175-2196. Turcotte, D. L., & Schubert, G. (2014). Geodynamics. Cambridge Unversity Press. van Hunen, J., van den Berg, A. P., & Vlaar, N. J. (2001). Latent heat effects of the major mantle phase transitions on low-angle subduction. Earth and Planetary Science Letters, 190((3-4)), 125-135.

---

## Author Comment (AC2) · 4 Aug 2020

We gratefully thank the reviewer for the very constructive criticism. Implementing the suggestions helped to better visualize the simulation results, focus on the main findings and significantly improve the manuscript. During the review process, we have changed the structure of the manuscript significantly. In the results section of the revised manuscript, we present the evolution of the reference model and the wet olivine model separately. The results of the remaining models are presented in comparison to the results of the reference run for the distinct deformation stages. We then discuss the implications of our findings on several aspects, such as for example the impact of

the viscosity structure on the convection, the onset of convection and the impact of convection on subduction, in the discussion section. The order of the figures and the style of visualization has been adapted accordingly.

Below we have answered to all the comments from the reviewer. Our answers marked with an "A:" and are below the original comments.

Major points: 1. Clarity of the manuscript. The model is very complex and it has lots of details, but the authors haven't always explained the concepts clearly or properly. I have written down some specific examples that the authors can fix easily. However, they should try to verify that their findings are backed by arguments that are explained in a logical way. The abstract should be shortened to include the top 3 most important results, and be revised for clarity and shorter sentences. For example, what do the authors want the paper be known/cited for?

A: The Abstract has been shortened and reformulated during the review process in order to address the reviewer's comment.

In the introduction, the link between mantle convection and lithosphere deformation is quite abrupt (with a sentence about age of the Earth that is irrelevant to this study). The question 'why is convection important?' is not satisfactorily introduced or linked to coupled lithosphere-mantle deformation.

A: Convection regulates the long-term temperature and mechanical structure of the lithosphere: the strength of the lithosphere is inter alia temperature dependent. Thus, convection may have a direct impact on the deformation of the lithosphere. Coupling convection to lithosphere deformation in numerical models can therefore improve our understanding of lithospheric scale processes, such as rifting and subduction. Also, convection can generate forces due to up- and down-welling of mantle material, which can affect lithosphere deformation. A paragraph has been added to the introduction for better explanation.
General suggestion: too many commas. Try to rephrase/split sentences with more than 2 commas or that are longer than 2 lines.

A: This has been implemented in the revised version of the manuscript.

2. Results section. I think the reference model (M1) should be described separately (evolution between extension, relation, convergence). Then compare models M2-M6 with M1 to highlight the effect of various factors. Figures should be adapted accordingly. The reason for this are the following: - in current form, the comparison is all over the place and it is confusing. It is not very clear which simulations the main text is referring to sometimes. - the current arrangement of figures is random. It starts with 2, 5, 8, 4, 10 etc. Their placement should follow a logical order of arguments.

A: We have changed the structure of the manuscript as follows: (1) Results section: (i) The evolution of the reference run and M6 is now described separately (two standalone figures) (ii) M2-5 are then compared to M1 at each deformation stage. (iii) Figures have been modified and reordered accordingly (2) Discussion section: We have restructured the discussion section and are now discussing implications of the models for several geodynamic problems including: (i) Spontaneous vs. Induced subduction initiation (ii) Mantle convection stabilising single-slab subduction (iii) Onset of upper mantle convection and thermo-mechanical evolution of the lithospheric plates (iv) Impact of mantle viscosity structure and effective conductivity on passive margin formation

The comparison between M1-M6 should be done in terms of Ra. The k, viscosity cutoff, flow laws, they essentially affect the Ra.

A: We have largely implemented this suggestion in the revised version of the manuscript.

3. Thermal softening. A quick search in the manuscript finds 'thermal softening' only in the abstract, very late discussion and conclusion, yet it is suggested as a key process that controls subduction initiation. I'm pointing out that it is incompletely described

SED
and linked to the hypothesis of the study and results. For example, Line 425: thermal softening is introduced only now. not clear why bring it up here? Line 461: say that structural and thermal softening are important, but they were intro- duced late, without much context. Moreover, the authors suggest in multiple places that it is the structural softening (in- heritance) and convection (slab suction) that help initiation. The authors need to clarify what are the main findings, and arguments need to be revised. One finding that I think is important: the required driving force to initiate subduction is much smaller, when convection and structural inheritance are considered.

A: We have discussed thermal and structural softening in more detail in the revised version of the manuscript.

4. Modelled vs parameterised convection. In Line 126, 3 types of simulations are introduced: 1) model convection with a weak asthenosphere, 2) parametrised convection, by scaling the thermal conductivity to the Nusselt number 3) impact of different viscosity structures First, the treatment of the mantle convection is not clear in the main text (Lines 124- 130). What drives convection? How is the applied parametrised convection different? When is the onset of convection? Is convection only during the thermal relaxation stage? What controls the size of the convection cells? Also, explain how the Ra\_avg is calculated.

A: This has been addressed during the review process. The Rayleigh number is calculated locally at each grid point. Ra\_avg is the arithmetic average of all local Rayleigh numbers > 1000. We have clarified in the revised version of the manuscript.

In point 3) above which approach are you using: modelled/parameterized convection? While it is explained better in Appendix B, the differences between them are not clear in the main text. For example, 1) would be M1, while 2) is M, and 3) is M6?

A: This was explained in lines 130-135, but we changed the numbering of the models in the revised version of the manuscript, for clarity.
5. Other questions. The geodynamic cycle modelled: 1) 30 Myrs extension at 2cm/yr 2) 70 Myrs thermal relaxation 3) 20 Myrs convergence at 3 cm/yr What is the motivation behind these choices: 1) why thermal relaxation 2) why those time intervals 3) why those extension/convergence rates? Also, what are the boundary conditions during thermal relaxation?

A: The aim is to model the opening of a ca. 400 km wide oceanic basin without formation of a mature oceanic crust in an ultra-slow to slow spreading rift system. The durations of the periods and boundary conditions are chosen to allow for comparison of model results to orogens that formed from the collision of magma-poor hyper-extended margins, such as the European Western and Central Alps. We have clarified this in the revised version of the manuscript.

Why the choice of those parameters to change?

A: The viscosity structure of the mantle is poorly constrained and has a direct impact on the convective flow of the mantle. The effective conductivity approach is used to stabilise the thermal field in numerical simulations, but its impact on the deformation of the rigid plates and on self-consistent subduction initiation has not been tested yet. These statements have been added as a motivation in the introduction.

Are the surface processes important? Have you run models without? Do they introduce further heterogeneities in the model that affect the outcome?

A: Testing in more detail coupled surface processes to the deformation in the lithosphere and convection in the upper mantle is beyond the scope of this study. We have included only a simple parameterisation of surface process into the model to avoid unrealistically high and low topography. This has been clarified in the revised version of the manuscript.

6. Subduction initiation. It seems like symmetric vs asymmetric spreading also controls to a large extent subduction initiation, whether it is single/double subduction. I feel very
little discussion is about that, and more on structural and thermal softening.

A: Subduction is always initiated during convergence. Most inheritance from the spreading is restricted to the margin geometry and is thus structural. Localisation occurs in the lithospheric mantle beneath the margins due stress concentration at the beginning of the necking zone. The heterogeneity in the upper mantle introduced by the convection and, therefore, the suction force of downward directed material flow seems to influence whether there is single or double-sided subduction. This has been clarified during the review process.

Also, there are other previous efforts to model extension/compression to obtain structural inheritance and subduction initiation (i.e. Gulcher et al 2019). The authors discuss simpler treatments of subduction initiation in paragraph 280, but do not relate to newer efforts to avoid the use of artificial features. So, are these newer models better for studying subduction initiation?

A: (Gülcher, Beaussier, & Gerya, 2019) investigate detachment faults as potential weak zones for intra-oceanic subduction initiation, not at a passive margin as wee do. Nevertheless, such detachment faults might be a geologically observable weak zone at which subduction could be potentially initiated. This topic has been addressed in the introduction of the revised manuscript.

Minor points: Line 8-10: revise sentence

A: We have incorporated this suggestion in the revised version of the manuscript.

Line 10: only from the abstract it is not clear what the parameters were used, so saying that a viscosity of 5e20 Pa.s was used (as compared to what?) is not very meaningful. Rephrase

A: We have rephrased the abstract in the revised version of the manuscript

Line 20: multiple use of 'geodynamic' in the same sentence Line 29-30: rephrase.

**SED**
A: We have rephrased this sentence in the revised version of the manuscript.

Line 31: while it is an interesting fact - the calculation of the age of the earth - is not very relevant to the manuscript.

A: We have deleted this sentence in the revised version of the manuscript.

Line 35: rephrase

A: We have rephrased this sentence in the revised version of the manuscript.

Line 40: unlikely to be problematic

A: We have changed this accordingly in the revised version of the manuscript.

Line 41: delete likely

A: We have changed this accordingly in the revised version of the manuscript.

Line 55: authors relate to numerical aspects such as time step size, without mentioning why? The context was on physical aspects of convection.

A: This has been moved to the appendix in the revised version of the manuscript.

Line 64-65: should be in the first paragraph of introduction

A: This suggestion has been implemented in the revised version of the manuscript.

Line 68: Why only upper mantle? This is discussed late in discussion (section 4.4, paragraph 395)

A: This has been moved to the introduction section in the revised version of the manuscript.

Line 69: delete 'of applying'

A: This has been changed in the revised version of the manuscript.

Line 74: revise sentence - its meaning is not clear to someone who hasn't read the
methods/results section.

A: We have restructured this part of the introduction in the revised version of the manuscript.

Line 88: reference to the code how it was benchmarked? (Info in appendix A, but should be in the main text too)

A: The benchmarks have been moved to the main text in the revised version of the manuscript.

Line 89: rephrase

A: We have rephrased this sentence in the revised version of the manuscript.

Line 96: repeats with Line 92, also Duretz et al 2016/2016a?

A: This has been changed in the revised version of the manuscript.

Line 105: based on the sentence the crust should be: 3\*5+4\*5 = 35 km thick. But a sentence earlier it is 33km

A: This mistake has been corrected and the description of the initial configuration has been clarified in the revised version of the manuscript.

Line 103: what is the mathematical expression for the perturbation? in case the model needs to be reproduced?

A: We have added the mathematical expression for the marker field perturbation to the algorithm description in the appendix of the revised version of the manuscript.

Line 111: more details on the rheology? Indicate appendix A for reference

A: We refer to the appendix in the revised version of the manuscript.

Line 113: reference to "corresponding laboratory flow law estimates"?

A: References are given in the footnote of table A1.
Line 113-114: rephrase. i.e. The mantle lithosphere is rheologically stronger than the mantle asthenosphere due to the temperature gradient.

A: With this sentence, we want to emphasize that we used the same material parameters for both the lithospheric and the upper mantle. We therefore keep the phrase as it is.

Line 120: what is the motivation for alternating between calcites and pelites for sedimentation algorithm?

A: To account for changes in sediment strength due to changing sedimentary environments. A detailed investigation on the impact of different implementations of surface processes is beyond the scope of this study. We have clarified this in the revised version of the manuscript.

Line 130 - give reference to Table 1.

A: This has been adapted in the revised version of the manuscript.

Line 133: viscosity cutoff for M1 is not provided to understand the difference.

A: The cut-off value is given in table 1.

Line 134: realistic value? Are the other values not realistic?

A: A thermal conductivity value of 36 is not realistic for a peridotite at upper mantle temperature and pressure conditions.

Line 148: Figure 2 -> can define a variable F = 2xtau\_II

A: We have introduced a variable for the plate driving forces in the revised version of the manuscript.

Line 160: introduce the horizontal driving force per unit length, but what is it proxy for?

A: It is a proxy for the strength of the lithosphere and it indicates how much force is needed to localise deformation. Has been clarified during the review process.

Interactive

comment

Line 168: you can't see to a depth of 660 km as indicated

A: Figures have been adapted accordingly.

Line 195: values

A: This sentence has been rephrased during the review process.

Line 197: what is the delta GPE showing? (Info given in appendix)

A: The gravitational potential energy (GPE) has been explained in the appendix. Delta means that it is the difference of the GPE compared to a reference value, commonly the value close to one of the boundaries.

Line 224: reference to fig 9a, yet that figure is for extension stage. Paragraph 220-225: confusing.

A: This has been addressed during the review process.

Line 228: what is mechanical heterogeneity? increases the strength of the weak layers

A: We have clarified this in the revised version of the manuscript.

Line 233: breaks later than the continental. after gives the impression of location.

A: This suggestion has been implemented into the revised version of the manuscript.

Line 234: what do you mean ' Mantle convection does not establish as early as rifting and crustal separation.'?

A: This sentence has been clarified in the revised version of the manuscript.

Lines 243, 245: use of realistic. close to the Ra estimated for the Earth. Line 250: which modulates mantle velocities.

A: We have implemented this suggestion in the revised version of the manuscript.

Line 250-254: why the discussion on time step size (a numerical feature) here? Lines
255-257: which simulation results are the authors referring here?

A: This has been addressed in the revised version of the manuscript

Paragraph 258: reference figure 5e,j in this paragraph. Also, maybe plot density averages in passive margins/exhumed mantle separately?

A: We have tried to modify this figure, but the figure becomes too busy.

Line 272: you jump from density differences to values of tectonic forces. An additional sentence needs to connect them (i.e. estimate the buoyancy force due to modelled density differences). How much is needed to initiate subduction? (a similar calculation is done in line 315)

A: This depends on the strength of the lithosphere, which is still subject to debate.

Paragraph 280: this should come before the Cloos 1993 paragraph

A: We have restructured the discussion accordingly during the review process.

Line 286: yes, but under convergence

A: When a major weak zone is imposed, one also has to push from the sides to initiated subduction. Only when you skip the process of subduction initiation and already assume the presence of an inclined slab, subduction continues freely (given that the initial slab is long enough and that boundary conditions have been chosen correctly).

Line 294-295: total convergence is double sided, while in M1 is single-sided (asymmetric). Not clear why subduction initiation is stable only in M1. Convection cell size important? how about thickness of lithosphere at the point? M2-M5 are quite symmetric and they all have Ra\_avg âLij1e5, while M1 has Ra\_avgâLij1e6. That should have an effect.

A: Likely the distribution of cells is important: the more asymmetric the mantle flow the more the model tends to produce single-slab subduction rather than double-slab
subduction. Asymmetry decreases with decreasing Ra. This has been explained in more detail during the review process.

Line 312-315: - suction force induced by down-welling in the convection cell in M1. What is the similar force in the other simulations?

A: It is likely similar in M2-3 compared to M1, because of similar density distributions and vertical velocities, but it has not been computed here. Due to higher temperatures induced by the effective conductivity, values for densities are lower in M4-5 compared to M1-3, which explains the reduced absolute speed of material in the convecting cells. Most important seems to be the asymmetric distribution of the cells. We have clarified this during the review process.

Line 330: not sure what the reference is for. The double-subduction term was not coined by those workers.

A: We cited this reference, because they also modelled double-sided subduction.

Line 332: sentence not clear. Which simulation are you referring? would say M2-5 are more or less symmetric double subduction

A: Has been reformulated during the review process.

Line 343: onset of convergence - unclear when this happens?

A: This happens at 100 Myrs. We have clarified this in the revised version of the manuscript.

Line 357-361: use of 'realistic'

A: This has been rephrased in the revised version of the manuscript.

Paragraph 395: this paragraph should be in the methods, as it motivates/explains your model domain until 660 km. The sentence 'The convective patterns simulated in our study are in agreement with these observations.' is irrelevant because you don't model

SED
the lower mantle.

A: This paragraph has been moved to the introduction in the revised version of the manuscript.

Paragraph 410: this should come earlier - I had questions about it earlier. on previous work on subduction initiation.

A: This paragraph has been moved to the introduction in the revised version of the manuscript.

Line 418: most definitely will have an impact

A: We have implemented this suggestion in the revised version of the manuscript.

Paragraph 430: and melting

A: We have accounted for this suggestion in the revised version of the manuscript.

Line 444-446: rephrase/simplify.

A: We have rephrased the conclusion section in the revised version of the manuscript.

Figures and Tables: Table 1: thermal conductivity should be 'k' without the 'th' subscript. The authors can also provide the formula for the Ra number in the main text. How was the Ra\_avg calculated?

A: This suggestion has been implemented in the revised version of the manuscript.

Table 2: there should be a column 'Description' to describe the meaning of each parameter i.e. 'rho0' - reference density. Use k instead of k\_th for thermal conductivity. What is dry/wet mantle? I assume wet mantle applies only to M6? Plastic and elastic parameters are also listed. Not very clear in the main text.

A: Description of the parameters is given in the appendix section. We also moved the table to the appendix for better comprehension. Dry and wet refer to the rheological parameters for dry and wet olivine. In the references given below the (now) appendix
table 1 it is explained in which model we use either wet or dry rheologies.

Figure 4: why plot the vertical velocity field separate from the horizontal? should plot arrow/streamlines field to see the convection cells.

A: The figure has been adapted accordingly in the revised version of the manuscript.

Figure 10: should be merged with Figure 4. One column velocity, one column temperature.

A: These two figures have been merged in the revised version of the manuscript.

Figure 5: What if you plot the profiles at the rift axis (within a distance) vs off-axis on either flanks of the rifts? Caption: g-j show enlarged areas.

A: We have tested this version of the figure, but it becomes too busy, unfortunately.

Figure 6: legend: temp contours are red.

A: We have corrected this mistake in the revised version of the manuscript.

Figure 8: the line plots are not entirely clear. Maybe use a dotted line instead of dashed line? and same thickness.

A: We have changed line style and colour for the figures in the revised version of the manuscript.

Appendix A Line 481: that's a strange notation of i,j indices (Einstein notation). Eq A2: if written in Einstein notation, then vectors are written in terms of scalar components (a\_i should not be bold). Same in Line 482 a=[0,g]. -> revise this appendix for completeness of sentences, and explanation of all parameters. For example, what is Ap, tauP etc. Gamma value? in eq A10

A: We have clarified these points in the revised version of the manuscript.

Appendix B Paragraph 531: rephrase

SED
A: This suggestion has been implemented in the revised version of the manuscript.

Line 560: not clear

A: This has been clarified in the revised version of the manuscript.

Appendix C Line 595: gamma\_T=1? Line 601: g=10ËĘ4?

A: To perform this benchmark one has to apply a local Rayleigh number of 100 at the top and using the Frank-Kamenetskii approximation a local Rayleigh number of 107 at the bottom. Choosing the dimensionless values as it is done here matches those numbers and reproduces the desired pattern with diagnostic quantities that are in the range of values reproduced by other algorithms as tested in (Tosi, et al., 2015).

Bibliography Gülcher, A. J., Beaussier, S. J., & Gerya, T. V. (2019). On the formation of oceanic detachment faults and their influence on intra-oceanic subduction initiation: 3D thermomechanical modeling. Earth and Planetary Science Letters, 506, 195-208. Tosi, N., Stein, C., Noack, L., Hüttig, C., Maierova, P., Samuel, H., . . . Glerum, A. (2015). A community benchmark for viscoplastic thermal convection in a 2-D square box. Geochemistry, Geophysics, Geosystems, 16(7), 2175-2196.

---

## Author Response (AR1)

We gratefully thank the reviewer for the very constructive criticism. Implementing the suggestions helped to better visualize the simulation results, focus on the main findings and significantly improve the manuscript. During the review process, we have changed the structure of the manuscript significantly. In the results section of the revised manuscript, we present the evolution of the reference model and the wet olivine model separately. The results of the remaining models are presented in comparison to the results of the reference run for the distinct deformation stages. We then discuss the implications of our findings on several aspects, such as for example the impact of the viscosity structure on the convection, the onset of convection and the impact of convection on subduction, in the discussion section. The order of the figures and the style of visualization has been adapted accordingly.

Below we have answered to all the comments from the reviewer. Our answers are highlighted in blue and the original comments are highlighted in red.

- the fluid material is described as incompressible in section 2.1, but the thermal gradient is adiabatic, mantle densities vary greatly along the domain high due to phase transitions (Fig. A1) and the authors use the extended Boussinesq approximation for compressible fluid to solve the conservation equations (Appendix B line 579)

> The maximum value for the density time derivative is two orders of magnitude smaller compared to the velocity divergence. Also, (Bercovici, Schubert , & Glatzmaier, 1992) concluded that compressibility effects on the spatial mantle structure are minor when the superadiabatic temperature drop is close to the adiabatic temperature of the mantle, which is the case for the Earth. We therefore assume here that the Boussinesq approximation is still valid and suggest that density changes due to volumetric deformation are negligible. We consider only small density changes affecting the buoyancy stresses. Not considering adiabatic heating in the energy conservation equation leads to a significant deviation of the thermal structure from the initially imposed adiabatic temperature gradient over large time scales (>100 Myrs). The resulting vertical temperature profile (if adiabatic heating is neglected) is constant throughout the upper mantle and the newly equilibrated vertically-constant temperature is equal to the imposed temperature at the bottom boundary. In consequence, the density structure taken from the phase diagram table according to pressure and temperature values is wrong. By using the extended Boussinesq approximation, i.e., the adiabatic heating term is included in the energy conservation equation but not in the continuity equation, the initially imposed adiabatic (or isentropic) gradient is maintained over long time scales. The resulting density structure agrees well with the PREM model as shown in this study. A more detailed comparison between different approximations of the continuity equation is beyond the scope of this study.

- phase transitions are implemented through the variable density (Fig. A1), but latent heat associated to phase change is missing from equation (A14), and this approximation should be justified.

Indeed, latent heat released or consumed by a phase transition can perturb the thermal field by up to 100 K and induce a buoyancy force aiding or inhibiting the motion of cold subducting slabs (van Hunen, van den Berg, & Vlaar, 2001) or hot rising plumes. However, when the lateral differences in temperature do not vary much, the deflection of the phase transition by an ascending plume or a subducting slab has a much bigger impact on the buoyancy stresses than the latent heat released or consumed by the phase transition (Christensen , 1995). Also, most of the studies on the impact of latent heat rely on the assumption that density is temperature dependent only. In the models presented here, density is a function of temperature and pressure, which makes it difficult to estimate the impact of temperature changes due to latent heat near a phase transition on the buoyancy term a priori. Because a detailed parametric investigation of the impact of latent heat on buoyancy stresses for temperature and pressure dependent density is beyond the scope of study, we neglect latent heat for simplicity.

- the side velocity boundary conditions during the extension or convergence phase (Fig. 1a,d) are likely to induce a sheared weak zone near the side boundaries at the transition depth between lateral inflow and outflow (340 km depth). Also, for the extension set-up, the suction created by the divergence in uppermost mantle is likely to generate a bulk ascending mantle channel in the middle of the domain (X=0 km).
- more generally, the flow pattern over the entire model domain is never shown when in/out flows are imposed at the sides, and this is an issue when discussing application to Earth (section 4.4): what is the geological justification that the divergence or convergence was not only active at the plates 'surface but also across 300 km in the mantle below the plates? The justification of the side velocity boundary conditions should be developed, and the global flow pattern (velocity glyphs/arrow) should be shown during extension and convergence phases.
I also have other issues with the methods and results analysis:

As mentioned correctly by the reviewer, material inflow/outflow velocity boundary conditions may lead to a shear zone forming at the transition point between inflow and outflow. To avoid such boundary condition effects close to the mechanical lithosphere, we set the transition point deeper than the initially imposed lithospheric thickness. We chose z=-330 km as the point of transition, because values for deviatoric stresses at this depth are significantly smaller compared to those at the base of the lithosphere. The mechanical thickness of the lithosphere may vary greatly over time mainly due to temperature changes and is therefore difficult to constrain a priori. By setting the transition point of the velocity inflow/outflow boundary condition deeper than the initially imposed mechanical thickness of the lithosphere, we allow for self-consistent adjustment of the mechanical thickness of the lithosphere during the evolution of the model. We have added velocity arrows to many of our revised figures. These figures show that the thickness of the lithosphere, with respect to consistent horizontal velocities, is not controlled by the inflow/outflow boundary away from the model boundaries.

- the choice of which input parameter are varied is not explained: why choose to vary the minimum viscosity between models M1, M3 and M5, rather than for example the initial extension rate, the duration of the thermal relaxation phase or the inflow/outflow side velocity profiles? The discussion does not well explain why there is a single subduction in M1, but double subductions in M3/M5.

> One aim of this study is to quantify the impact of convection in the upper mantle on the long-term extension—cooling—convergence cycle of the lithosphere. Convection is controlled by the Rayleigh-number, which is a function of the viscosity of the convecting layer. Since the viscosity of the mantle is poorly constrained and estimated values vary by two orders of magnitude, this parameter is the most interesting for us to investigate. A frequently used technique to parameterize the impact of convection is the effective conductivity approach. This way the temperature field can be stabilized without having to calculate an enhanced convective velocity field in the mantle (which can be numerically challenging). We therefore compare this approach to the explicitly modelled convection, which calculates an enhanced convective velocity field in the mantle. The initial extension rate has been chosen to model the formation of an approximately 400 km wide basin containing exhumed mantle in an ultra-slow to slow spreading environment. The timing of the extension, cooling and convergence phases are motivated by data from the Alpine orogeny, which is now clearer explained in the revised manuscript. Testing the impact of different extension and compression rates as well as different cooling durations is beyond the scope of this study.

- a methodology study on the comparison of "explicitly modelled convection" and "effective conductivity mimicking a convective heat flow" is inserted in the middle of the main geodynamics study. This hinders the continuous read of the paper, and I suggest all analysis and related figures of models M2/M4 are moved to Appendix B, along with the heat flow profiles of Figure 10.

> The manuscript and result presentation have been re-structured based on the constructive comments of both reviewers. However, we keep the results of the simulations with an "effective conductivity" in the main manuscript, because these results are part of our main geodynamics study.

- the simulations have numerous features that do not seem relevant for the scientific question (erosion, sedimentations of alternating calcites and pelites), but add yet another set of free parameters that make the interpretation of the simulations more complex.

> We include erosion and sedimentation to avoid too high or low topography. To achieve this, we decided on one particular erosion/sedimentation model. A parametric study on the impact of different sedimentation/erosion processes or

sediment transport mechanisms on the deformation of the lithosphere is beyond the scope of this study.

- despite its central importance, the paper lacks a clear definition of "convection", that sometimes means "advection" or "drag" or "flow".

The word "convection" can be used to describe the motion of any fluid. Convection in a fluid can either develop freely, via thermal or compositional variations, or it can be induced by external forces (Ricard, 2007). In mantle convection simulations, free material motion in the mantle can be initiated by for example buoyancy contrasts due to variations in temperature or chemical composition. A geological example for induced mantle flow is a rigid plate that moves on top of the mantle. These statements have been incorporated to the introduction during the review process.

The manuscript could also maybe reference the following papers dealing with the plate-asthenosphere interactions or subduction initiation or various scales of mantle convection:
L. Husson. The dynamics of plate boundaries over a convecting mantle. Physics of the Earth and Planetary Interiors, Elsevier, 2012, 212-213, pp.32-43.
V. S. Solomatov. Initiation of subduction by small-scale convection. Journal of Geophysical Research: Solid Earth, 2004.
F. Lévy, C. Jaupart. The initiation of subduction by crustal extension at a continental margin. Geophysical Journal International, Volume 188, Issue 3, March 2012, Pages 779–797.
N. Coltice et al. Interactions of scales of convection in the Earth's mantle. Tectono-physics. Volume 746, 30 October 2018, Pages 669-677

We have added some of the references to the revised version of the manuscript.

Presentation quality: poor

This is a major flaw of the manuscript, which scientific contributions are hard to unearth because of confusing text and figure organization.
For example :
- showing vertical and horizontal velocity background colors (Fig. 4, Fig. 9) makes it difficult for the reader to visualize the flow pattern > could the authors show velocity glyphs or arrows, to better reveal e.g. the wavelength of small-scale convection in Fig. 4

We have added the velocity arrows to the figures. This was a very constructive comment for better visualization – thank you.

- Figure 5 should be referenced in the methods section since achieving realistic temperature, density and viscosity model output is rather a constrain on the input parameters than a surprising result

This has been done during the review process.

- Figure 8 and 10 are referred to very early in the text, whereas they belong in the discussion (or appendix?) rather than in the results section same for text on lines 187-199,156-164

The figure order has been changed in the review process.

- Appendix A belongs to the main text, otherwise the parameters of Table 2 are not defined

We have decided to shift the parameter table 2 to the appendix rather than describing all equations in the main text.

- a time-bar could be included in Fig. 1 showing to scale the 3 stages of boundary conditions with colours corresponding to the velocity profiles shown in Figure 1a,d (also please add a null-velocity profile fo the thermal relaxation). The same time bar could then be put on other figure to know at a glance which stage the figures belong to.

We have tried to add the time bar, but the figures become too busy. We have changed the order of the figures and implemented much of the constructive comments.

SPECIFIC COMMENTS
- in the introduction, the authors should define what they mean by "convection" and discuss the different scales

This suggestion has been implemented in the revised version of the manuscript.

- the exhumation of hot mantle (Fig. 2) is expected to lead to melting, please comment

This is explained in the introduction line 70 ff. and also in the methods section.

- you need to support some statements with results/data, i.e. "Alternating activity of the subduction zones is observed." (line 219)

This has been done in the review process.

- if you mention the importance of apply a force rather than a velocity BC, then you should also mention the importance of setting the lateral flow in/out of the domain

The timing of the deformation periods is chosen to allow for comparison with orogenies such as, f.e. the Alps. By choosing inflow/outflow we simulate the movement of the plates, decoupled from the rest of the domain, which we

consider here more realistic than extending or compressing the entire side walls of the model domain with a height of 660 km.

- I am not sure you can compare your model to the Atlantic (line 272) since old oceanic lithosphere there is much older than in your models

… Even that old oceanic crust did not undergo spontaneous subduction yet, which was the point to mention it as an example here.

- I disagree with the statement "the models are in a state of isostatic equilibrium at the onset of subduction initiation." (line 278): the convergence velocity and the topographic low above the new trench (Fig. 8) suggest a dynamic topography

Indeed, this was our mistake: we chose the wrong time step to show the topography at the end of the cooling period. This has been updated during the revision. In general, it is also true that the system cannot be in isostatic equilibrium by definition, since there are deviatoric stresses holding the topography of the passive margins. However, the difference between the height of margins and the depth of the basin is ca. 5 km, which is the calculated topographic difference for an idealised block of 30 km thick crust floating on top of the mantle (Turcotte & Schubert, 2014). We therefore argue that the topography across the passive margin system produced by our model is close to isostatic equilibrium at the end of the cooling period.

- line 294-307: the explanation of the control of single-sided subduction is not clear

The down-welling of two convecting cells meet directly below the margin at which subduction is going to be initiated later. The enhanced downward motion of upper-mantle material in this region likely exerts a suction force that assists in initiating and stabilising only one single-slab subduction.

- you cannot claim that "mantle convection seems active and largely confined to the upper region of the upper mantle. The convective patterns simulated in our study are in agreement with these observations." (line 399) since you impose the height of your simulation domain to be restricted to the upper mantle.

We want to highlight here that we model a convective pattern that is observed in nature, namely convection in the upper mantle, that is below the lithosphere and above 660 km. Other people might argue that such "upper-mantle convection" does not exist and only "one-layered convection" of the entire mantle, down to ca. 2900 km, exists in nature. We confined our model domain using the justification that in the Alps the convection seems to be two-layered. Besides that, all models are confined to the upper mantle, but the reach of the convection cells extends to different depth, depending on the Rayleigh-number (compare M1 to M3). This is now more obvious when visualising the flow pattern with velocity vectors.

- you should not boast that "the model has captured correctly the first order physics of the investigated processes." since model M6 shows the immense importance of rheology parameterisation - that is far from being constrained...

As mentioned, we calibrated our initial model configurations in such a way that they match data from the PREM model and from GIA estimates. Starting from this point, we let the model freely evolve and do not change material parameters or geometries anymore. During rifting we generate margins of realistic first-order geometry, during cooling, the basin subsides to realistic depths and convection has realistic Rayleigh numbers, and finally subduction is initiated self-consistently (i.e. without imposing any major triangular weak zone cutting through the entire lithosphere at the OCT) via thermal softening. Therefore, we argue that model M1 captures the first order physics of the extension-cooling-subduction cycle correctly. M6 shows a scenario in which the initial viscosity profile is not backed-up by natural data. This model becomes unrealistic and is not applicable to the present-day Earth after the rifting phase. Hence, this model does not capture the first-order physics correctly, because Ra-numbers are too high and the lithospheric thickness becomes too thin. We therefore show end-member models that either capture the first order physics of the present-day mantle correctly, or not.

- what do you mean by "If convection in the mantle is suppressed by high effective thermal conductivities or high, lower viscosity limits" (line 453)? During the convergence phases, the mantle still flows in the domain (which is why you should show the velocity glyphs).

We want to say that the vigor of convection is significantly reduced, for example, absolute magnitudes of convection velocities are significantly reduced. We modified the text accordingly.

- did you try to run models without shear heating to estimate the relative role of structural vs. thermal softening for localization? (lines460-462)

We did not run models without shear heating. (Jaquet & Schmalholz, 2018) and (Kiss, Candioti, Duretz, & Schmalholz, 2020) investigated the importance of shear heating for shear zone formation and subduction initiation. They showed that in absence of any other active weakening mechanism (f.e. brittle-plastic strain weakening) the deactivation of thermal softening results in large scale folding of the crust without localisation of a major shear zone. The geometry at the onset of convergence in the models presented here is similar to the initial geometry of (Kiss, Candioti, Duretz, & Schmalholz, 2020). We, therefore, expect similar behaviour for our model. Since we do not employ any other strain weakening mechanism in the models presented here, we rely on thermal softening for localisation of shear zones.

- Figure 9 and Figure 2a,b: comment on the "slab-like" features between 100 and 200 km depth below the extended margins in M1 and M2

    These features result from the convection cells that are already active.

- Fig.8d: what is the X-locations and the depth range for integration of the second invariant of deviatoric stress tensor?

    The second invariant of the deviatoric stress tensor is integrated vertically over the entire domain. To estimate the plate driving forces we first average the vertically integrated second invariant of the deviatoric stress tensor horizontally over the left most and right most 100 km. Second, the estimated value for the plate driving force is computed as the average of the two average values obtained before. We have clarified this in the revised version of the manuscript.

- explain in caption of Fig. 8 "values for tau_II remain constant when no deformation is applied to the system" whereas Fig. 4 shows large convection cells un the mantle that may deform the plates above

    We wanted to say: "when no far-field inflow/outflow deformation is applied"; we clarified the text. Values for tau_II are not equal to 0, because there are always deviatoric stresses, f.e. to sustain the margin geometry or shear forces induced by mantle flow at the bottom of the more-or-less rigid plates. However, there is no significant deviation from these" background" stresses when no far-field deformation is applied to the system. Thus, the values for tau_II remain relatively constant during the cooling period.

- appendix B: it is not clear why D should be thickness of the whole upper mantle whereas Fig. 4a,f shows small convection cells

    This paragraph has been rephrased for clarity during the review process.

- equation B1: how is effective viscosity average over the domain?

    In our study, we calculate a local Rayleigh-Number on each grid point. The average Rayleigh number is calculated as the arithmetic average of local Rayleigh numbers >1000.

- the explanation on lines 554-561 is not convincing: what take a constant Rayleigh number that on D and on k and then claim that D and k can be adjusted?

    The goal of this exercise is to match a realistic Nusselt number for the Earth's mantle by assuming a conductive heat flow through the upper mantle. This means that q_LAB is parameterized via an enhanced, conductive heat flux. To match the Nusselt number of the Earth's mantle, this artificial heat flow has to be

13x larger than the realistic conductive heat flow of the mantle. We therefore enhance the thermal conductivity in the upper mantle by a factor 13.

- the isentrop is Fig. A1 does not match the temperature profile in Fig. 1 TECHNICAL CORRECTIONS

This has been corrected in the revised version of the manuscript.

- "cooling" is more appropriate than "thermal relaxation" for stage 2

This has been changed in the revised version of the manuscript.

- the initial velocity condition is not given

This been corrected in Fig. 1 in the revised version of the manuscript

- do you have more references for the "common approach" to indirectly include the effects of thermal convection ? (line 48)

They are given line 52.

- line 97-98 : what does "free slip with constant material inflow/outflow velocities" mean?
- 7 units of 5 km each make a thickness of 35 km (not 33)(line 104)

This mistake has been corrected in the revised version of the manuscript. We have also clarified the description of the initial configuration.

- line 104 : why describe a 87-km thick mantle lithosphere if all parameters are the same (line 114)

To indicate the initial depth of the LAB. We have explained it in more detail during the review process.

- Table 1 : please highlight (bold ?) which parameters differ from model M1 for all models.

This suggestion has been implemented in the revised version of the manuscript.

- Table 2 : how are the column of the 2 sediments different? link with pelites/calcites or with sediments 1/2 of Figure 6?

This has been changed in the figure legends.

- Table 2 : why no diffusion creep in the crust ?

At low temperatures, the strain rate is a nonlinear function of the stress, which suggests that the active deformation mechanism of the crust is likely dominated by dislocation creep. Diffusion creep is usually active at high temperatures and low deviatoric stresses and is therefore more important in the upper mantle.

- Table 2 : which rock are analogue for strong and weak crust?

   The strength of crustal rocks depends on temperature, pressure and deformation rates. In the models presented here, the strength of the weak and strong layers should be regarded relative to each other. They represent a more heterogeneous crust, which is more realistic than a unified homogenous material for the entire crust. The weak layers represent for example silica-rich metasediments and the strong units represent for example mafic material (see also (Petri, et al., 2019) for more details).

- section 3.1.1: how do you define the length of the margin (threshold in crust thickness?)

   Thickness reduction from original thickness to <10 km following (Sutra & Manatschal, 2012).

- line 160: is the second invariant tensor of the deviatoric stress calculated for the whole lithosphere including the crust?]

   Yes, we clarified this in the revised version of the manuscript.

- why do you take the 10^21 Pa.s contour as the base of the lithosphere? Why not take th 1350◦C isotherm?

   The viscosity contour remains horizontally straight, whereas the 1350 °C isotherm is deflected by the convection cells indicating mantle material flow rather than a rigid plate boundary. Above the viscosity contour the length of the velocity vectors is essentially zero, but below this contour line the convection cells are active. We have clarified this in the revised version of the manuscript.

- line 176: give X-location of special flow field at 120 km depth

   This has been clarified in the revised version of the manuscript.

- line 184: why do you claim that the lithosphere is delaminating whereas the iso-viscous contour is almost flat?

   This has been addressed in the revised version of the manuscript.

- line 207: define GPE

A definition of GPE is given in line 197. A more detailed explanation is given in the appendix.

- line 223: Figure 8d rather than 8b?

Yes, this has been changed in the revised version of the manuscript.

- rephrase "In our models, subduction is initiated self-consistently, without prescribing any major weak zone or an already existing slab." (line 286) since they are weak heterogeneities in the passive margin

We argue that in our models, subduction is initiated "self-consistently", because we do not ad-hoc prescribe any major weak zone cutting through the entire lithosphere at the passive margin. Also, in our initial model configuration, we do not impose any major weak zones, or seeds, to force mantle exhumation and separation of the continental crust. The heterogeneities at the passive margin have been modelled self-consistently within the same continuous numerical simulation. Also, the layered heterogeneities are only present in the crust and no major heterogeneities are imposed in the mantle lithosphere. We modified the text to clarify our statement.

- line 319 "if shear stresses are negligible" = is that really the case at subduction onset?

Horizontally far away from the subduction zone, which is where we calculate the force, this assumption is valid.

- equation A3: define alpha and beta (which is different from the beta in Eq. B2 I guess...)

This has been corrected in the revised version of the manuscript.

- Fig. 1a,d: the depth looks smaller than 680 km

Indeed, this is not clear enough in the model configuration: depth is -660 km, surface level is 0 km and +20 km are left free to allow for topography. This has been addressed during the review process.

- Fig. 1c: initial random perturbations look denser between -20 and + 20 km, is that the case?

This has also been clarified during the review process.

- Fig. 3d: issue with the bottom of the plit nera -200 km (vertical grey line?)

I do not see the vertical grey line. The comment is not clear enough.

- Fig. 5: dashed lines for M4 and M5 are barely visible, I suggest you use thick lines with other colours

    We have changed the line style and colour for the figure in the revised version of the manuscript.

- Fig. 8: it would be helpful to have label on the topography such as "trench", and to mark the subduction initiation in the timeline of Figure 8d

    This suggestion has been implemented in the revised version of the manuscript.

- Fig. 10: what is the new information brought by this figure compared to Fig. 4?

    Figure 10 shows the conductive heat flow of the entire domain. Whenever convection is modelled properly, the conductive heat flow in the upper mantle must be close to 0. In models M4, M5 the conductive heat flow is still high due to the enhanced thermal conductivity. This illustrates, that this approach mimics the convective thermal structure but does not capture the physical process of convection in the upper mantle correctly. We have combined figure 10 with figure 4 in the review process.

The abstract should be shortened to include the top 3 most important results, and be revised for clarity and shorter sentences. For example, what do the authors want the paper be known/cited for?

     The Abstract has been shortened and reformulated during the review process in order to address the reviewer's comment.

In the introduction, the link between mantle convection and lithosphere deformation is quite abrupt (with a sentence about age of the Earth that is irrelevant to this study). The question 'why is convection important? 'is not satisfactorily introduced or linked to coupled lithosphere-mantle deformation.

     Convection regulates the long-term temperature and mechanical structure of the lithosphere: the strength of the lithosphere is inter alia temperature dependent. Thus, convection may have a direct impact on the deformation of the lithosphere. Coupling convection to lithosphere deformation in numerical models can therefore improve our understanding of lithospheric scale processes, such as rifting and subduction.
Also, convection can generate forces due to up- and down-welling of mantle material, which can affect lithosphere deformation.
A paragraph has been added to the introduction for better explanation.

General suggestion: too many commas. Try to rephrase/split sentences with more than 2 commas or that are longer than 2 lines.

     This has been implemented in the revised version of the manuscript.

2. Results section. I think the reference model (M1) should be described separately (evolution between extension, relation, convergence). Then compare models M2-M6 with M1 to highlight the effect of various factors. Figures should be adapted accordingly. The reason for this are the following: - in current form, the comparison is all over the place and it is confusing. It is not very clear which simulations the main text is referring to sometimes. - the current arrangement of figures is random. It starts with 2, 5, 8, 4, 10 etc. Their placement should follow a logical order of arguments.

     We have changed the structure of the manuscript as follows:
Results section:

- The evolution of the reference run and M6 is now described separately (two standalone figures)
- M2-5 are then compared to M1 at each deformation stage.
- Figures have been modified and reordered accordingly

Discussion section:
We have restructured the discussion section and are now discussing implications of the models for several geodynamic problems including:
- Spontaneous vs. Induced subduction initiation
- Mantle convection stabilising single-slab subduction
- Onset of upper mantle convection and thermo-mechanical evolution of the lithospheric plates
- Impact of mantle viscosity structure and effective conductivity on passive margin formation

The comparison between M1-M6 should be done in terms of Ra. The k, viscosity cutoff, flow laws, they essentially affect the Ra.

We have largely implemented this suggestion in the revised version of the manuscript.

3. Thermal softening. A quick search in the manuscript finds 'thermal softening 'only in the abstract, very late discussion and conclusion, yet it is suggested as a key process that controls subduction initiation. I'm pointing out that it is incompletely described and linked to the hypothesis of the study and results.
For example, Line 425: thermal softening is introduced only now. not clear why bring it up here?
Line 461: say that structural and thermal softening are important, but they were intro- duced late, without much context.
Moreover, the authors suggest in multiple places that it is the structural softening (in- heritance) and convection (slab suction) that help initiation. The authors need to clarify what are the main findings, and arguments need to be revised. One finding that I think is important: the required driving force to initiate subduction is much smaller, when convection and structural inheritance are considered.

We have discussed thermal and structural softening in more detail in the revised version of the manuscript.

4. Modelled vs parameterised convection. In Line 126, 3 types of simulations are introduced: 1) model convection with a weak asthenosphere, 2) parametrised convection, by scaling the thermal conductivity to the Nusselt number 3) impact of different viscosity structures
First, the treatment of the mantle convection is not clear in the main text (Lines 124- 130). What drives convection? How is the applied parametrised convection different? When is the onset of convection? Is convection only during the thermal relaxation stage? What controls the size of the convection cells? Also, explain how the Ra_avg is calculated.

This has been addressed during the review process. The Rayleigh number is calculated locally at each grid point. Ra_avg is the arithmetic average of all local Rayleigh numbers > 1000. We have clarified in the revised version of the manuscript.

In point 3) above which approach are you using: modelled/parameterized convection? While it is explained better in Appendix B, the differences between them are not clear in the main text. For example, 1) would be M1, while 2) is M, and 3) is M6?

This was explained in lines 130-135, but we changed the numbering of the models in the revised version of the manuscript, for clarity.

5. Other questions.

The geodynamic cycle modelled: 1) 30 Myrs extension at 2cm/yr 2) 70 Myrs thermal relaxation 3) 20 Myrs convergence at 3 cm/yr
What is the motivation behind these choices: 1) why thermal relaxation 2) why those time intervals 3) why those extension/convergence rates? Also, what are the boundary conditions during thermal relaxation?

The aim is to model the opening of a ca. 400 km wide oceanic basin without formation of a mature oceanic crust in an ultra-slow to slow spreading rift system. The durations of the periods and boundary conditions are chosen to allow for comparison of model results to orogens that formed from the collision of magma-poor hyper-extended margins, such as the European Western and Central Alps. We have clarified this in the revised version of the manuscript.

Why the choice of those parameters to change?

The viscosity structure of the mantle is poorly constrained and has a direct impact on the convective flow of the mantle. The effective conductivity approach is used to stabilise the thermal field in numerical simulations, but its impact on the deformation of the rigid plates and on self-consistent subduction initiation has not been tested yet. These statements have been added as a motivation in the introduction.

Are the surface processes important? Have you run models without? Do they introduce further heterogeneities in the model that affect the outcome?

Testing in more detail coupled surface processes to the deformation in the lithosphere and convection in the upper mantle is beyond the scope of this study. We have included only a simple parameterisation of surface process into the model to avoid unrealistically high and low topography. This has been clarified in the revised version of the manuscript.

6. Subduction initiation. It seems like symmetric vs asymmetric spreading also controls to a large extent subduction initiation, whether it is single/double subduction. I feel very little discussion is about that, and more on structural and thermal softening.

Subduction is always initiated during convergence. Most inheritance from the spreading is restricted to the margin geometry and is thus structural. Localisation occurs in the lithospheric mantle beneath the margins due stress concentration at the beginning of the necking zone. The heterogeneity in the upper mantle introduced by the convection and, therefore, the suction force of downward directed material flow seems to influence whether there is single or double-sided subduction. This has been clarified during the review process.

Also, there are other previous efforts to model extension/compression to obtain structural inheritance and subduction initiation (i.e. Gulcher et al 2019). The authors discuss simpler treatments of subduction initiation in paragraph 280, but do not relate to newer efforts to avoid the use of artificial features. So, are these newer models better for studying subduction initiation?

(Gülcher, Beaussier, & Gerya, 2019) investigate detachment faults as potential weak zones for intra-oceanic subduction initiation, not at a passive margin as wee do. Nevertheless, such detachment faults might be a geologically observable weak zone at which subduction could be potentially initiated. This topic has been addressed in the introduction of the revised manuscript.

Minor points:
Line 8-10: revise sentence

We have incorporated this suggestion in the revised version of the manuscript.

Line 10: only from the abstract it is not clear what the parameters were used, so saying that a viscosity of 5e20 Pa.s was used (as compared to what?) is not very meaningful. Rephrase

We have rephrased the abstract in the revised version of the manuscript

Line 20: multiple use of 'geodynamic 'in the same sentence Line 29-30: rephrase.

We have rephrased this sentence in the revised version of the manuscript.

Line 31: while it is an interesting fact - the calculation of the age of the earth - is not very relevant to the manuscript.

We have deleted this sentence in the revised version of the manuscript.

Line 35: rephrase

We have rephrased this sentence in the revised version of the manuscript.

Line 40: unlikely to be problematic

We have changed this accordingly in the revised version of the manuscript.

Line 41: delete likely

We have changed this accordingly in the revised version of the manuscript.

Line 55: authors relate to numerical aspects such as time step size, without mentioning why? The context was on physical aspects of convection.

This has been moved to the appendix in the revised version of the manuscript.

Line 64-65: should be in the first paragraph of introduction

This suggestion has been implemented in the revised version of the manuscript.

Line 68: Why only upper mantle? This is discussed late in discussion (section 4.4, paragraph 395)

This has been moved to the introduction section in the revised version of the manuscript.

Line 69: delete 'of applying'

This has been changed in the revised version of the manuscript.

Line 74: revise sentence - its meaning is not clear to someone who hasn't read the methods/results section.

We have restructured this part of the introduction in the revised version of the manuscript.

Line 88: reference to the code how it was benchmarked? (Info in appendix A, but should be in the main text too)

The benchmarks have been moved to the main text in the revised version of the manuscript.

Line 89: rephrase

We have rephrased this sentence in the revised version of the manuscript.

Line 96: repeats with Line 92, also Duretz et al 2016/2016a?

This has been changed in the revised version of the manuscript.

Line 105: based on the sentence the crust should be: 3*5+4*5 = 35 km thick. But a sentence earlier it is 33km

This mistake has been corrected and the description of the initial configuration has been clarified in the revised version of the manuscript.

Line 103: what is the mathematical expression for the perturbation? in case the model needs to be reproduced?

We have added the mathematical expression for the marker field perturbation to the algorithm description in the appendix of the revised version of the manuscript.

Line 111: more details on the rheology? Indicate appendix A for reference

We refer to the appendix in the revised version of the manuscript.

Line 113: reference to "corresponding laboratory flow law estimates"?

References are given in the footnote of table A1.

Line 113-114: rephrase. i.e. The mantle lithosphere is rheologically stronger than the mantle asthenosphere due to the temperature gradient.

With this sentence, we want to emphasize that we used the same material parameters for both the lithospheric and the upper mantle. We therefore keep the phrase as it is.

Line 120: what is the motivation for alternating between calcites and pelites for sedi- mentation algorithm?

To account for changes in sediment strength due to changing sedimentary environments. A detailed investigation on the impact of different implementations of surface processes is beyond the scope of this study. We have clarified this in the revised version of the manuscript.

Line 130 - give reference to Table 1.

This has been adapted in the revised version of the manuscript.

Line 133: viscosity cutoff for M1 is not provided to understand the difference.

The cut-off value is given in table 1.

Line 134: realistic value? Are the other values not realistic?

A thermal conductivity value of 36 is not realistic for a peridotite at upper mantle temperature and pressure conditions.

Line 148: Figure 2
-> can define a variable F = 2xtau_II

We have introduced a variable for the plate driving forces in the revised version of the manuscript.

Line 160: introduce the horizontal driving force per unit length, but what is it proxy for?

It is a proxy for the strength of the lithosphere and it indicates how much force is needed to localise deformation. Has been clarified during the review process.

Line 168: you can't see to a depth of 660 km as indicated

Figures have been adapted accordingly.

Line 195: values

This sentence has been rephrased during the review process.

Line 197: what is the delta GPE showing? (Info given in appendix)

The gravitational potential energy (GPE) has been explained in the appendix. Delta means that it is the difference of the GPE compared to a reference value, commonly the value close to one of the boundaries.

Line 224: reference to fig 9a, yet that figure is for extension stage.
Paragraph 220-225: confusing.

This has been addressed during the review process.

Line 228: what is mechanical heterogeneity? increases the strength of the weak layers

We have clarified this in the revised version of the manuscript.

Line 233: breaks later than the continental. after gives the impression of location.

This suggestion has been implemented into the revised version of the manuscript.

Line 234: what do you mean 'Mantle convection does not establish as early as rifting and crustal separation.'?

This sentence has been clarified in the revised version of the manuscript.

Lines 243, 245: use of realistic. close to the Ra estimated for the Earth. Line 250: which modulates mantle velocities.

We have implemented this suggestion in the revised version of the manuscript.

Line 250-254: why the discussion on time step size (a numerical feature) here? Lines 255-257: which simulation results are the authors referring here?

This has been addressed in the revised version of the manuscript

Paragraph 258: reference figure 5e,j in this paragraph. Also, maybe plot density aver- ages in passive margins/exhumed mantle separately?

We have tried to modify this figure, but the figure becomes too busy.

Line 272: you jump from density differences to values of tectonic forces. An additional sentence needs to connect them (i.e. estimate the buoyancy force due to modelled density differences). How much is needed to initiate subduction? (a similar calculation is done in line 315)

This depends on the strength of the lithosphere, which is still subject to debate.

**Paragraph 280: this should come before the Cloos 1993 paragraph**

We have restructured the discussion accordingly during the review process.

**Line 286: yes, but under convergence**

When a major weak zone is imposed, one also has to push from the sides to initiated subduction. Only when you skip the process of subduction initiation and already assume the presence of an inclined slab, subduction continues freely (given that the initial slab is long enough and that boundary conditions have been chosen correctly).

**Line 294-295: total convergence is double sided, while in M1 is single-sided (asym- metric). Not clear why subduction initiation is stable only in M1. Convection cell size important? how about thickness of lithosphere at the point? M2-M5 are quite symmet- ric and they all have Ra_avg ~1e5, while M1 has Ra_avg~1e6. That should have an effect.**

Likely the distribution of cells is important: the more asymmetric the mantle flow the more the model tends to produce single-slab subduction rather than double-slab subduction. Asymmetry decreases with decreasing Ra. This has been explained in more detail during the review process.

**Line 312-315: - suction force induced by down-welling in the convection cell in M1. What is the similar force in the other simulations?**

It is likely similar in M2-3 compared to M1, because of similar density distributions and vertical velocities, but it has not been computed here. Due to higher temperatures induced by the effective conductivity, values for densities are lower in M4-5 compared to M1-3, which explains the reduced absolute speed of material in the convecting cells. Most important seems to be the asymmetric distribution of the cells. We have clarified this during the review process.

**Line 330: not sure what the reference is for. The double-subduction term was not coined by those workers.**

We cited this reference, because they also modelled double-sided subduction.

**Line 332: sentence not clear. Which simulation are you referring? would say M2-5 are more or less symmetric double subduction**

Has been reformulated during the review process.

**Line 343: onset of convergence - unclear when this happens?**

This happens at 100 Myrs. We have clarified this in the revised version of the manuscript.

**Line 357-361: use of 'realistic'**

This has been rephrased in the revised version of the manuscript.

**Paragraph 395: this paragraph should be in the methods, as it motivates/explains your model domain until 660 km. The sentence 'The convective patterns simulated in our study are in agreement with these observations. 'is irrelevant because you don't model the lower mantle.**

This paragraph has been moved to the introduction in the revised version of the manuscript.

Paragraph 410: this should come earlier - I had questions about it earlier. on previous work on subduction initiation.

This paragraph has been moved to the introduction in the revised version of the manuscript.

Line 418: most definitely will have an impact

We have implemented this suggestion in the revised version of the manuscript.

Paragraph 430: and melting

We have accounted for this suggestion in the revised version of the manuscript.

Line 444-446: rephrase/simplify.

We have rephrased the conclusion section in the revised version of the manuscript.

Figures and Tables: Table 1: thermal conductivity should be 'k 'without the 'th 'sub- script. The authors can also provide the formula for the Ra number in the main text. How was the Ra_avg calculated?

This suggestion has been implemented in the revised version of the manuscript.

Table 2: there should be a column 'Description 'to describe the meaning of each pa- rameter i.e. 'rho0 '- reference density. Use k instead of k_th for thermal conductivity. What is dry/wet mantle? I assume wet mantle applies only to M6? Plastic and elastic parameters are also listed. Not very clear in the main text.

Description of the parameters is given in the appendix section. We also moved the table to the appendix for better comprehension. Dry and wet refer to the rheological parameters for dry and wet olivine. In the references given below the (now) appendix table 1 it is explained in which model we use either wet or dry rheologies.

Figure 4: why plot the vertical velocity field separate from the horizontal? should plot arrow/streamlines field to see the convection cells.

The figure has been adapted accordingly in the revised version of the manuscript.

Figure 10: should be merged with Figure 4. One column velocity, one column temperature.

These two figures have been merged in the revised version of the manuscript.

Figure 5: What if you plot the profiles at the rift axis (within a distance) vs off-axis on either flanks of the rifts? Caption: g-j show enlarged areas.

We have tested this version of the figure, but it becomes too busy, unfortunately.

Figure 6: legend: temp contours are red.

We have corrected this mistake in the revised version of the manuscript.

Figure 8: the line plots are not entirely clear. Maybe use a dotted line instead of dashed line? and same thickness.

We have changed line style and colour for the figures in the revised version of the manuscript.

Appendix A
Line 481: that's a strange notation of i,j indices (Einstein notation).
Eq A2: if written in Einstein notation, then vectors are written in terms of scalar com- ponents (a_i should not be bold). Same in Line 482 a=[0,g]. -> revise this appendix for completeness of sentences, and explanation of all parameters. For example, what is Ap, tauP etc. Gamma value? in eq A10

We have clarified these points in the revised version of the manuscript.

Appendix B
Paragraph 531: rephrase

This suggestion has been implemented in the revised version of the manuscript.

Line 560: not clear

This has been clarified in the revised version of the manuscript.

Appendix C
Line 595: gamma_T=1? Line 601: g=10ˆ4?

To perform this benchmark one has to apply a local Rayleigh number of 100 at the top and using the Frank-Kamenetskii approximation a local Rayleigh number of 10^7 at the bottom. Choosing the dimensionless values as it is done here matches those numbers and reproduces the desired pattern with diagnostic quantities that are in the range of values reproduced by other algorithms as tested in (Tosi, et al., 2015).

**List of most relevant changes**

**Text:**

- Abstract has been shortened and reformulated highlighting the most important findings
- Introduction has been rewritten and restructured:
  - o Definition of convection and description of convection in the Earth's mantle
  - o Motivation for studying coupled lithosphere-mantle deformation
- Method section:
  - o More detailed description of the initial configuration and assumptions made
  - o Table 2 has been moved to the appendix
- Results section:
  - o Evolution of reference model and Wet Olivine model are described separately
  - o Results of other models are presented in comparison to the reference model for the different deformation stages
- Discussion section has been completely restructured discussing:
  - o Spontaneous vs. induced subduction initiation and estimates for plate driving forces
  - o Mantle convection stabilising single-slab subduction
  - o Onset of upper mantle convection and thermo-mechanical evolution of the lithospheric plates
  - o Impact of mantle viscosity structure and effective conductivity on passive margin formation
  - o Mantle convection, thermal erosion and tectonics in the Archean
  - o Comparison with estimates of Earth's mantle viscosity and thermal structure
  - o Formation and reactivation of magma-poor rifted margins: potential applications
- Conclusion has been shortened and reformulated
- Appendix: more detailed explanation on the assumptions made:
  - o Justification for extended Boussinesq approximation
  - o Justification for neglecting latent heat
- Mistakes have been corrected and paragraphs have been rephrased for clarity

**Figures:**

- Velocity glyphs have been added to figures for better visualisation of the material flow field
- Reference model and the Wet Olivine model are shown in separate figures
- Figures have been rearranged in a logical order following the structure of the revised version of the manuscript
- Figure 10 has been merged with figure 4
- Colour and style of lines have been changed for better visualisation

[revised manuscript text omitted]
$ $-1$ $-1$ 1050 1050 1050 1050 1050 1050 $k$ $-1$ $-1$ 2.25 2.25 2.37 2.55 2.75 2.75 $H_R$ $-3$ $0.9\times10^{-6}$ $0.9\times10^{-6}$ $0.56\times10^{-6}$ $2.9\times10^{-6}$ $2.1139\times10^{-8}$ $2.1139\times10^{-8}$ $C$ $10^7$ $10^6$ $10^7$ $10^6$ $10^7$ $10^7$ $\varphi$ 30 5 30 5 30 30 $\alpha$ $-1$ $3\times10^{-5}$ $3\times10^{-5}$ $3\times10^{-5}$ $3\times10^{-5}$ $3\times10^{-5}$ $3\times10^{-5}$ $\beta$ $-1$ $1\times10^{-11}$ $1\times10^{-11}$ $1\times10^{-11}$ $1\times10^{-11}$ $1\times10^{-11}$ $1\times10^{-11}$ Dislocation $A$ $-n-r$ $-1$ $5.0477\times10^{-28}$ $5.0717\times10^{-18}$ $1.5849\times10^{-25}$ $10^{-138}$ $1.1\times10^{-16}$ $5.6786\times10^{-27}$ $n$ $-$ 4.7 2.3 4.7 18 3.5 3.5 $Q$ $-1$ $485\times10^3$ $154\times10^3$ $297\times10^3$ $51\times10^3$ $530\times10^3$ $460\times10^3$ $V$ $3$ $-1$ 0 0 0 0 $14\times10^{-6}$ $11\times10^{-6}$ $r$ $-$ 0 0 0 0 0 1.2 $f_{H_2O}$ 0 0 0 0 0 $10^9$ Diffusion $A^*$ $-n-r$ $m$ $-1$ - - - - $1.5\times10^{-15}$ $2.5\times10^{-23}$ $n$ - - - - - 1 1 $Q$ $-1$ - - - - $370\times10^3$ $375\times10^3$ $V$ $3$ $-1$ - - - - $7.5\times10^{-6}$ $20\times10^{-6}$ $m$ - - - - - 3 3 $r$ - - - - - 0 1 $f_{H_2O}$ - - - - 0 $10^9$ $d$ - - - - $10^{-3}$ $10^{-3}$ Peierls $A_P$ $-1$ - - - - $5.7\times10^{11}$ $5.7\times10^{11}$ $Q$ $-1$ - - - - $540\times10^3$ $540\times10^3$ $\sigma_P$ - - - - $8.5\times10^9$ $8.5\times10^9$ $\gamma$ - - - - - 0.1 0.1

[revised manuscript text omitted]

$$Nu = \frac{q_{\text{Adv}}}{q_{\text{Dif}}} = \frac{q_{\text{LAB}}}{\frac{k_{\text{th}}(T_{\text{M}}-T_{\text{LAB}})}{D}} \; , \cdot \tag{B3}$$

930  Using this relationship, it is possible to scale the thermal conductivity to the Nusselt number of the Earth's mantle and to maintain a constant heat flow through the base of the lithosphere via conduction when convection is absent. Assuming  $Ra = 2 \times 10^6$ and $\beta = 1/3$ for the Earth's upper mantle convection, eq. B2 predicts $Nu = 13$. This implies that the heat flow provided by advection is $13\times$ higher than the heat flow provided by conduction.
935   Using an effective conductivity approach, the heat flow provided by advection is mimicked using an enhanced conductive heat flow in the upper mantle . The effective conductivity can be determined by scaling the standard value of thermal conductivity of the upper mantle material to the
940  Nusselt number of the convecting system like

$$k_{\text{eff}} = Nu \, k. \tag{B4}$$

For this study, the  standard value for $k_{\text{th}} = 2.75$ and

$$q_{\text{eff}} = k_{\text{eff}} \frac{T_{\text{LAB}} - T_{\text{bot}}}{D}$$

[revised manuscript text omitted]

---

## Author Response (AR2)

Relevant changes made during the review process:
- Title has been changed
- Abstract has been restructured and shortened
- Introduction and model description has been rearranged
- A figure has been added showing density contrasts (fig. 6(f)-(j))
- Discussion section has been rearranged
- Some equations have been reformulated
- Two videos have been uploaded to the AV Portal of TIB Hannover. Once the videos have received a DOI, we will add them to the manuscript

The authors gratefully thank Zhong-Hai Li for the constructive comments and questions. We have listed the original comments by the referee marked in black (RC) and below the corresponding replies of the authors marked in red (AR).

In the manuscript by Candioti et al., a series of 2D thermo-mechanical models are used to study the successive processes from continental break-up and oceanic lithospheric cooling to the following subduction initiation under compression. In addition, the mantle convection is included in the model, which is shown to play important roles in regulating the whole model evolution, especially may facilitate the formation of asymmetric, one-sided subduction initiation. The study is interesting and the clearance of the manuscript is greatly improved by considering the comments/suggestions in the first round of review. Thus, I believe it is worth for the final publication. Since it is the first time for me to read the manuscript, I do have some additional comments for further improvement of the paper as shown below.

Major points:

RC: (1) Sections 3.3-3.5, comparisons of M1-M5: The one-sided versus two-sided subduction initiation (SI) is an interesting point of the study. The authors propose that the suction force induced by asymmetric down-welling controls the one-sided SI in M1. However, on the one hand, how large is the suction force in M1 comparing to those in other models M2-M5? On the other hand, it is still not clear about the dominance of this force comparing to other factors, e.g. the asymmetric passive margin, for the strain localization. In Figure 8, it clearly demonstrates that the strain localizations in M1 and M4 are both focusing on the single margin. I am curious why the single-sided SI is only developed in M1, but not in M4 (Figure 9). It may help better understanding the processes by checking the continuous movie of the model evolution. Please add a paragraph for discussion.

AR: In our models, subduction initiation via thermal softening requires local stress concentrations, which occur for example at the transition from the proximal margin to the necking domain. The more asymmetric the passive margin geometry, the higher are the differences in stress concentration locally between the two margins. This likely favours the formation of a single-sided subduction zone without any additional asymmetry in the upper mantle. The initial configuration of the models presented here leads to the formation of

rather symmetric passive margins in all models. Therefore, stress concentrations are distributed horizontally quite symmetric. In consequence, two shear zones form in all models presented here, one below each passive margin. Any additional asymmetry in the model is then decisive for whether one or both shear zones evolve into subduction zones. For clarity, we have changed the figure arrangement. In the revised version of the manuscript, fig. 6(f)-(j) shows the differences of the density field w.r.t. the horizontally averaged density profile shown in fig. 2(e). Fig. 7(b) shows a horizontal profile of normalised density differences. These figures show that density differences are distributed symmetrically in M2-5. Only in M1, the distribution of density differences is laterally asymmetric. In the region between $x$=200 km and $x$=400 km, $z$=-250 and $z$=-100 a distinct body with a positive density anomaly can be identified (see black contour line in fig. 6(f) and arrows in fig. 7(b)). Integrating the density differences in M1 that lie within the 2 kg/m^3 contour in the region mentioned above yields a value for the suction, or buoyancy, force of 0.25 TN/m. Since the distribution of density differences is laterally symmetric and defining an integration area in M2-5 is not trivial, a calculation of suction forces is not attempted for M2-5.
In M4, subduction is initiated first only on the right side, but after ca. 10 Myrs of single-slab subduction, subduction is also initiated on the left side.

RC: (2) Section 3.6 and Appendix D, plate driving forces: The calculation of plate driving force is not very clear. Why is the Tau_II(x) integrated vertically over the whole model domain, ie. from the model bottom to the top surface (Equation D3), but argued in the main text that "the vertically-integrated second invariant of the deviatoric stress tensor Tau_II is a measure for the strength of the lithosphere"? I understand the deviatoric stress should be mainly localized in the lithosphere. Then why does it need to integrate over the whole model domain down to 660km? Could you compare the exact values?

AR: Using the second invariant of the deviatoric stress tensor to calculate the plate driving force F_D requires to define the integration bounds at levels where the shear stresses are negligible. This condition is strictly satisfied by the choice of boundary conditions at the top and the bottom of the model domain. We therefore chose to show the profiles which have been integrated over the entire domain height ($z$=-660). Since the deviatoric stresses below the lithosphere are indeed negligibly small, values for F_D integrated from depths of $z$=-660 km, $z$=-330 km and $z$=-120 km do not reveal significant differences. Therefore, the value of F_D is essentially independent on the integration depth (if deeper than the lithosphere thickness).

RC: (3) An important finding of this study is the decreased pushing force (i.e. 15 TN/m) required for SI at passive margins, comparing to the previous works with less heterogeneity (e.g. 37 TN/m in Kiss et al., 2020). However, the force still seems to be large. As shown in the current models, the mantle convection induced force could be about 2 TN/m. With the possible ridge push force of another ~3 TN/m (Turcotte & Schubert, 1982), it is still not enough. What kind of additional forces can be used to explain the prescribed boundary convergence? How about the suction force?

AR: Indeed, within the modelled domain the plate driving forces are too low to initiate subduction spontaneously. In our models, the boundary convergence is assumed to

be caused by far-field plate driving forces generated by global processes that are not modelled inside our model domain, such as slab pull, whole mantle convection, ridge push etc. For example, the closure of the Piemonte-Liguria basin is assumed to be caused by the much larger scale convergence of the African and European plates (McCarthy, et al., 2020). These large-scale processes could locally generate the required driving forces, also due to some geometrical focusing effects. Additionally, we suppose that the plate driving force necessary for horizontally forced subduction initiation in our models could be further reduced by considering more heterogeneities, or a smaller yield stress in the mantle lithosphere. However, the minimum value for the driving force, which can still generate horizontally forced subduction initiation in our models has to be quantified in future studies.

Minor points:

- RC: Title: The phrase "convergence-induced subduction" may be modified to "convergence-induced subduction initiation", because only the subduction initiation process is studied.

    AR: We have changed the title of the revised version of the manuscript.

- RC: Abstract: The structure of the abstract is rather loose. For example, from the 3rd line to the 8th line, many sentences are used to describe the details of the model setups, which could be shortened. In addition, the organization of the main findings is also not very clear, which may be more logically organized as: 1) Our model generates a 120 Myrs long geodynamic cycle of subsequent extension (30 Myrs), cooling (70 Myrs) and convergence (20 Myrs) coupled to upper mantle convection in a single and continuous simulation. 2) The model results indicate that "xxx", where to show the impact of upper mantle convection on lithosphere hyper-extension, as highlighted in the title. 3) The subsequent, compression-induced subduction initiation should be horizontally forced, rather than vertically, pure gravity forced. 4) The single-sided SI versus double sided SI, as well as the controlling factors, i.e. the mantle convection and others. 5) The forces required for SI and related aspects. 6) Geological implications.

    AR: We have logically reordered and shortened the abstract.

- RC: Lines 30-35, "The 660-km phase transition can therefore represent a natural impermeable boundary, that": I do not think it is impermeable, since it is just a certain resistance due to the viscosity jump and a negative Clapeyron slope.

    AR: This sentence has been changed during the review process.

- RC: Lines 73-75, rephrase.

    AR: This sentence has been rephrased during the review process.

- RC: Line 124, "as mentioned above": which is it and where is it? It is too far away from this point.

AR: This sentence has been rephrased during the review process.

- RC: Line 135, "In these models": which models are they? The models you referred to in the last sentence, or the models you have conducted.

AR: This paragraph has been restructured during the review process.

- RC: Line 145, "The top surface is stress-free": are you using a free surface? Is the sticky air still used, since there is a 20-km domain above the crustal surface.

AR: We use a free surface boundary condition. No sticky air is employed in the models presented here. This has been clarified during the review process.

- RC: Section 3.1, Model-M1: the different margin widths of 200 and 150 km. I am wondering 1) is the model setup purely symmetric, and 2) the effect of the contrasting margin width on the single sided SI.

AR: The initial perturbation of the marker field is not perfectly symmetric (see appendix A). Regarding the impact of margin asymmetry, see also our reply to major point (1) above. This has been explained in more detail in the revised version of the manuscript.

- RC: Section 3.2, Model-M6: Actually, I do not know the purpose of this model with rather weak upper mantle and the resulting unstable lithosphere, which is definitely contrasting to the processes studied in this paper, ie. extension-cooling-SI. In the discussion of the paper, I finally find that this model is compared a bit to the Archean Earth, which is also not related to the focus of this study.

AR: M6 underlines the importance of better constraining the rheology of the mantle. In case the viscosity of the mantle lithosphere and the upper mantle is low, the model results are not applicable to present-day Earth, but may be important for the early Earth. We show the results for a weak mantle, because a weak mantle rheology is frequently applied in lithosphere-only models, particularly when discussing yield-strength envelops. Our models suggest that such weak mantle rheology is actually not feasible for present day plate tectonics, but this non-feasibility can only be observed when performing coupled mantle-lithosphere models as we have presented. In lithosphere-only models, a weak mantle would not generate the thermal erosion of the lithosphere bottom as observed in model M6, because the lithosphere bottom would be "stabilized" by the bottom boundary condition.

- RC: Section 3.6, FD is defined as 'per unit length': per unit length along-strike?

AR: The plate driving force is defined per unit length in the third dimension, because our models are only two-dimensional.

- RC: Line 298, "At ca. 105 Myrs, values for FD increase again until the end of the simulation." Why? Since the subduction has already formed, which should 'release' the force.

AR: At this stage, folding of the mechanically strong crustal layers occurs. Although folding is a structural softening mechanism (Schmalholz, Podladchikov, & Jamtveit, 2005) (Schmalholz & Mancktelow, 2016), when the folds become more and more isoclinal, the horizontal stresses in the crust increase again. This likely causes the increase of the value for $F_D$.

- RC: Section 4.1: I agree that it is very difficult for the passive margin to collapse spontaneously. However, a weak zone is generally considered for such kind of spontaneous SI model (e.g., Stern and Gerya, 2018, Tectonophysics), which may be briefly noted.

AR: It is possible to model spontaneous subduction initiation for an ad-hoc constructed passive margin geometry, if the employed mechanical resistance is small, the density difference between lithospheric and oceanic mantle is large, and/or if an additional weak zone is imposed. Such passive margin configurations are indeed unstable and lead to subduction initiation within a few million years (Stern & Gerya, 2018). However, the passive margins we consider in our study have been stable for at least 60 Myrs before subduction initiation. Therefore, spontaneous subduction initiation for unstable passive margins is in contrast with the observation of long-term stability of the ancient Alpine Tethys margins (McCarthy, et al., 2020) and the recent passive margins in the South Atlantic (Müller, Sdrolias, Gaina, & Roest, 2008). Modelling long-term geodynamic cycles, applicable to the evolution of the Alpine Tethys and the South Atlantic, requires appropriate density and rheological models which generate passive margins that are stable for more than 60 Myrs (Alpine Tethys) or more than 180 Myrs (South Atlantic). To evaluate whether models of spontaneous SI at passive margins are feasible, these models need to explain why the passive margins have been stable for more than 60 Myrs and only afterwards "collapse" spontaneously, although they are cooled and mechanically strong.

- RC: Line 329, "In our models, subduction is initiated self-consistently": I do not know whether it can be regarded as 'self-consistently', since the model is pushed by prescribed boundary velocities.

AR: Indeed, it is necessary to activate convergence to initiate subduction in the models presented here. For clarification, we better explain the intended meaning of the term "self-consistently" in the following sentences of the manuscript.

- RC: Line 385, the suction force is very large. Firstly, the approximation of suction force is kind of dependent on the size of the domain for integration (Equation D4). Secondly, how much of the gravity anomaly induced force (defined as suction force in this ms) acting the passive margin, ie. to facilitate SI. It may be difficult to quantify, but a short discussion may be necessary.

AR: As mentioned correctly, the high value for the suction force results from integrating the density differences over a too large area. We have now better constrained the integration boundaries by defining the area of the high-density anomaly below the right margin in M1 according to contour lines of density differences. The discussion section has been updated accordingly.

- RC: Line 505, the divergent subduction of Adriatic plate: Did the SI on both sides start at the same/similar time? If they are not, then it is not quite applicable to the current model.

AR: For the Adriatic plate, the subduction started significantly earlier below the Dinarides compared to the westward directed subduction. Since about 30 Ma, the Adriatic plate undergoes a divergent double slab subduction, for which the two subduction zones have started at different times. In model M4, subduction initiation does not occur simultaneously below both margins. The subduction initiation below the left margin occurs ca. 10 Myrs after subduction initiation below the right margin. During the evolution of the model, the subduction switches from the right margin to the left margin and then back again. Therefore, divergent double-slab subduction does not require that subduction initiation occurs at the same time. For clarity, we have added a video supplement that shows the entire evolution of M1 and the evolution of M4 during the convergence stage. To be clear, the current model M4 is not directly applicable to the timing of double subduction of the Adriatic plate, which additionally involves a complex 3D evolution.

**Impact of upper mantle convection on lithosphere hyper-extension and subsequent  __horizontally forced__ subduction __initiation__**

Lorenzo G. Candioti[1], Stefan M. Schmalholz[1], and Thibault Duretz[2,1]

[1]Institut des sciences de la Terre, Bâtiment Géopolis, Quartier UNIL-Mouline, Université de Lausanne, 1015 Lausanne (VD), Switzerland
[2]Univ Rennes, CNRS, Géosciences Rennes UMR 6118, Rennes, France
**Correspondence:** Lorenzo G. Candioti (Lorenzo.Candioti@unil.ch)

**Abstract.**

 Many plate tectonic processes, such as subduction initiation, are embedded in long-term (>100 Myrs) geodynamic cycles often involving subsequent phases of extension, cooling without plate deformation and convergence. However, the impact of upper mantle convection on lithosphere dynamics during such long-term cycles is  __still poorly__ understood. We have designed two-dimensional  upper mantle-scale (down to a depth of $660\,\mathrm{km}$) thermo-mechanical numerical models of coupled lithosphere-mantle deformation. We consider visco-elasto-plastic deformation ~~and for the lithospheric and upper mantle__ __including__ a combination of diffusion, dislocation and Peierls creep __law mechanisms__. Mantle densities are calculated from petrological phase diagrams (Perple_X) for a Hawaiian pyrolite. __Our models exhibit realistic Rayleigh numbers between $10^6$ and $10^7$ and model temperature, density and viscosity structures agree with geological and geophysical data and observations.__ We tested the impact of the viscosity structure in the asthenosphere on upper mantle convection and lithosphere dynamics. __We also compare models in which mantle convection is explicitly modelled with models in which convection is parameterized by Nusselt number scaling of the mantle thermal conductivity.__ Further, we quantified the plate driving forces necessary for subduction initiation in 2D thermo-mechanical models of coupled lithosphere-mantle deformation. Our model generates a $120\,\mathrm{Myrs}$ long geodynamic cycle of subsequent extension ($30\,\mathrm{Myrs}$), cooling ($70\,\mathrm{Myrs}$) and convergence ($20\,\mathrm{
[revised manuscript text omitted]